# Pan-cancer whole-genome comparison of primary and metastatic solid tumours

Francisco Martínez-Jiménez[1,2,3], Ali Movasati[1,5], Sascha Remy Brunner[1,5], Luan Nguyen[1,4,5], Peter Priestley[4], Edwin Cuppen[1,3✉] & Arne Van Hoeck[1]

Metastatic cancer remains an almost inevitably lethal disease[1–3]. A better understanding of disease progression and response to therapies therefore remains of utmost importance. Here we characterize the genomic differences between early-stage untreated primary tumours and late-stage treated metastatic tumours using a harmonized pan-cancer analysis (or reanalysis) of two unpaired primary[4] and metastatic[5] cohorts of 7,108 whole-genome-sequenced tumours. Metastatic tumours in general have a lower intratumour heterogeneity and a conserved karyotype, displaying only a modest increase in mutations, although frequencies of structural variants are elevated overall. Furthermore, highly variable tumour-specific contributions of mutational footprints of endogenous (for example, SBS1 and APOBEC) and exogenous mutational processes (for example, platinum treatment) are present. The majority of cancer types had either moderate genomic differences (for example, lung adenocarcinoma) or highly consistent genomic portraits (for example, ovarian serous carcinoma) when comparing early-stage and late-stage disease. Breast, prostate, thyroid and kidney renal clear cell carcinomas and pancreatic neuroendocrine tumours are clear exceptions to the rule, displaying an extensive transformation of their genomic landscape in advanced stages. Exposure to treatment further scars the tumour genome and introduces an evolutionary bottleneck that selects for known therapy-resistant drivers in approximately half of treated patients. Our data showcase the potential of pan-cancer whole-genome analysis to identify distinctive features of late-stage tumours and provide a valuable resource to further investigate the biological basis of cancer and resistance to therapies.

Metastatic spread involves tumour cell detachment from a primary tumour, colonization of a secondary tissue and growth in a hostile environment[1,2]. Advanced metastatic tumours are frequently able to resist aggressive treatment regimes[6]. Despite the many efforts to understand these phenomena[3,7–9], we still have limited knowledge of the contribution of genomic changes that equip tumours with these extraordinary capacities. Thus, it is essential to characterize genomic differences between primary and metastatic cancers and quantify their effect on therapy resistance to understand and harness therapeutic interventions that establish more effective and more personalized therapies[10].

Although extensive whole-genome analyses of primary or metastatic tumour types have been conducted[4,5], large-scale comparative studies between the two tumour stages remain limited due to the logistical challenges associated with obtaining pan-cancer cohorts of primary and metastatic cancers. To circumvent this issue, most comparison studies have relied on unpaired whole-exome data or have adopted more targeted approaches with a specific focus on driver gene landscapes[11–13]. However, these efforts have frequently involved separated processing pipelines for primary and metastatic cohorts, complicating the analysis of genomic features that are highly sensitive to the selected data-processing strategy[14,15]. A recent study that uniformly analysed more than 25,000 tumours[16] has provided a comprehensive overview of the genomic differences, driver alteration patterns and organotropism using clinical gene-panel sequencing as a base. However, this genomic analysis approach prevented the exploration of the full spectrum of genomic alterations that have a role in tumorigenesis, such as structural variation and mutational scarring.

## Harmonized whole-genome-sequenced tumours

Here we created a uniformly processed whole-genome-sequenced (WGS) inventory of 7,108 matched tumour and normal genomes from two unpaired primary and metastatic cohorts. We first collated the Hartwig Medical Foundation (Hartwig) dataset[5], which included 4,784 samples from 4,375 patients with metastatic cancers. Then, we reprocessed 2,835 primary tumour samples from the Pan-Cancer Analysis of Whole Genomes (PCAWG) consortium[4] using the open-source Hartwig analytical pipeline[5,17] to harmonize somatic calling and annotations of

[1]Center for Molecular Medicine, Oncode Institute, University Medical Center Utrecht, Utrecht, The Netherlands. [2]Vall d'Hebron Institute of Oncology (VHIO), Barcelona, Spain. [3]Hartwig Medical Foundation, Amsterdam, The Netherlands. [4]Hartwig Medical Foundation Australia, Sydney, New South Wales, Australia. [5]These authors contributed equally: Ali Movasati, Sascha Remy Brunner, Luan Nguyen. ✉e-mail: e.cuppen@hartwigmedicalfoundation.nl

events, and to eliminate processing biases (Extended Data Fig. 1 and Supplementary Table 1). Reassuringly, per-sample comparison of the number of single-base substitutions (SBSs), double-base substitutions (DBSs), indels (IDs) and structural variants (SVs) revealed a strong agreement between our results and the consensus calls originally generated by the PCAWG consortium (Supplementary Note 1). In addition, our processing pipeline strategy was minimally affected by differences in sequencing coverage, enabling a reasonable comparison of WGS samples from heterogeneous sources (Supplementary Note 1). A total of 7,108 tumour samples from 71 cancer types met the processing pipeline quality standards (Methods and Extended Data Fig. 1a) and constitutes one of the largest publicly available datasets of WGS tumours.

We next focused on 23 cancer types from 14 tissues with sufficient sample representation, comprising 5,365 tumour samples (1,914 primary and 3,451 metastatic) to explore genomic differences between primary and metastatic tumours (Fig. 1a and Extended Data Fig. 1a). Within this dataset, patients with metastatic tumours were slightly older at biopsy than patients with primary tumours (mean of 1.67 years older across all cancer types), although patients with metastatic prostate and thyroid carcinomas, and diffuse large B cell lymphoma were markedly older than their primary counterparts. Consistent gender proportions were observed across all cancer types except for thyroid adenocarcinomas (metastatic: 72% male and 28% female; primary: 25% male and 75% female). Treatment information was available for 83.7% of patients with metastatic tumours, which is essential to gauge specific treatment-induced contributions to genomic differences between primary and metastatic tumours (Fig. 1a). Finally, biopsy locations were annotated for 84.2% of patients with metastatic tumours (12.2% from metastatic lesions in the primary tissue (local), 16.2% in lymph nodes and 55.7% in distant locations) and displayed a highly tumour-type-specific distribution pattern, probably reflecting both the dissemination patterns of the tumours and the accessibility for safe clinical sampling.

## Comparison of global genomic features

We first explored global genomic differences between primary and metastatic tumours across the aforementioned 23 cancer types. Metastatic tumours showed an overall increase in clonality compared with their primary tumour counterparts (Fig. 1b). Particularly, five cancer types had a significantly higher metastatic average clonality ratio, ranging from 13.6% increased mean clonality in pancreatic carcinoma to 37.2% in thyroid carcinoma. Within the group of patients with metastatic breast carcinoma, distant and lymph node tumour biopsies showed significantly higher clonality ratios than local metastatic lesions (Fig. 1c and Supplementary Note 2). This increase in clonality was also observed in distant tumour biopsies of oesophageal and colorectal carcinomas (Fig. 1c). Nevertheless, the biopsy location did not influence tumour clonality in other cancer types such as lung adenocarcinoma and skin melanoma (Extended Data Fig. 1b), suggesting that patterns of tumour dissemination are highly tumour-type specific[9]. Our results support the model that metastatic lesions generally have lower intratumour heterogeneity[16], which may be explained by a single major subclone seeding event from the primary cancer and/or by severe evolutionary constraints imposed by anticancer therapies.

Comparison of chromosome arm aneuploidy profiles revealed a generally conserved portrait, which was strongly shaped by the cell of origin (Fig. 1d and Supplementary Table 2), supporting the notion that tumour karyotype is generally defined at early stages of tumorigenesis[18]. Only metastatic kidney renal clear cell, prostate and thyroid carcinomas showed substantial changes compared with primary tumours, encompassing 91% (43 of 47) of all significant discrepancies. Besides the poor prognostic marker 8q gain in metastatic prostate carcinoma[19], all discrepancies were associated with an increased prevalence of chromosomal arm losses at the metastatic setting. Remarkably,

30% (14 of 46) of the metastatic-enriched chromosome arm losses were retained when comparing non-WGD tumours (Extended Data Fig. 1c,d), indicating that other factors, aside from whole-genome doubling[20,21], have an important role in the accumulation of arm-sized chromosomal aberrations.

The same three cancer types also showed persistent increases in specific genomic instability indicators (that is, chromosomal aneuploidy score[20], loss of heterozygosity (LOH) genome fraction, WGD[22] and *TP53* alterations[23]) in the metastatic cohort (Fig. 1e and Supplementary Table 2). Although all of these indicators are elevated pan-cancer in WGD tumours (Extended Data Fig. 1c,d), non-WGD metastatic tumours of these three cancer types also had significantly greater aneuploidy and LOH scores. Furthermore, patients with metastatic tumours from other cancer types, including lung and colorectal adenocarcinomas, also displayed a moderate increase in aneuploidy and LOH scores, although they seemed to be primarily associated with higher metastatic WGD rates (Extended Data Fig. 1c). Our results thus revealed that the majority of cancer types have already acquired variable degrees of chromosomal arm aneuploidy in early stages of tumorigenesis. However, in certain cancers, such as kidney renal clear cell, prostate and thyroid carcinomas, significantly increased levels of genomic instability were induced in later evolutionary stages, which were, in turn, associated with substantial additional karyotypic changes.

## Tumour mutation burden

We observed that the small variant tumour mutation burden (TMB), collectively encompassing SBSs, DBSs and IDs, was only moderately increased in metastatic tumours compared with primary tumours across the 23 cancer types tested (fold-change increases of $1.25 \pm 0.47$ for SBSs, $1.55 \pm 0.86$ for DBSs and $1.45 \pm 0.53$ for IDs; mean ± standard deviation (s.d.)). In fact, 15 of the 23 cancer types had no significant increase in mutation burden for any mutation type. Only five cancer types (breast, cervical, thyroid and prostate carcinomas and pancreatic neuroendocrine tumour) had a consistent increase for the three mutation types at the metastatic stage, although the mutation profiles lacked systematic differences between primary and metastatic tumours (Fig. 2b and Extended Data Fig. 2a). Finally, further TMB comparisons grouping by tumour subtypes, metastatic biopsy locations and primary clinical progression status generally provided consistent results, although cancer-type-specific particularities are present (Supplementary Note 2). These results show that TMB is not necessarily indicative of tumour progression status and that the overall mutational spectra are tightly shaped by the mutational processes that were already active before and during primary tumour development.

## Mutational processes activity comparison

To determine whether the TMB differences may be attributed to differential activity of environmental or endogenous mutational processes, we assessed the activities of all operative mutational signatures in a quantitative and relative manner. We found that mutations attributed to cytotoxic treatments were significantly enriched in ten cancer types (Fig. 2c (red top bars) and Extended Data Figs. 2b,c and 3a for relative contributions). Platinum-based chemotherapies (SBS31/SBS35 and DBS5) showed the strongest mutagenic effect with $551 \pm 575$ (mean ± s.d.) SBS mutations and $32 \pm 22$ (mean ± s.d.) DBS-attributed mutations on average per sample. In fact, the excess in DBS mutation burden observed in eight cancer types (breast, oesophageal, stomach, cervical, ovarian serous and lung squamous carcinomas, cholangiocarcinoma and lung adenocarcinoma) was uniquely linked to platinum treatment mutations (Extended Data Fig. 2b, top bars). Likewise, median mutation contribution from the radiotherapy ID signature[24] (ID8) was enriched in six cancer types commonly exposed to radiation-based treatment (Extended Data Fig. 2c), whereas the 5-fluorouracil[25] (SBS17a/b) and

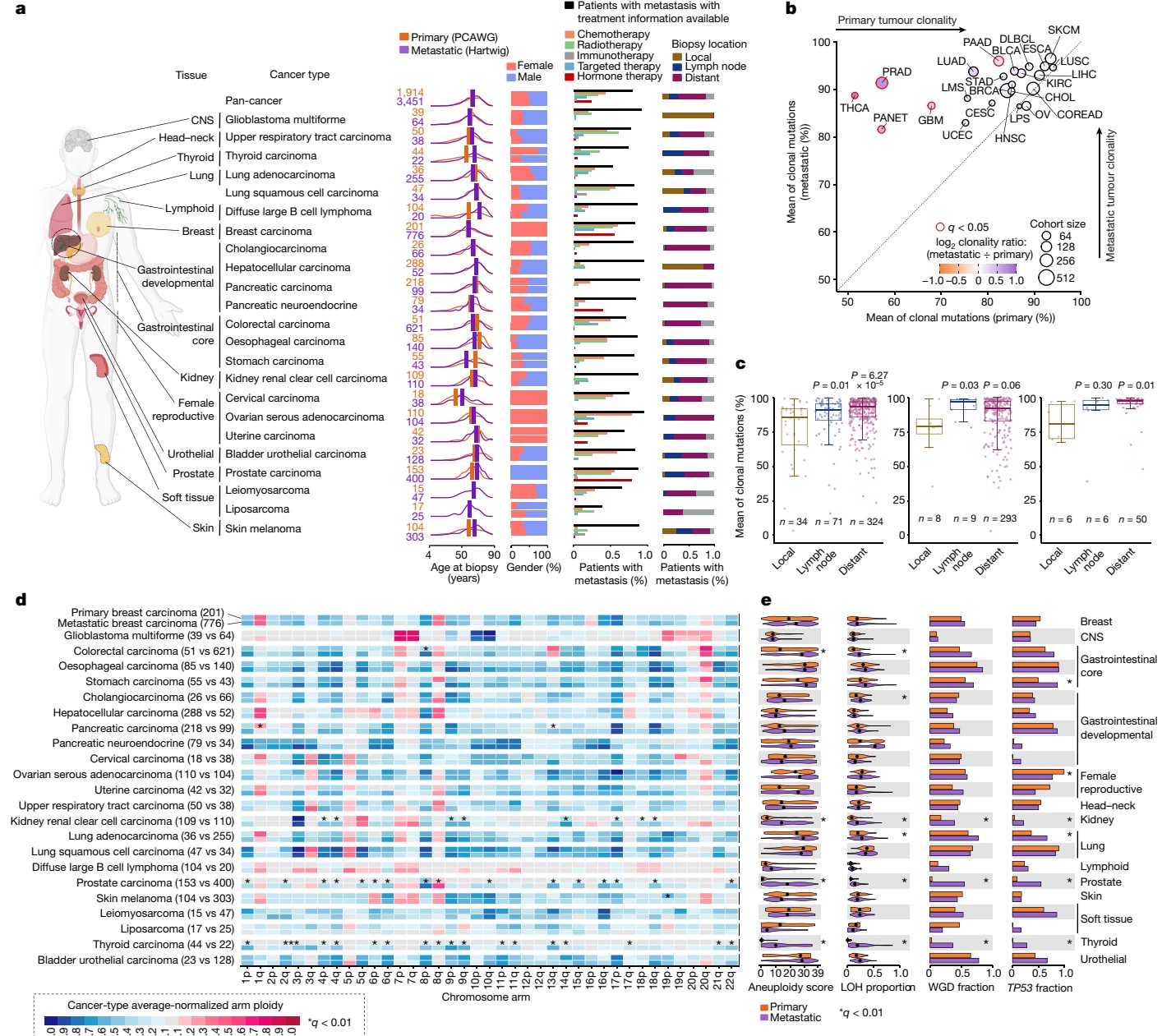

**Fig. 1 | Database overview and global genomic features. a**, Anatomical location of the 23 cancer types, ordered by tissue or origin, included in this study. From left to right: sample size, age at biopsy, gender, treatment type and biopsy site of the cohort with metastatic tumours. CNS, central nervous system. The image in **a** was created using BioRender (https://biorender.com). **b**, Mean percentage of clonal mutations in primary (x axis) and metastatic (y axis) tumours. The dots are coloured according to the log$_2$ of the clonality ratio (metastatic divided by primary). The size of the dots is proportional to the total number of samples (primary and metastatic). The red edge lines represent a two-sided Mann–Whitney adjusted $P < 0.05$. BLCA, bladder urothelial carcinoma; BRCA, breast carcinoma; CESC, cervical carcinoma; CHOL, cholangiocarcinoma; COREAD, colorectal carcinoma; DLBCL, diffuse large B cell lymphoma; ESCA, oesophageal carcinoma; GBM, glioblastoma multiforme; HNSC, upper respiratory tract carcinoma; KIRC, kidney renal clear cell carcinoma; LIHC, hepatocellular carcinoma; LMS, leiomyosarcoma; LPS, liposarcoma; LUAD, lung adenocarcinoma; LUSC, lung squamous cell carcinoma; PAAD, pancreatic carcinoma; PANET, pancreatic neuroendocrine tumour; PRAD, prostate carcinoma; OV, ovarian serous adenocarcinoma;

SKCM, skin melanoma; STAD, stomach carcinoma; THCA, thyroid carcinoma; UCEC, uterine carcinoma. **c**, Tumour clonality according to the metastatic biopsy location in breast (left), colorectal (middle) and oesophageal (right) carcinomas. n Refers to the number of samples. P refers to Mann–Whitney two-sided P value. For boxplots, the centre line indicates the median; the box limits denote the first and third quartiles; and the whiskers indicate the lowest or highest data points at the first quartile minus or plus 1.5× the interquartile range. **d**, Heatmap representing the normalized mean chromosome arm ploidy gains and losses relative to the expected 2n ploidy status in primary (top) and metastatic (bottom) tumours. *Adjusted $P < 0.01$ (two-sided Mann–Whitney). **e**, Comparison of four genomic instability indicators between primary (top) and metastatic (bottom) tumours. From left to right: aneuploidy score from ref. 20, the proportion of genome undergoing LOH, and the fraction of samples bearing whole-genome duplication (WGD) and TP53 alterations. The black dots represent the median values. *Adjusted $P < 0.01$ using two-sided Fisher's exact test for WGD and TP53, and two-sided Mann–Whitney test for the continuous features.

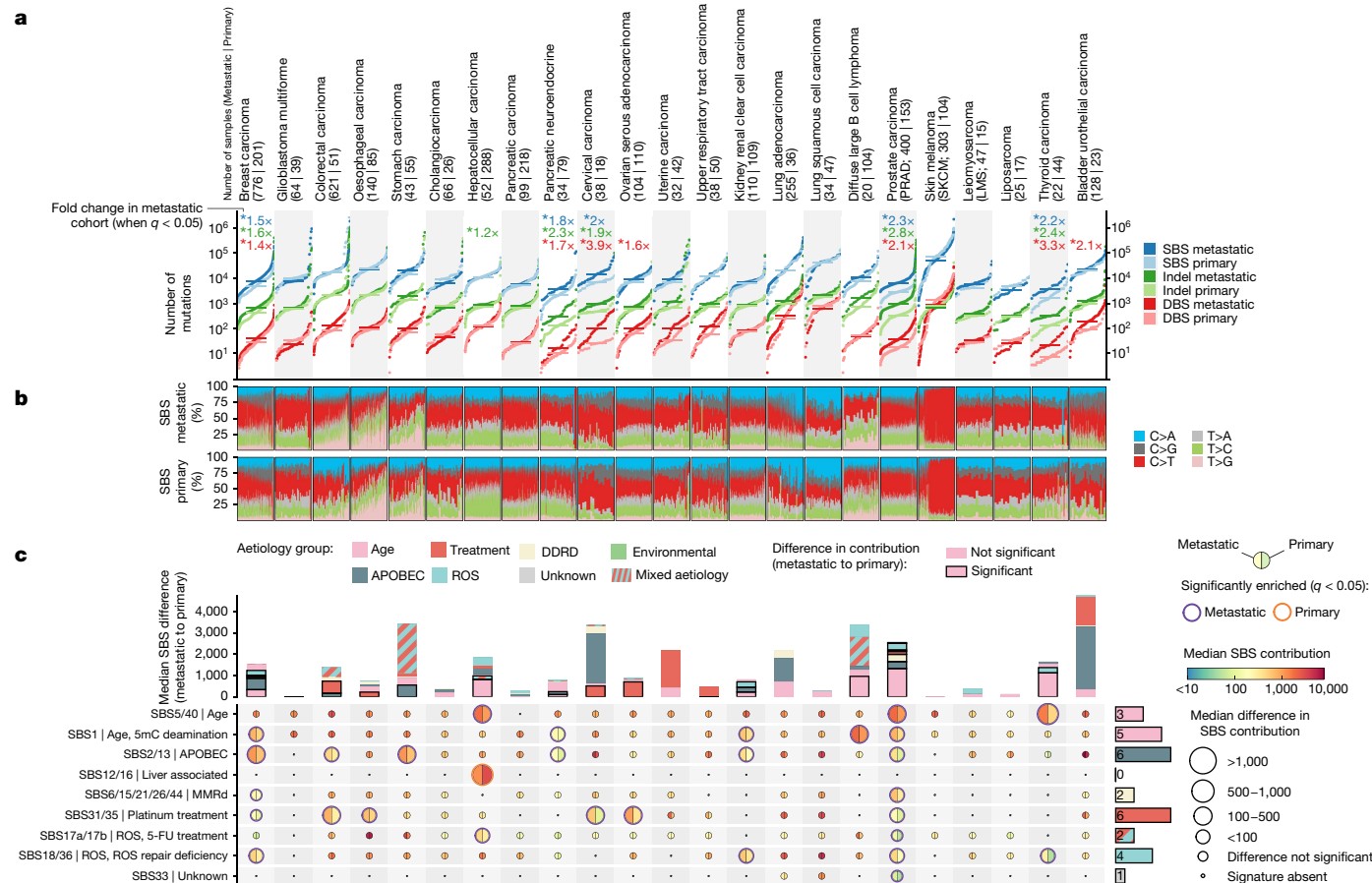

**Fig. 2 | TMB and mutational processes. a**, Cumulative distribution function plot (samples were ranked independently for each variant type) of TMB for each cancer type for SBS (blue), IDs (green) and DBS (red). The horizontal lines represent median values. The fold-change labels are included only when two-sided Mann–Whitney comparison renders a significant adjusted $P < 0.05$. **b**, SBS mutational spectra of patients with metastatic (top) and primary (bottom) tumours. Patients are ordered according to their TMB. DDRD, DNA damage repair deficiency; ROS, reactive oxygen species. **c**, Moon plot representing the SBS mutational burden differences attributed to each mutational signature in metastatic (main plot, left) and primary (main plot, right) tumours. The edge thickness and colours represent significant

differences (two-sided Mann–Whitney adjusted $P < 0.05$, $\pm1.4\times$ fold change) and the direction of the enrichment, respectively. The size of the circles are proportionate to the mutation burden difference. The bars on the right indicate the number of metastatic cancer types with a mutational signature with significant enrichment. The top stacked bars represent the cumulative signature exposure difference. The thicker bar edge lines represent significance. Bars are coloured according to the annotated aetiology. Only mutational signatures with known aetiology or with at least one cancer type with significant metastatic enrichment are included. 5-FU, 5-fluorouracil; 5mC, 5-methylcytosine; MMRd, mismatch repair deficiency.

polycyclic aromatic hydrocarbon metabolites from chemotreatments[26] (DBS2) also displayed greater metastatic mutation burden contribution in a tumour-type-specific manner (Extended Data Fig. 2b,c).

The broad enrichment of SBS2/SBS13 mutations in metastatic cancers suggests enhanced activity of APOBEC mutagenesis during the progression of advanced tumours. Specifically, our results revealed an increase in APOBEC mutation burden of $325 \pm 178$ (mean ± s.d.) mutations per sample in six metastatic tumours (breast, colorectal, stomach, kidney, prostate and pancreatic neuroendocrine carcinomas) that reached statistical significance, with breast and stomach cancers the types with the strongest increase (more than 500 APOBEC mutations per sample). Other cancer types, such as cervical and bladder urothelial carcinomas, also showed enhanced APOBEC activity (more than 2,500 median mutations per sample), but they did not reach significance due to already high intrinsic APOBEC activity in the primary tumours. The metastatic breast cancer samples also had a higher percentage in clustered APOBEC hypermutation variants than primary tumours (15% versus 5%; Extended Data Fig. 2d and Supplementary Table 3).

Six metastatic cancer types also displayed more mutations from the clock-like mutational processes, including five cancer types (diffuse

large B cell lymphoma, breast, prostate, pancreatic neuroendocrine and kidney renal clear cell carcinomas) that exhibited an increased SBS1 contribution and three cancer types (hepatocellular, prostate and thyroid carcinomas) that had an increased SBS5/SBS40 mutation burden. The increase in clock-like mutations in thyroid and prostate cancers, as well as diffuse B cell lymphomas to a lesser extent, may be explained by a larger proportion of older patients with metastatic disease. However, SBS1 metastatic enrichment was also present in cancer types with highly similar age population distributions (Fig. 1a).

Additional focused analyses will be needed to obtain a better understanding of the mutational signature differences that we observed in smaller subsets of cancer types (and subtypes) and metastatic locations (full data in Supplementary Table 3 and Supplementary Data 1).

## Differential SBS1 mutation burden

To investigate the SBS1 mutation burden differences in more detail, we evaluated their SBS1 mutation burden by the age of biopsy separately for both cohorts. As expected, the SBS1 mutation burden per year was highly tissue specific[27,28] and displayed an increase with age

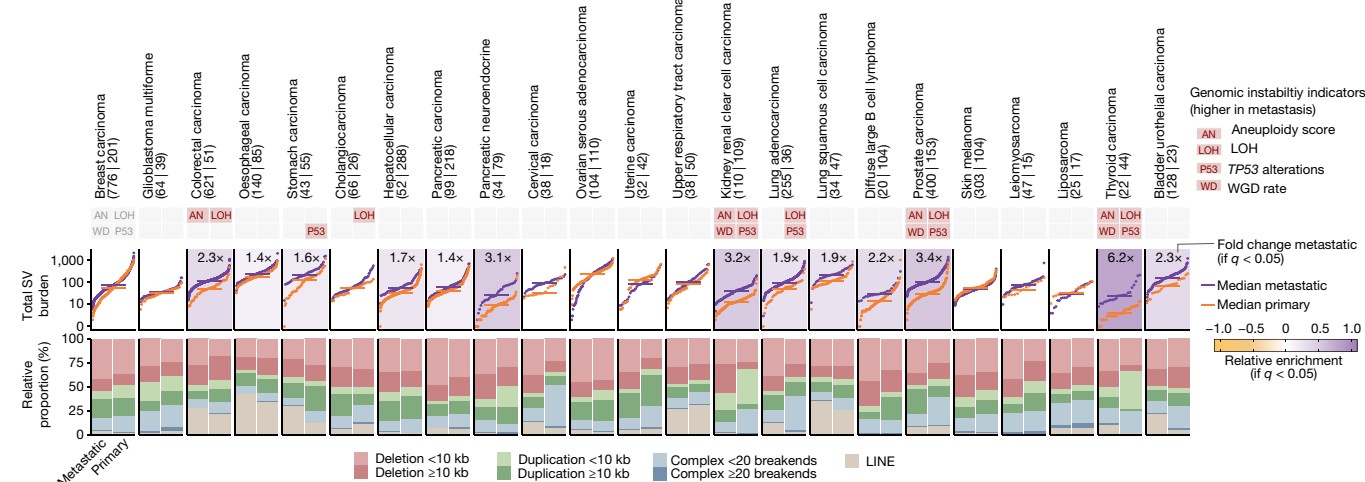

**Fig. 3 | SV burden.** The top rectangles represent the four genomic instability features defined in Fig. 1e. The red background represents significant enrichment in the metastatic cohort (two-sided Mann–Whitney adjusted $P < 0.01$). S-plots and cumulative distribution function plots (samples ranked independently for each SV type) of the aggregated SV burden for each cancer type. The horizontal lines represent median values. Backgrounds are coloured according to the relative enrichment, defined as: $\log_{10}$(median SV-type burden in metastatic tumours + 1) – $\log_{10}$(median SV-type burden in primary tumours + 1). Fold-change labels and coloured backgrounds are displayed when Mann–Whitney comparison renders a significant $q < 0.05$. Fold-change labels are displayed with '>' when the SV burden for primary tumours is 0 (see Methods for more details). For each cancer type, the bottom bar plots represent the relative fraction of each SV type in the metastatic (left) and primary (right) datasets. LINE, long interspersed nuclear element.

in the majority of cancer types in both primary and metastatic cohorts (Pearson's $R > 0.1$, 15 of 23 tumour types; Extended Data Fig. 4a and Supplementary Table 4). However, four cancer types (that is, breast, prostate, kidney renal clear cell and thyroid carcinomas) showed an age-independent and significant enrichment of SBS1 mutations in metastatic lesions (Extended Data Fig. 4a). For instance, metastatic breast cancer had a nearly uniform fold-change increase of 1.46 over primary tumours (188 ± 16 SBS1 mutations, mean ± s.d.) across the ages of biopsies, and which was generally consistent across breast cancer subtypes (Extended Data Fig. 4b and Supplementary Note 3). This pattern was highly cancer-type specific and was not observed for most cancer types, including those with similar intratumour heterogeneity in the primary cohort (for example, colorectal, ovarian serous and pancreatic carcinomas) (Extended Data Fig. 4a). Moreover, this pattern was not explained by differences in tumour genome ploidy (Extended Data Fig. 4c) or by metastatic biopsy location (Supplementary Note 3), was observed in paired primary–metastatic biopsies from individual patients with breast and kidney renal clear cell carcinomas and rendered consistent patterns when relying on independent unpaired cohorts (Supplementary Note 3). Finally, other mutational processes that operate over the evolution of the somatic tissues (for example, clock-like mutations attributed to SBS5/SBS40 that accumulate with age in a cell-cycle-independent manner[28,29]) did not show such enrichment (Extended Data Fig. 4d).

SBS1 mutation burden has been extensively correlated with estimated stem cell division rates[30]. Therefore, an increase in age and tumour-type-specific SBS1 mutation burden in treated metastatic tumours may indicate that these tumours have undergone a higher number of cell divisions. However, the estimated number of years to explain the SBS1 mutation burden shift (23 and 71 years for breast and prostate cancers, respectively; see Supplementary Table 4) shows that this is unlikely to be the main cause. Hence, a more plausible explanation, which also supports previous observations[31,32], is that these metastatic tumours display accelerated cell division rates compared with their primary tumour counterparts (Extended Data Fig. 4e). Supporting this hypothesis, metastatic tumours also had a lower normalized fraction of clonal SBS1 mutations (Extended Data Fig. 4f). Of note, this pattern was not observed in cancer types with consistently high SBS1

mutagenic dynamics (Extended Data Fig. 4f) and was indistinguishable for SBS5/SBS40 mutations (Supplementary Note 3).

Finally, we observed a negative association between the yearly rate of SBS1 mutation accumulation in primary tumours (a proxy of stem cell division rates[30]) and the estimated fold change of the SBS1 mutation rate in the metastatic cohort (Extended Data Fig. 4g,h). This suggests that tumours with an intrinsically active turnover rate (for example, colorectal carcinomas) preserve their high proliferation rates, whereas others with lower cell division rates (for example, breast, prostate, kidney and thyroid carcinomas) may acquire higher proliferation rates during the course of cancer progression. Nevertheless, we cannot rule out the contribution of other tissue-type-specific or tumour-type-specific mechanisms, such as higher rates of 5-methylcytosine deamination, decreased fidelity to repair these mismatches or higher contribution from other metastatic-specific mutational processes with overlapping mutational contexts.

## SV burden

Comparison of the total number of SVs per tumour revealed an extensive increase in the metastatic tumours (fold change of 2.5 ± 1.3, mean ± s.d.). This increase was observed in 13 of 23 (56%) cancer types (Fig. 3a, Extended Data Fig. 5a and Supplementary Table 5), and was not generally explained by differences in sequencing coverage, tumour clonality (Supplementary Note 1) or cancer-subtype composition (Supplementary Note 2). Moreover, the increased SV burden was also observed for cancer types lacking substantial changes in genomic instability indicators, such as oesophageal and lung squamous cell carcinomas (Fig. 3a). Finally, we observed an increased SV burden in prostate and pancreatic neuroendocrine primary tumours that eventually progressed compared with those with relatively better prognosis, which in both cases were in turn lower than the median values in metastatic tumours (Supplementary Note 2). Overall, compared with TMB, the SV analyses revealed a much more widespread pan-cancer effect, with larger increases per metastatic cancer type that affected almost every cancer type studied.

Small (less than 10 kb) deletions were the most enriched SV types in metastatic tumours (2.7 ± 1.2 fold change in 15 of 23 cancer types with

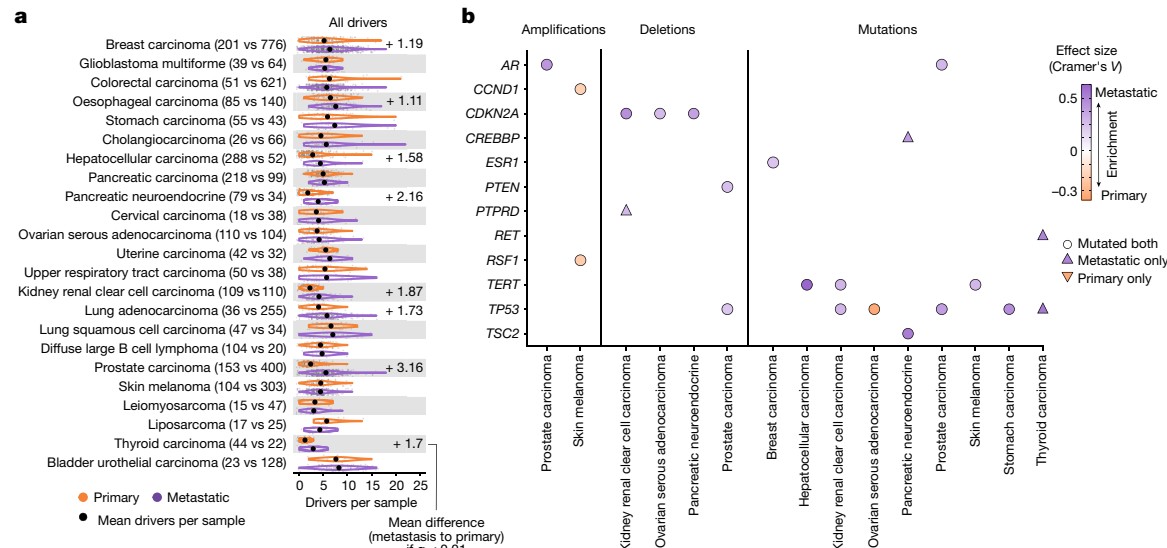

**Fig. 4 | Driver alterations in primary and metastatic tumours. a**, Cancer-type-specific distribution of the number of driver alterations per patient in primary (top) and metastatic (bottom) tumours. The black dots represent the mean values. Labels display mean differences (metastatic to primary) in cancer types with a significant difference (two-sided Mann–Whitney adjusted $P < 0.01$). **b**, Heatmap representing the cancer genes displaying significant mutational frequency differences between primary and metastatic tumours (two-sided Fisher's exact test adjusted $P < 0.01$). Circles denote mutation frequency enrichment in both cohorts, whereas triangles facing upwards and downwards represent drivers that are exclusively enriched in metastatic and primary cohorts, respectively. Colours represent the direction of the enrichment.

significant enrichment, mean ± s.d.; Extended Data Fig. 5a,b). Larger (10 kb or larger) deletions and duplications had a similar pan-cancer enrichment, although generally with slightly lower fold changes. Complex SVs with 20 or more breakpoints, encompassing events such as chromothripsis and chromoplexy, were enriched in metastatic prostate carcinomas (more than threefold enrichment). Finally, a strong cancer-type-specific metastatic enrichment was also noted for long interspersed nuclear element insertions (LINE), with an increased fold change of 12.2 and 12.5 in stomach and bladder urothelial carcinomas, respectively.

We next used linear regression models to unravel the underlying features associated with the observed increase in SV burden in metastatic tumours (Extended Data Fig. 6a–k and Supplementary Table 5). Our approach confirmed the role of previously described cancer-type-specific driver-induced SV phenotypes, including homologous recombination deficiency[33] in metastatic breast carcinoma tumours, *CDK12* (ref. 34) alterations in prostate carcinoma and *MDM2* (ref. 35) amplifications in breast ER⁺/HER2⁻ carcinomas, among others. Genomic instability features, such as genome ploidy and *TP53* alterations, showed a strong pan-cancer association with deletions and duplications (Extended Data Fig. 6a,d), and thus very likely contributed to the observed SV increase in metastatic tumours[20,22,23]. Finally, previous exposure to radiotherapy treatment was strongly associated with small deletions in breast ER⁺/HER2⁻ and prostate carcinomas[36].

## Cancer driver gene landscape

Metastatic tumours showed a moderate increase in the total number of driver gene alterations per patient (a mean of 4.5 and 5.3 driver alterations per sample in primary and metastatic tumours, respectively), including 8 (34%) tumour types with a significant increase (Fig. 4a). Prostate carcinoma (average increase of 3.16 driver alterations per sample), pancreatic neuroendocrine tumour (2.16), thyroid carcinoma (1.7) and kidney renal clear cell carcinoma (1.87) showed the strongest increases (more than 1.5 driver alterations per patient), whereas the majority of cancer types showed a mean increase below 1.5 driver

alterations per sample. All mutation types (amplifications, deletions and mutations) contributed to the increased driver alterations in metastatic tumours (Extended Data Fig. 7a).

Comparison of gene and cancer-type frequencies revealed that only 12 genes had a significant frequency bias in at least one cancer type (22 gene and cancer-type pairs in total; Fig. 4b, Extended Data Fig. 7b and Supplementary Table 6). The majority (19 out of 22, 86%) of the significant pairs had enrichment towards higher metastatic frequency, including four driver genes that were exclusively mutated in metastatic tumours and were not found in the primary tumour equivalents (*PTPRD* in kidney renal clear cell carcinoma, *CREBBP* in pancreatic neuroendocrine tumour, and *RET* and *TP53* alterations in thyroid carcinoma). Most metastatic-enriched cancer drivers had a cancer-type-specific enrichment, including well-established resistance gene drivers associated with anticancer therapies, such as *AR* and *ESR1* alterations in patients with prostate and ER⁺ breast carcinomas treated with hormone deprivation therapies[37,38] (Fig. 4b and Supplementary Note 2). Nevertheless, three driver genes (that is, *TP53*, *CDKN2A* and *TERT*) showed a metastatic enrichment across multiple cancer types (Fig. 4b), indicating that alterations of these genes may enhance aggressiveness by disturbing pan-cancer hallmarks of tumorigenesis.

We next investigated whether the reported driver differences may have an effect on potential clinical actionability. Cancer-type-specific comparison of therapeutically actionable variants revealed an overall larger fraction of patients with therapeutically actionable variants in the metastatic cohort, with high variability across cancer types (Extended Data Fig. 8a and Supplementary Table 7). Subsetting by A-on label variants (that is, approved biomarkers in the specific cancer type) revealed a consistent pattern in which only cholangiocarcinoma (*FGFR2* fusions and *IDH1* mutations) and lung adenocarcinoma (*EGFR* alterations) showed a substantial proportional increase in the metastatic cohort (Extended Data Fig. 8a,b). Non-A-on label biomarkers (A-off label, B-on label and B-off label) showed a modest and tumour-type-dependent metastatic increase, which was mainly linked to the increased alteration frequency of *KRAS* exon 2 mutations and *CDKN2A* loss in advanced tumour stages (Extended Data Fig. 8b).

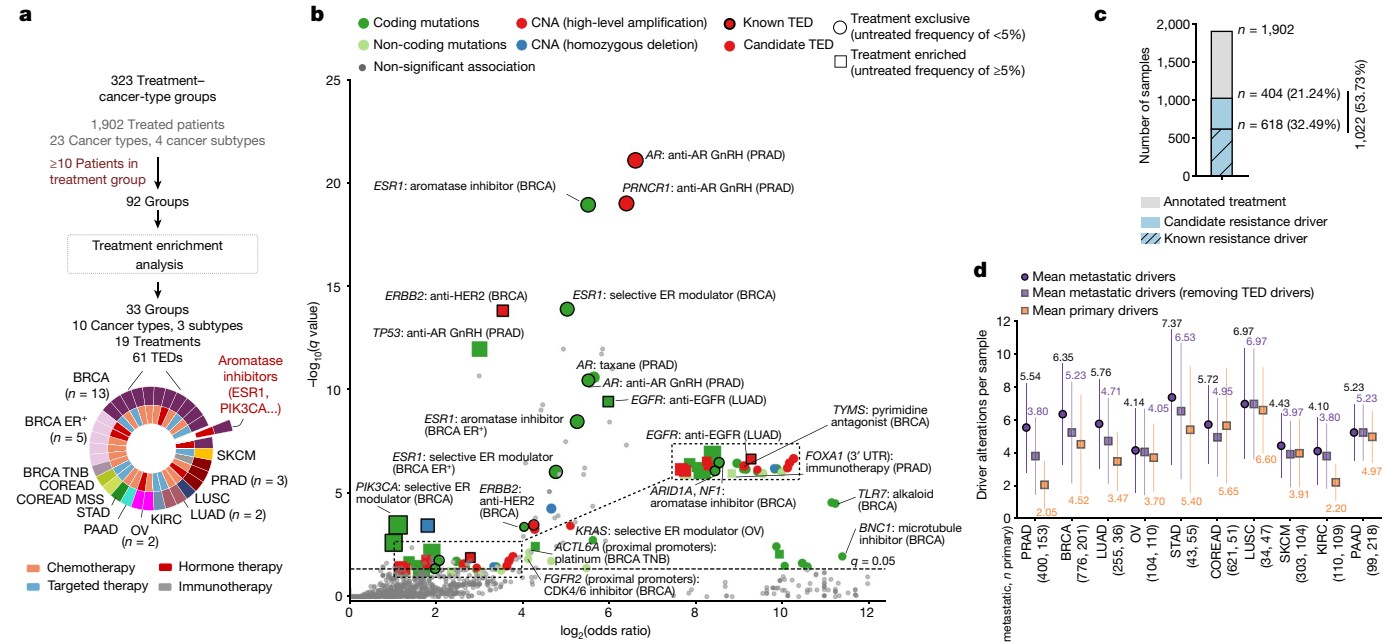

**Fig. 5 | TEDs. a**, Workflow representing the number of treatment groups in each step of the analysis. For the external layers of the pie chart, the number of treatments with identified TEDs is coloured by cancer type. For the internal layers of the pie chart, the category of the corresponding treatment is shown. *n* Refers to the number of treatment groups with at least one TED. MSS, microsatellite stable; TNB, triple negative breast cancer. **b**, Volcano plots displaying the identified TEDs. Each dot represents one cancer gene alteration type in one treatment group. The *x* axis displays the effect size (as $\log_2$(odds ratio)) and the *y* axis shows the significance ($-\log_{10}$(*q* value)). The circle markers denote TEDs exclusively mutated in the treatment group (squared makers are used otherwise). Markers are coloured according to the type of alteration. The thicker edge lines indicate known resistance drivers. CNA, copy number alteration; UTR, untranslated region. **c**, Global proportion of patients with TEDs treated for metastasis. **d**, Mean number of driver alterations per patient with a metastatic tumour before (purple circle) and after (purple square) excluding TEDs compared with patients with primary tumours (orange square). The vertical lines indicate s.d. The mean number of driver alterations are labelled. *n* Metastatic and *n* primary denote the number of metastatic and primary samples, respectively.

## Treatment-associated drivers

The presence of treatment resistance driver genes in late-stage tumours prompted us to devise a test that aimed to identify treatment-enriched drivers (TEDs) that were either significantly enriched (that is, treatment enriched) or exclusively found (that is, treatment exclusive) in a cancer-type-specific and treatment-specific manner (Extended Data Fig. 9a and Methods). Our analytical framework provided 61 TEDs associated with 33 treatment groups from 8 cancer types and 4 cancer subtypes (Fig. 5a,b, Supplementary Table 8 and Supplementary Note 4). Of the identified TEDs, 33 of 61 (54%) were coding mutation drivers, 16 (26%) were copy number amplifications, 9 (14%) were non-coding drivers and 3 (6%) were recurrent homozygous deletions (Fig. 5b and Extended Data Fig. 9b,c). Reassuringly, the majority of the top hits were known treatment resistance drivers, including *AR*-activating mutations and gene amplifications in patients with prostate cancer treated with androgen-deprivation therapy[38] (Extended Data Fig. 9d,e), *ESR1*[536–538] mutations in patients with breast cancer treated with aromatase inhibitors[37] (Extended Data Fig. 9f), and *EGFR*[T790M] mutations (Extended Data Fig. 9g) and *EGFR* copy number gains in patients with lung adenocarcinoma treated with *EGFR* inhibitors[39,40] (Extended Data Fig. 9h), among others. Moreover, we also found that *TP53* alterations were recurrently associated with resistance to multiple treatments, which may indicate that these alterations are prognostic markers for enhanced tumour aggressiveness and plasticity rather than being a cancer-type-specific mechanism of drug resistance (Supplementary Note 4).

Our results also provided a long tail of candidate drivers of resistance, some of them with orthogonal evidence by independent reports (Fig. 5b and Extended Data Fig. 9b,c). Examples of the latter group include *TYMS* amplification in patients with breast cancer treated with pyrimidine

antagonists[41] (Extended Data Fig. 9i), *PRNC1* and *MYC* co-amplifications in patients with prostate cancer treated with androgen deprivation[42] (Extended Data Fig. 9j), *ACTL6A* promoter mutations in patients with triple-negative breast cancer treated with platinum-based therapies[43], and *FGFR2* promoter mutations in patients with breast cancer treated with CDK4/CDK6 inhibitors[44]. The full TEDs catalogue is provided in Supplementary Table 8 and constitutes a valuable resource for investigating resistance mechanisms to common cancer therapies.

Overall, 53% of patients with metastatic disease with annotated treatment information had TEDs, including 32% with annotations of known resistance drivers and an additional 21% of patients with candidate resistance drivers derived from our analysis (Fig. 5c). We identified $0.70 \pm 0.53$ (mean ± s.d.) TEDs per metastatic sample across the 8 cancer types that had reported TEDs (Fig. 5d), with prostate and breast carcinomas displaying the greatest prevalence of TEDs (that is, 1.74 and 1.12 drivers per patient with prostate and breast cancers, respectively). Therefore, after excluding TEDs, primary and metastatic tumours had a 36% reduction of their original differences in the number of drivers per sample (from 5.3 to 5.0 mean drivers per sample in the metastatic cohort after excluding TEDs, compared with 4.5 mean drivers per sample in the primary cohort) (Fig. 5d and Supplementary Table 8), indicating that an important proportion of the metastatic-enriched drivers are probably associated with resistance to anticancer therapies.

## Discussion

In this study, we describe a cohort of more than 7,000 uniformly reprocessed WGS samples from patients with primary untreated and metastatic treated tumours. We compared genomic features across 23 cancer types and confirmed previous cancer-type-specific observations while

also providing novel biological findings, such as the clock-like molecular features, the prevalence of SV burden across different tumorigenic stages and the incidence of TEDs in treated patients.

Specifically, metastatic tumours displayed high genomic instability, low intratumour heterogeneity and strong accumulation of SVs. However, the magnitude of genomic differences between primary and metastatic tumours was highly cancer-type specific and was influenced by the exposure to cancer treatments. Overall, five cancer types (prostate, thyroid, kidney renal clear cell, breast and pancreatic neuroendocrine carcinomas) showed an intense transformation of the genomic landscape in advanced tumorigenic stages (Extended Data Fig. 10, labelled as strong). The other cancer types displayed variable genomic differences, although the chromosomal genomic portrait tended to be conserved.

Cancer types with the strongest genomic differences between primary and metastatic settings in our analyses typically have a good prognosis in the primary setting. But then, whether the metastatic tumours representing a unique set of primary patients that eventually progressed (that is primaries from the metastatic cohort were 'born to be bad') or whether there are stochastic triggers of metastatic disease in relatively indolent primaries are still to be determined. To fully address these, larger pan-cancer sets of matched biopsies from the same patient, as already implemented in various cancer-type-specific studies[45,46], would be needed.

This study faced various limitations, such as the use of different laboratory workups and sequencing parameters used for primary and metastatic tumour samples, although we demonstrated that this does not severely have an effect on the overall detectability of clonal somatic variants (Supplementary Note 1). However, we cannot exclude the possibility of missing highly subclonal driver mutations. Furthermore, our observations are unlikely to be exhaustive, especially for lower frequency events, because of limited cohort (or subcohort) sizes. Expanding cancer cohorts in research or clinical settings will be essential to advance our understanding of tumour progression. Finally, genomic changes alone cannot entirely explain how tumour cells are able to colonize other organs. Therefore, additional information from complementary tumour omics[47] and from the tumour microenvironment[48] will be needed to further dissect and better understand metastasis and resistance to cancer therapies.

To conclude, our dataset constitutes a valuable resource that can be leveraged to further study other aspects of tumour evolution, such as genomic differences across metastatic biopsy locations (Supplementary Note 2), dedicated analysis for cancer subtypes (Supplementary Note 2), genetic immune escape alterations in primary and metastatic tumours[49] as well as for the development of machine learning tools to foster cancer diagnostics[50].

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

## Methods

### Cohort gathering and processing

We have matched tumour–normal WGS data from patients with cancer from two independent cohorts: Hartwig and PCAWG. A detailed description of the Hartwig and PCAWG cohort gathering and processing as well as comprehensive documentation of the PCAWG sample reanalysis with the Hartwig somatic pipeline is described in the Supplementary Note 1.

### Tumour clonality analysis

Each mutation in the .vcf files was given a subclonal likelihood by PURPLE. Following PURPLE guidelines, we considered mutations with subclonal scores equal or higher than 0.8 to be subclonal and mutations below the 0.8 threshold to be clonal. For each sample, we then computed the average proportion of clonal mutations by dividing the number of clonal mutations by the total mutation burden (including SBS, multinucleotide variants and IDs). Finally, for each cancer type, we used Mann–Whitney test to assess the significance of the clonality difference between the primary and metastatic tumours. $P$ values were adjusted for false discovery rate (FDR) using the Benjamini–Hochberg procedure. An adjusted $P < 0.05$ was deemed as significant.

In addition, we leveraged biopsy site data in patient reports to further investigate differences in metastatic tumour clonality according to the metastatic biopsy site (see also Supplementary Note 2). If the metastatic biopsy site was in the same organ or tissue as the primary tumour, we considered them as 'local', whereas if the metastatic biopsy site was reported in the lymphoid system or other organs or tissues, they were dubbed as 'lymph' and 'distant', respectively. Cancer types for which there was a minimum of five samples available for each of the biopsy groups were selected and Mann–Whitney test was used to compare the clonality between the biopsy groups.

### Karyotype

Chromosome arm level and genome ploidy was estimated as previously described[20].

First, for each chromosome arm, tumour purity and ploidy-adjusted copy number (CN) segments (as determined by PURPLE) were rounded to the nearest integer. Second, arm coverage of each integer CN was calculated as the fraction of chromosome arm bases with the specific CN divided by the chromosome arm length (for example, 60% of all chromosome 5p segments have a CN of 2, 30% have a CN of 1 and 10% have a CN of 3). We defined the arm-level ploidy level as the CN with the highest coverage across the whole arm (in the example above it would be 2). Third, we computed the most recurrent chromosome arm ploidy levels across all chromosome arms per sample (that is, observed genome ploidy).

Next, we estimated the true genome ploidy by taking WGD status (given by PURPLE) into account. If a sample did not undergo WGD, its total expected genome ploidy was deemed to be $2n$. If a sample did undergo WGD and its observed genome ploidy was less than six, the estimated genome ploidy was deemed to be $4n$, and $8n$ if the observed genome ploidy was six or more. An observed genome CN of more than eight was not found in our dataset.

Then, for each chromosome arm in each sample, we defined the normalized arm ploidy as the difference between the arm-level ploidy level and the expected genome ploidy. The resulting value was classified as 1 for differences higher than or equal to 1 (representing arm gains), as −1 for differences lower than or equal to −1 (representing arm losses) or as 0 (no difference). Normalized arm ploidy values were averaged across all samples from a cancer type in a cohort-specific manner (that is, separating primary and metastatic samples). A Mann–Whitney test was performed per cancer type and chromosome arm to assess the mean difference in arm gains or losses at the cancer-type level. The resulting $P$ value was FDR adjusted across all arms per cancer type. Finally, $q < 0.01$ and a normalized arm ploidy difference higher than 0.25 was deemed to be significant.

### Genomic instability indicators

To compare the differences in aneuploidy scores and the LOH proportions in each group, a Mann–Whitney test was performed per cancer type. The aneuploidy score represents the number of arms per tumour sample that deviate from the estimated genome ploidy as previously described[20]. The LOH score of a given sample represents the sum of all LOH regions divided by the GRCh37 total genome length. A genomic region is defined as LOH when the minor allele CN < 0.25 and major allele CN ≥ 0.8.

To compare the fraction of samples with a driver mutation in *TP53* as well as the fraction of WGD samples per cohort, a Fisher's exact test was performed per cancer type. Any *TP53* driver alteration (non-synonymous mutation, biallelic deletion and homozygous disruption) was considered in the analysis. Multiple driver mutations per sample in a single gene were considered as one driver event. WGD was defined as present if the sample had more than 10 autosomes with an estimated chromosome CN of more than 1.5. $P$ values were FDR corrected across all cancer types. A $q < 0.01$ was deemed to be significant for all statistical tests.

### Mutational signature analysis

**Signature extraction.** The number of somatic mutations falling into the 96 SBS, 78 DBS and 83 ID contexts (as described in the COSMIC catalogue[51]; https://cancer.sanger.ac.uk/signatures/) was determined using the R package mutSigExtractor (https://github.com/UMCUGenetics/mutSigExtractor, v1.23).

SigProfilerExtractor (v1.1.1) was then used (with default settings) to extract a maximum of 21 SBS, 8 DBS and 10 ID de novo mutational signatures. This was performed separately for each of the 20 tissue types that had at least 30 patients in the entire dataset (aggregating primary and metastatic samples; see Supplementary Table 3). Tissue types with less than 30 patients as well as patients with metastatic tumours with unknown primary location type were combined into an additional 'Other' group, resulting in a total of 21 tissue-type groups for signature extraction. To select the optimum rank (that is, the eventual number of signatures) for each tissue type and mutation type, we manually inspected the average stability and mean sample cosine similarity plot output by SigProfilerExtractor. This resulted in 440 de novo signature profiles extracted across the 21 tissue-type groups (Supplementary Table 3). Least squares fitting was then performed (using the fitToSignatures() function from mutSigExtractor) to determine the per-sample contributions to each tissue-type-specific de novo signature.

**Aetiology assignment.** The extracted de novo mutational signatures with high cosine similarity (≥0.85) to any reference COSMIC mutational signatures with known cancer-type associations[51] were labelled accordingly (288 de novo signatures matched to 57 COSMIC reference signatures).

For the remaining 152 unlabelled de novo signatures, we reasoned that there could be one or more signatures from one cancer type that is highly similar to those found in other tissue types, and that these probably represent the same underlying mutational process. We therefore performed clustering to group likely equivalent signatures. Specifically, the following steps were performed:

(1) We calculated the pairwise cosine distance between each of the de novo signature profiles.
(2) We performed hierarchical clustering and used the base R function cutree() to group signature profiles over the range of all possible cluster sizes (minimum number of clusters = 2; maximum number of clusters = number of signature profiles for the respective mutation type).
(3) We calculated the silhouette score at each cluster size to determine the optimum number of clusters.
(4) We grouped the signature profiles according to the optimum number of clusters. This yielded 27 SBS, 7 DBS and 8 ID de novo signature clusters (see Supplementary Table 3).

For certain de novo signature clusters, we could manually assign the potential aetiology based on their resemblance to signatures with known aetiology described in COSMIC[51], Kucab et al.[26] and Signal (access date 1 February 2023)[52]. Some clusters were an aggregate of two known signatures, such as SBS_denovo_clust_2, which was a combination of SBS2 and SBS13, both linked to APOBEC mutagenesis. Other clusters had characteristic peaks of known signatures, such as DBS_denovo_clust_4, which resembled DBS5 based on having distinctive CT>AA and CT>AC peaks. Finally, DBS_denovo_clust_1 was annotated as a suspected POLE mutation and MMRd, as samples with high contribution (more than 150 mutations) of this cluster are frequently microsatellite instable (MSI) or have POLE mutations. Likewise, DBS_denovo_clust_2 was annotated as a suspected MMRd as the aetiology, as samples with high contribution (more than 250 mutations) of this cluster were all MSI. See Supplementary Table 3 for a list of all the manually assigned aetiologies.

After applying the aetiology assignment, the de novo extraction resulted in 69 SBS, 13 DBS and 18 ID representative mutational signatures (Supplementary Table 3). Most of these (42 of 69 SBSs, 7 of 13 DBSs and 8 of 18 IDs) mapped onto the well-described mutational signatures in human cancer[35,53].

Comparing the prevalence of mutational processes between primary and metastatic cancer. We then compared the activity (that is, the number of mutations contributing to) of each mutational process between primary and metastatic tumours. For each sample, we first summed the contributions of signatures of the same mutation type (that is, SBS, DBS or ID) with the same aetiology, hereafter referred to as 'aetiology contribution'. Per cancer type and per aetiology, we performed two-sided Mann–Whitney tests to determine whether there was a significant difference in aetiology contribution of primary and metastatic tumours. Per cancer type and per mutation type, we used the p.adjust() base R function to perform multiple testing correction using Holm's method. Next, we added a pseudocount of 1 to the contributions (to avoid dividing by 0) and calculated the median contribution log2 fold change, that is, log2((median contribution in metastatic tumours + 1)/(median contribution in primary tumours + 1)). We considered the aetiology contribution between primary and metastatic tumours to be significantly different when q < 0.05, and log2 fold change ≥ 0.4 or log2 fold change ≤ −0.4 (= ± ×1.4).

### Relative contribution

Relative aetiology contribution was calculated by dividing aetiology contribution by the total contribution of the respective mutation type (that is, SBS, DBS or ID). To determine the significant difference in relative aetiology contribution, we performed two-sided Mann–Whitney tests as described above. We also calculated the median difference in contribution (that is, median relative contribution in metastatic tumours − median relative contribution in primary tumours). We considered the relative aetiology contribution between primary and metastatic tumours to be significantly different when q < 0.05 and median difference was 0.01 or more.

We also determined whether there was an increase in the number of samples with high aetiology contribution (that is, hypermutators) in metastatic versus primary cohorts. For each signature, a sample was considered a hypermutator if the aetiology contribution was 10,000 or more for SBS signatures, 500 or more for DBS signatures or 1,000 or more for ID signatures. For each cancer type, for each aetiology, we performed pairwise testing only for cases in which there were five or more hypermutator samples for either metastatic or primary tumours. Each pairwise test involved calculating P values using two-sided Fisher's exact tests, and effect sizes by multiplying Cramer's V by the sign of the $\log_2$(odds ratio) to calculate a signed Cramer's V value that ranges from −1 to +1 (indicating enrichment in primary or metastatic, respectively). We then used the p.adjust() base R function to perform multiple testing correction using Bonferroni's method.

### SBS1–age correlations in primary and metastatic tumours

To count the SBS1 mutations, we relied on the definition from ref. 54 that is based on the characteristic peaks of the COSMIC SBS1 signature profile: single-base CpG > TpG mutations in NpCpG context. To ensure that these counts and the downstream analyses are not affected by differential APOBEC exposure in primary and metastatic cohorts, we excluded CpG > TpG in TpCpG, which is also a characteristic peak in the COSMIC SBS2 signature profile. In addition, for skin melanoma, CpG > TpG in [C/T]pCpG, which overlaps with SBS7a, was excluded. To obtain the SBS5 and SBS40 counts, we relied on their exposures derived from the mutational signature analyses performed in this study (described above).

To assess the correlation between SBS1 burden and the age of the patient, at biopsy we performed a cancer-type and cohort-specific linear regression (that is, separate regression for primary and metastatic tumour samples). To avoid spurious effects caused by hypermutated tumours, samples with a TMB greater than 30,000 as well as those with SBS1 burden greater than 5,000 were excluded.

For each cancer type and cohort, we then computed 100 independent linear regressions by randomly selecting 75% of the available samples. We selected the median linear regression (based on the regression slope) as representative regression for further analyses. Similarly, confidence intervals were derived from the 1st and 99th percentile of the computed regressions.

To evaluate the significance of the differences between primary and metastatic representative linear regressions (hereafter referred to as linear regression for simplicity), we first filtered out cancer types that failed to show a positive correlation trend between SBS1 burden and age at biopsy in both primary and metastatic tumours (that is, Pearson's correlation coefficient of primary and metastatic regression greater than 0.1). Next, for each selected cancer type, we computed the regression residuals of primary and metastatic SBS1 mutation counts using, in both cases, the primary linear regression as baseline. The primary and metastatic residual distributions were then compared using a Mann–Whitney test to evaluate significance. Cancer types with a Mann–Whitney P < 0.01 were deemed as significant. Finally, to ensure that the differences were uniform across different age ranges (that is, not driven by a small subset of patients), we only considered significant cancer types in which the metastatic linear regression intercept was higher than the primary intercept.

SBS5/SBS40 correlations were computed following the same procedure and using the sum of SBS5 and SBS40 exposures for each tumour sample. If none of the mutations were attributed to SBS5/SBS40 mutational signatures, the aggregated value was set to zero. In the ploidy-corrected analyses, we divided the SBS1 mutation counts (and SBS5/SBS40 mutation counts for the SBS5/SBS40 ploidy-corrected regression, respectively) by the PURPLE-estimated tumour genome ploidy.

For each cancer type, the mean fold change (fc) was defined as $\underline{fc} = \frac{1}{40} \sum_{i=40}^{80} \frac{MPred_i}{PPred_i}$ where $MPred_i$ and $PPred_i$ are the estimated number of SBS1 mutations for a given age $i$th according to the metastatic and primary linear regressions, respectively. Similarly, the mean estimated SBS1 burden difference ($SBS1_{diff}$) was defined as: $\underline{SBS1diff} = \frac{1}{40} \sum_{i=40}^{80} MPred_i - PPred_i$.

### Clonality of clock-like mutations

SBS1 individual mutations were identified as described in the previous section. For SBS5 and SBS40 mutations, we used a maximum likelihood approach to assign individual mutations to the SBS5 and SBS40 mutational signatures in a cancer-type-specific manner.

For every SBS1 (and SBS5/SBS40 mutation), we then assign the clonality according to the PURPLE subclonal likelihood estimation, in which only mutations with subclonal (SUBCL) likelihood ≥ 0.8 were considered as such (see above).

For each tumour sample, the SBS1 clonality ratio (or respectively SBS5/SBS40 clonality ratio) was defined as the ratio between the proportion of clonal SBS1 mutations ($\frac{\text{SBS 1 clonal mutations}}{\text{SBS 1 mutations}}$) divided by the total proportion of clonal alterations in the sample ($\frac{\text{Total clonal mutations}}{\text{Total mutations}}$).

## Primary SBS1 mutation rate and metastatic SBS1 age-corrected enrichment

We computed for each primary cancer type the average number of SBS1 per year as the number of SBS1 mutations divided by the age of the patient at biopsy (only considering primary samples and excluding hypermutated samples as described above). We then used a Spearman's correlation to assess its association with the estimated mean SBS1 mutation rate fold change in metastatic tumours (see above). In addition, to exclude potential biases in our primary cohort, we repeated the same analysis relying on an independent measurement of primary cancer SBS1 yearly accumulation. Specifically, we used the best-estimated accumulation of SBS1 per year from ref. 30 (Supplementary Table 6) and regressed it to the fold-change estimates for the matching cancer types present in both datasets.

## SV analysis

**Definitions of SV type.** LINX[55] chains one or more SVs and classifies these SV clusters into various event types ('ResolvedType'). We defined deletions and duplications as clusters with a ResolvedType of 'DEL' or 'DUP' whose start and end breakpoints are on the same chromosome (that is, intrachromosomal). Deletions and duplications were split into those less than 10 kb and 10 kb or more in length (small and large, respectively), based on observing bimodal distributions in these lengths across cancer types (Extended Data Fig. 5b). We defined complex SVs as clusters with a 'COMPLEX' ResolvedType, an inversion ResolvedType (including: INV, FB_INV_PAIR, RECIP_INV, RECIP_INV_DEL_DUP and RECIP_INV_DUPS) or a translocation ResolvedType (including: RECIP_TRANS, RECIP_TRANS_DEL_DUP, RECIP_TRANS_DUPS, UNBAL_TRANS and UNBAL_TRANS_TI). Complex SVs were split into those with less than 20 and 20 or more SVs (small and large, respectively), based on observing similar unimodal distributions in the number SVs across cancer types whose tail begins at approximately 20 breakpoints (Extended Data Fig. 5b). Finally, we defined long interspersed nuclear element insertions (LINEs) as clusters with a ResolvedType of 'LINE'. For each sample, we counted the occurrence (that is, SV burden) of each of the seven SV types described above. In addition, we determined the total SV burden by summing counts of the SV types.

**Comparing SV burden between primary versus metastatic cancer.** We then compared the SV-type burden between primary versus metastatic tumours as shown in Fig. 3a. First, we performed two-sided Mann–Whitney tests per SV type and per cancer type to determine whether there was a statistically significant difference in SV-type burden between primary versus metastatic. The Bonferroni method was used for multiple testing correction on the $P$ values from the Mann–Whitney tests (to obtain $q$ values). Next, we calculated relative enrichment as follows: $\log_{10}$(median SV-type burden in metastatic tumours + 1) − $\log_{10}$(median SV-type burden in primary tumours + 1); and calculated fold change as follows: (median SV-type burden in metastatic tumours + 1) / (median SV-type burden in primary tumours + 1). When calculating relative enrichment and fold change, the pseudocount of 1 was added to avoid the log(0) and divide by zero errors, respectively. Fold changes are displayed with a '>' in Fig. 3a when the SV burden for primary tumours is 0 (that is, when a divide by 0 would occur without the pseudocount). We considered the SV-type burden between primary versus metastatic to be significant when: $q$ < 0.05, and fold change ≥ 1.2 or fold change ≤ 0.8

**Identifying features associated with increased SV burden in metastatic cancer.** To identify the features that could explain increased SV burden, we correlated SV burden with various tumour genomic

features. This included: (1) genome ploidy (determined by PURPLE); (2) homologous recombination deficiency (determined by CHORD[33]) and MSI (determined by PURPLE) status; (3) the presence of mutations in 345 cancer-associated genes (excluding fragile site genes that are often affected by CN alterations[5]), hereafter referred to as 'gene status'; and (4) treatment history, including the presence of radiotherapy, the presence of one of the 79 different cancer therapies as well as the total number of treatments received. All primary samples and all metastatic samples without treatment information were considered to have no treatment. Genome ploidy and total number of treatments received were numeric features, whereas all of the remaining were boolean (that is, true or false) features. In total, there were 429 features.

SV-type burden was transformed to $\log_{10}$(SV-type burden + 1) and was correlated with the 429 features using multivariate linear regression models (LMs). This was performed separately for each of the seven SV types, and for each cancer type (or subtype). In the SV main analysis (Fig. 4b–f), there were 23 cancer types, resulting in a total of 161 (23 cancer types × 7 SV types) LM models.

Each LM model (that is, per SV type and cancer type) involved training of three independent LMs with (1) both metastatic and primary samples (primary + metastatic), (2) only Hartwig samples (metastatic only), and (3) only PCAWG samples (primary only). This was done to filter out correlations between features and increased SV-type burden solely due to differences in feature values between primary and metastatic tumours. We then required features that positively correlated with SV-type burden in the primary + metastatic LM to independently show the same association in the metastatic-only or primary-only LMs. Only genomic features that independently showed positive correlation with the SV burden were further considered as significant (that is, represented in the lollipop plots).

Each of the three LMs was trained as follows:
(1) Remove boolean features with too few 'true' samples.
   (i) For the primary + metastatic LM, remove gene status features with less than 15 'true' samples.
   (ii) For the metastatic-only and primary-only LMs, remove gene status features with less than 10 'true' samples.
   (iii) For the remaining boolean features, remove features with less than 5% 'true' samples.
(2) Fit a LM using the lm() base R function to correlate $\log_{10}$(SV-type burden + 1) versus all features.

For each LM analysis, we used the following filtering criteria to identify the features that were correlated with increased SV-type burden:
(1) Only keep LM analyses for which there was significant increase in SV-type burden for the respective cancer type ($P$ < 0.01 as described in the previous section 'Comparing SV burden between primary versus metastatic cancer').
(2) For primary + metastatic LM:
   (i) Regression $P$ < 0.01
   (ii) Coefficient $P$ < 0.01
   (iii) Coefficient more than 0
(3) For metastatic-only LM or primary-only LM:
   (i) Coefficient $P$ < 0.01
   (ii) Coefficient more than 0

Finally, to determine which features (of those correlated with increased SV-type burden) were enriched in metastatic tumours compared with primary tumours (and vice versa), we calculated Cliff's delta for numeric features and Cramer's $V$ for boolean features. Cliff's delta ranges from −1 to +1, with −1 representing complete enrichment in primary tumours, whereas +1 represents complete enrichment in metastatic tumours. Cramer's $V$ only ranges from 0 to 1 (with 1 representing enrichment in either primary or metastatic tumours), the sign of the log(odds ratio) was assigned as the sign of the Cramer's $V$ value so that it ranged from −1 to +1. Features with an effect size of more than 0

were considered as those that could explain the SV burden increase in metastatic cancer when compared with primary cancer.

### Driver alterations

We relied on patient-specific cancer driver and fusion catalogues constructed by PURPLE[5] and LINX[55]. Only drivers with a driver likelihood of more than 0.5 were retained. Fusion drivers were filtered for those that were previously reported in the literature. Similarly, we manually curated the list of drivers and removed *SMAD3* hotspot mutations because of the high-burden mutations in low-mappability regions. The final driver catalogue contained a total of 453 driver genes and the final fusion catalogue contained 554 reported fusions.

To compare the number of drivers in primary and metastatic tumours, we then combined fusions with the LINX driver variants to calculate a patient-specific number of driver events. Drivers that concern the same driver gene but a different driver type were deemed to be single drivers (for example, *TP53* mutation and *TP53* deletion in the same sample were considered as one driver event). Cancer-type-specific Mann–Whitney test was performed to assess differences between primary and metastatic tumours. An adjusted $q < 0.01$ was deemed to be significant.

To assess the driver enrichment, a contingency matrix was constructed from the driver catalogue, containing the frequency of driver mutations per driver type (that is, deletion, amplification or mutations) and cancer type in each cohort (metastatic and primary). A second contingency matrix was constructed for the fusions. Partial amplifications were considered as amplifications, whereas homologous disruptions were considered as deletions. These contingency matrices were filtered for genes that show a minimum frequency of five mutated samples in either the primary or the metastatic cohorts. Then, a two-sided Fisher's exact test for each gene, cancer type and mutation type was performed and the *P* value was adjusted for FDR per cancer type. Cramer's *V* and the odds ratio were used as effect size measures. An adjusted $P < 0.01$ was deemed to be significant.

### Therapeutic actionability of variants

To determine the amount of actionable variants observed in each sample, we compared our variants annotated by SnpEff (v5.1)[56] to those derived from three different databases (OncoKB[57], CIViC[58] and CGI[59]) that were classified based on a common clinical evidence level (https://civic.readthedocs.io/en/latest/model/evidence/level.html) as previously described[5]. In our study we only considered A and B levels of evidence, which represent variants that have been FDA approved for treatment and are currently being evaluated in a late-stage clinical trial, respectively. A variant was determined to be 'on-label' when the cancer type matches the cancer type for which the treatment was approved for or is being investigated for, and 'off-label' otherwise. Only actionable variants of the sensitive category were considered (that is, tumours containing the variant are sensitive to a certain treatment). Sample-level actionable variants such as TMB high/low or MSI status were not evaluated, because of their tendency to overshadow the other variants, especially in the off-label category. Furthermore, wild-type actionable variants were not considered in this analysis for the same reason. Variants related to gene expression or methylation were not considered due to lack of available data. In addition, we found actionable variants derived from leukaemias to be very different from the solid tumours in our dataset, which is why we excluded them for this analysis. For the analysis of proportion of samples bearing therapeutically actionable variants, we considered that the highest evidence level was retained for each sample following the order A on/off-label to B on/off-label. To assess enrichment of actionable variants globally and at the A on-label level in metastatic tumours, a Fisher's exact test was performed pan-cancer-wide and per cancer type. An adjusted $P < 0.05$ was deemed to be significant. Fold changes in frequency are only shown for cancer types with a global significant difference.

To determine which variants contribute the most to the observed significant frequency differences, individual actionable variants were tested for enrichment in metastatic tumours using a Fisher's exact test per cancer type and tier level. *P* values were FDR adjusted per cancer type and $q < 0.05$ was deemed to be significant. In Extended Data Fig. 8, only actionable variants from cancer types with a global significant difference (see above) and that were found at a minimum frequency of 5% in either primary or metastatic cohort and a minimum frequency difference of 5% between them were shown. However, the differences across all screened variants are available as part of Supplementary Table 7.

### TEDs

We aimed to pinpoint drivers that are potentially responsible for lack of response to certain cancer treatments in the metastatic cohort. Hence, we devised a test that identifies driver alterations that are enriched in groups of patients treated with a particular treatment type compared with the untreated group of patients from the same cancer type (see Extended Data Fig. 9a for illustration of the workflow).

Treatments were grouped according to their mechanism of action so that multiple drugs with a shared mechanism of action were grouped into the mechanistic treatment category (for example, cisplatin, oxaliplatin and carboplatin were grouped as platinum). We created 323 treatment and cancer-type groups by grouping patients with treatment annotation according to their treatment record before the biopsy. One patient might be involved in multiple groups if they have received multiple lines of therapy or a simultaneous combination of multiple drugs. Only 92 treatment and cancer-type groups with at least ten patients were further considered in the analysis.

Hence, for each cancer type (or subtype, in the case of breast and colorectal) and treatment group, we performed the following steps:

(1) We first performed a driver discovery analysis in treatment and cancer-type (or subtype)-specific manner. We explored three types of somatic alterations: coding mutations, non-coding mutations and CN variants (see below for detailed description of each driver category). Driver elements from each alteration category were selected for further analysis.

(2) For each driver alteration from (1), we compared the alteration frequency in the treated group to the untreated group of the same cancer type. Each driver category (coding and non-coding mutations and CN variants) were evaluated independently. We performed a Fisher's exact test to assess the significance of the frequency differences. Similarly, we computed the odds ratio of the mutation frequencies for each driver alteration. The *P* values were adjusted with a multiple-testing correction using the Benjamini–Hochberg procedure ($\alpha = 0.05$). An adjusted *P* value of 0.05 was used for coding mutations and CN variants. An adjusted *P* value of 0.1 was used for non-coding variants due to the overall low mutation frequency of the elements included in this category, which hampered the identification of significant differences.

(3) We then annotated each driver element with information about the exclusivity in the treatment group. We labelled drivers as treatment exclusive if the mutation frequency in the untreated group was lower than 5% or we annotated as treatment enriched otherwise. In addition, we manually curated the identified drivers with literature references of their association with each treatment category.

(4) Finally, the overlap of patients in multiple treatment groups (see above) in the same cancer type prompted us to prioritize the most significant treatment association for each driver gene in a particular cancer type. In other words, for each driver gene that was deemed as significantly associated with multiple treatment groups in the same cancer type, we selected the most significant treatment association, unless a driver-treatment annotation was clearly reported in the literature.

The full catalogue of TEDs and their mutation frequencies can be found in Supplementary Table 8.

## Coding mutation drivers

We used dNdScv (v0.0.1)[60] with default parameters to identify cancer driver genes from coding mutations. A global $q < 0.1$ was used as a threshold for significance. Mutation frequencies for each driver gene were extracted from the dNdScv output. We defined the mutation frequency as the number of samples bearing non-synonymous mutations.

## Non-coding mutation drivers

We used ActiveDriverWGS[61] (v1.1.2, default parameters) to identify non-coding driver elements in five regulatory regions of the genome including 3′ untranslated regions (UTRs), 5′ UTRs, long non-coding RNAs, proximal promoters and splice sites. For each element category, we extracted the genomic coordinates from Ensembl v101. Each regulatory region was independently tested. To select for significant hits, we filtered on adjusted $P$ values (FDR < 0.1) and a minimum of three mutated samples. We defined the mutation frequency as the number of mutated samples for each significantly mutated element in the treatment group.

## CN variant drivers

We ran GISTIC2 (ref. 62) (v2.0.23) on each of the 92 treatment and cancer-type groups using the following settings:

```
gistic2 -b <inputPath> -seg <inputSegmentation> -refgene hg19.
UCSC.add_miR.140312.refgene.mat -genegistic 1 -gcm extreme
-maxseg 4000 -broad 1 -brlen 0.98 -conf 0.95 -rx 0 -cap 3 -savesseg 0
-armpeel 1 -smallmem 0 -res 0.01 -ta 0.1 -td 0.1 -savedata 0 -savegene
1 -qvt 0.1.
```

The focal GISTIC peaks ($q \leq 0.1$ and <1 Mb) were then annotated with functional elements using the coordinates from Ensembl v101. The frequency differences between treated and untreated cohorts on every gene was assessed with Fisher's exact test as described above. For this, we first calculated the focal amplification and deep depletion status of every gene within each sample. A gene was amplified when the ploidy level of the gene was 2.5 ploidy levels higher than its genome-wide mean ploidy level (as measured by PURPLE), and deleted when the gene ploidy level was lower than 0.3 (that is, deep deletion). We observed that the majority of the peaks contained multiple significant gene candidates (after multiple correction $q < 0.05$) and therefore we retained the gene most closely positioned to the peak summit, which is the most significantly enriched region across the treated samples. Next, we also found recurrent peaks across multiple treatment groups per cancer type that are not, or less, present in the untreated control group because most of the Hartwig samples have received multiple treatment types. We therefore merged peaks with overlapping ranges to produce a single peak per genomic region per cancer type. For each collapsed peak, we selected the treatment type showing the lowest $q$ value for the gene near the peak summit. Deletion and amplification peaks were processed separately.

## Group-level aggregation of treatment resistance-associated variants

To estimate the contribution of TEDs to the total number of drivers per sample in the metastatic cohort, we excluded any TED from the catalogue of driver mutations (see the above section 'Driver alterations') in a cancer-type-specific, gene-specific and driver-type-specific manner.

## Reporting summary

Further information on research design is available in the Nature Portfolio Reporting Summary linked to this article.

## Data availability

Metastatic WGS data and metadata from the Hartwig Medical Foundation are freely available for academic use through standardized procedures. Request forms can be found at https://www.hartwigmedicalfoundation.nl/en/data/data-acces-request/. Somatic variant calls, gene driver lists, CN profiles and other core data of the PCAWG cohort generated by the Hartwig analytical pipeline are available for download at https://dcc.icgc.org/releases/PCAWG/Hartwig. Researchers will need to apply to the ICGC data access compliance office (https://daco.icgc-argo.org) for the ICGC portion of the dataset. Similarly, users with authorized access can download the TCGA portion of the PCAWG dataset at https://icgc.bionimbus.org/files/5310a3ac-0344-458a-88ce-d55445540120. Additional information on accessing the data, including raw read files, can be found at https://docs.icgc.org/pcawg/data/. References and download links to the original independent datasets used in the analyses are included in each of the pertinent sections of the Methods and Supplementary Notes, and a full list of all datasets used in the present study can be found in the data availability section of the Reporting Summary file.

## Code availability

The Hartwig analytical processing pipeline is available (https://github.com/hartwigmedical/pipeline5) and implemented in Platinum (https://github.com/hartwigmedical/platinum). The source code to reproduce the analysis of the manuscript is available in the following repository: https://github.com/UMCUGenetics/primary-met-wgs-comparison.

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

**Acknowledgements** This publication and the underlying study have been made possible partly on the basis of the data that Hartwig Medical Foundation and the Center of Personalized Cancer Treatment (CPCT) have made available to the study. We thank R. Janssen for the technical assistance in the collection and processing of the PCAWG raw sequencing data using the Hartwig tumour analytical pipeline; USEQ from UMC Utrecht for providing high-performance computing power; J. Mateo for his valuable scientific input; L. Stein and L. Xiang for their assistance in the publication of the reprocessed ICGC part of the PCAWG dataset; R. Grossman, C. Meyer and T. Simmons for their assistance in the publication of the reprocessed TCGA part of the PCAWG; D. Quigley for providing the somatic mutation calls and age at biopsy of the patients with prostate cancer from their study; S. Turajlic and H. Pallikonda for their responsiveness concerning the TRACERx renal dataset; and P. Wolfe and other staff of the Hartwig Medical Foundation team for aligning the processing of PCAWG and Hartwig datasets.

**Author contributions** F.M.-J., A.V.H. and E.C. conceptualized the study. F.M.-J., A.M., S.R.B., L.N., P.P. and A.V.H. came up with the methodology. F.M.-J., A.M., S.R.B., L.N., P.P. and A.V.H. provided software. F.M.-J., A.M., S.R.B., L.N., P.P. and A.V.H. validated results. F.M.-J., A.M., S.R.B., L.N., P.P., A.V.H. and E.C. conducted formal analysis. F.M.-J., A.M., S.R.B., L.N., P.P. and A.V.H.

performed the investigations and provided resources. F.M.-J., A.M., S.R.B., L.N., P.P. and A.V.H. curated the data. F.M.-J. and A.V.H. wrote the original draft of the manuscript. F.M.-J., A.M., S.R.B., L.N., P.P., A.V.H. and E.C. reviewed and edited the manuscript. F.M.-J., A.M., S.R.B., L.N., A.V.H. and E.C. performed visualization. F.M.-J., A.V.H. and E.C. supervised the project. A.V.H. and E.C. provided project administration. E.C. acquired funding.

**Competing interests** The authors declare no competing interests.

**Additional information**

**Correspondence and requests for materials** should be addressed to Edwin Cuppen.

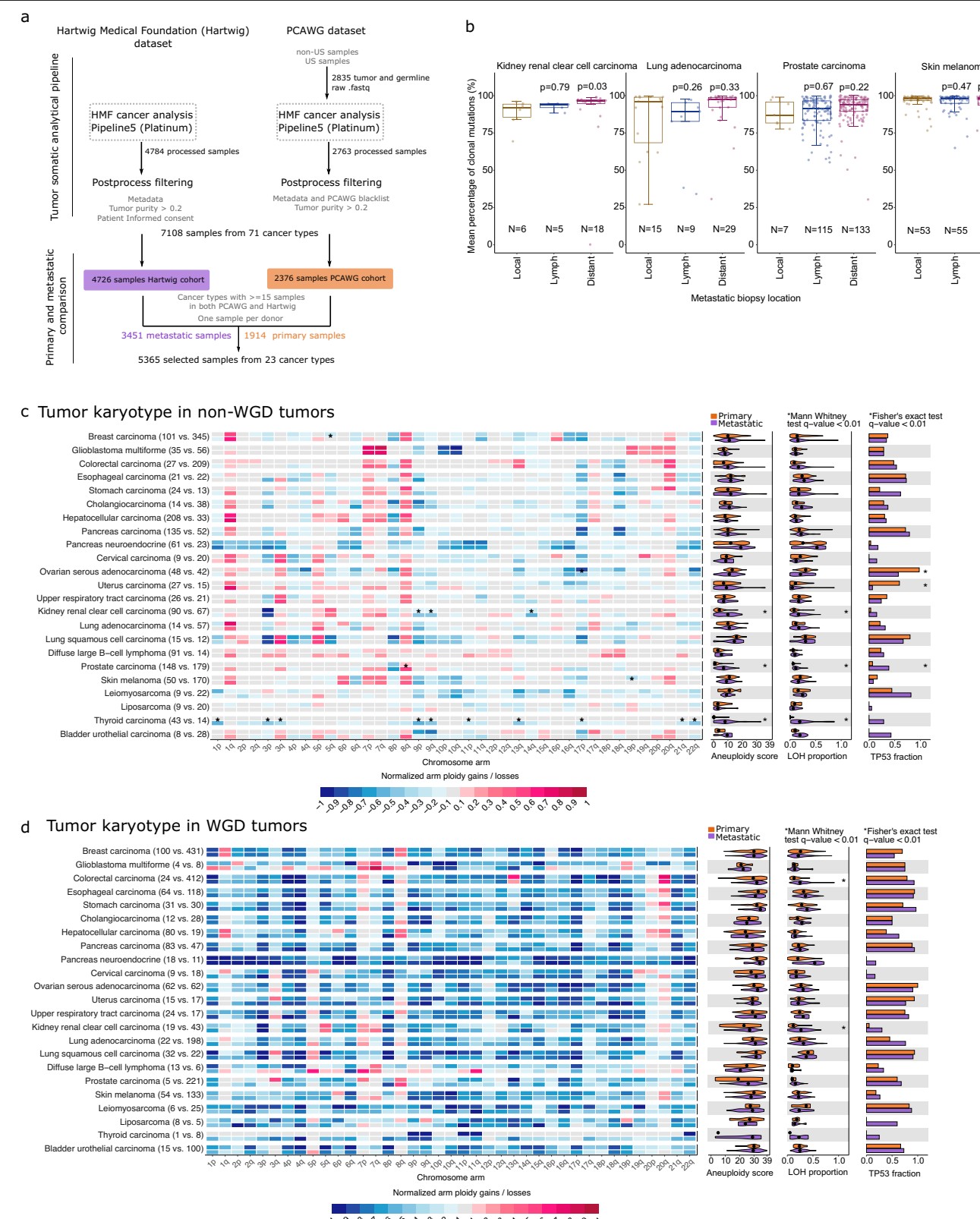

**Extended Data Fig. 1** | See next page for caption.

**Extended Data Fig. 1 | Cohort overview and global genomic features.**
**a)** Workflow of the unified processing pipeline used in this study for Hartwig (left) and PCAWG (right) WGS samples. First, PCAWG tumor and matched normal raw sequencing files were gathered and re-processed using the Hartwig tumor analytical pipeline. Next, the output of tumor samples that were correctly processed by the pipeline were further subjected to a strict quality control filtering. As a result, a total of 7,108 samples from 71 cancer types compose the harmonized dataset. 5,365 patient tumor samples from 23 cancer types with sufficient representation in both primary and metastatic datasets were selected for this study. **b)** Tumor clonality according to the metastatic biopsy location in kidney renal clear cell carcinoma, lung adenocarcinoma, prostate carcinoma and skin melanoma. N, number of samples in the group. p, two-sided Mann-Whitney p-value. Box-plots: center line, median; box limits, first and third quartiles; whiskers, lowest/highest data points at first quartile minus/plus 1.5× IQR. **c)** Left, similar to Fig. 1d only including non-WGD tumors. Right, similar to Fig. 1e for non-WGD tumors. **d)** Equivalent to c), but limited to WGD tumors. '*', two-sided Mann-Whitney adjusted p-value < 0.01 for continuous variables and two-sided Fisher's exact test adjusted p-value < 0.01 for *TP53*.

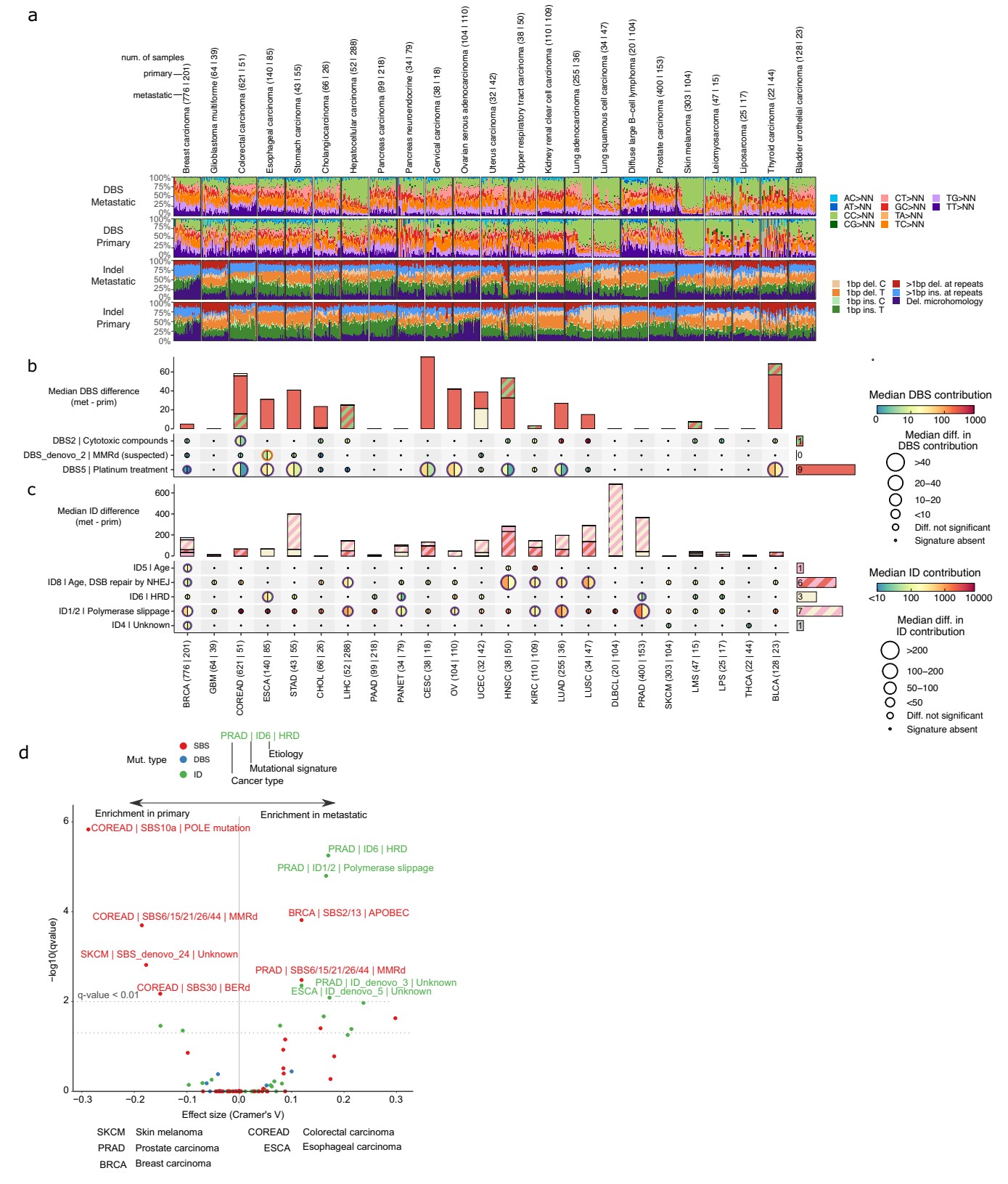

**Extended Data Fig. 2 |** See next page for caption.

**Extended Data Fig. 2 | Mutation burden and mutational signatures. a)** double-base substitutions (DBSs, top) and indels (IDs, bottom) mutational spectra of metastatic and primary tumors. Patients are ordered according to their TMB burden. **b)** Moon plot representing the DBS burden differences attributed to each mutational signature in metastatic (left) and primary (right) tumors. Edge thickness and colors represent significant differences (Mann-Whitney adjusted p-value<0.05, ±1.4x fold change) and the direction of the enrichment, respectively. The size of circles are proportionate to the mutation burden difference. Right bars, number of metastatic cancer types with a mutational signature significant enrichment. Top stacked bars represent the cumulative signature exposure difference. Thicker bar edge lines represent significance. Bars are coloured according to the annotated etiology. Only mutational signatures with known etiology or with at least one cancer type with significant metastatic enrichment are included. **c)** analogous representation for IDs. **d)** Volcano plot representing the mutational signature hypermutation (>10,000 mutations for SBS, >500 for DBS, and >1000 for ID) prevalence comparison between primary and metastatic tumor patients. Y-axis, $\log_{10}$(two-sided Mann-Whitney adjusted p-value). X-axis, effect size as Cramer's V. Each dot represents a mutational signature in a cancer type. Dots are coloured according to the mutation type. Diff., difference. Muts. mutations. Sig., mutational signature. Mut. mutational. Susp., suspected. Def., deficiency.

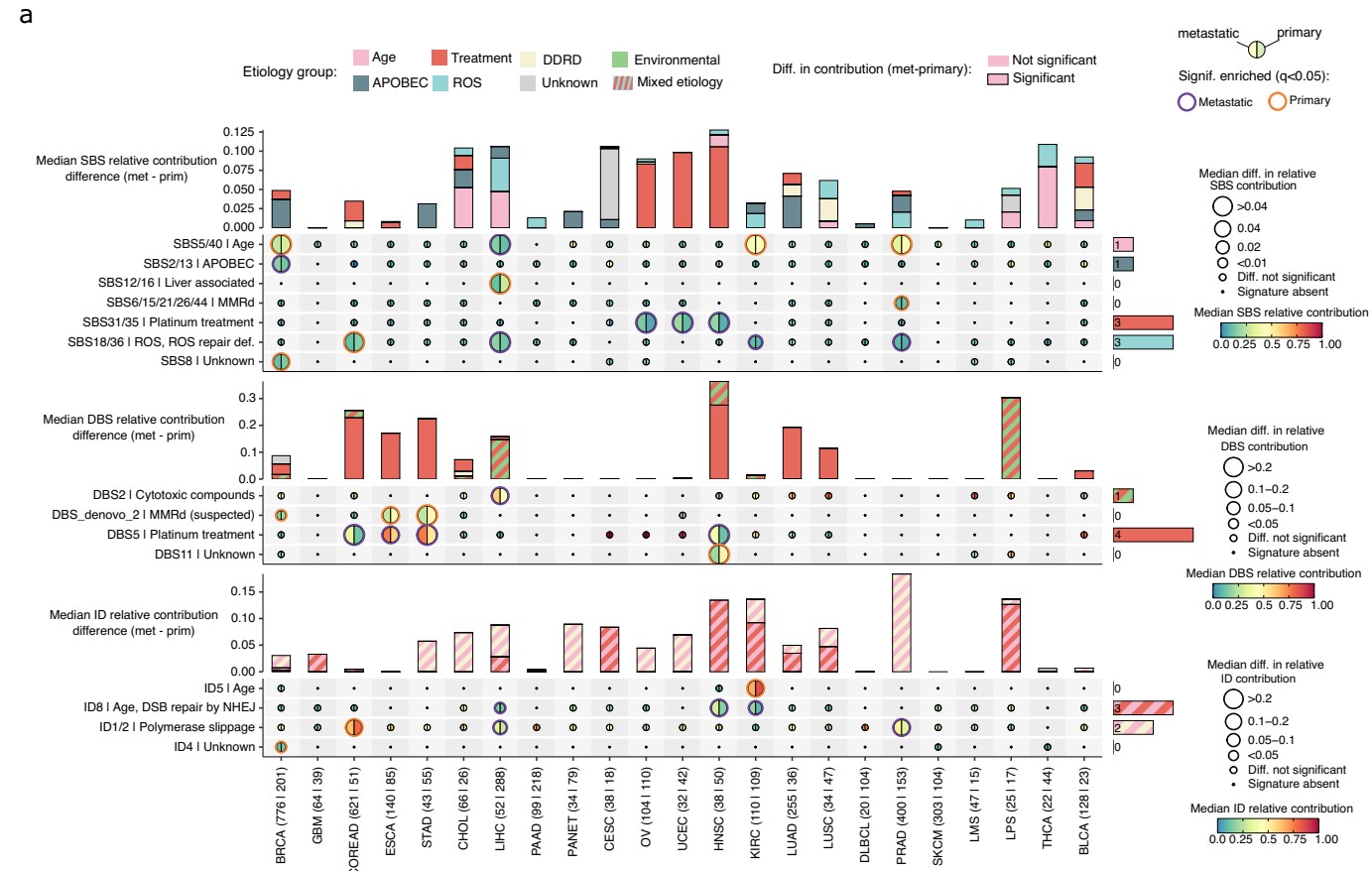

**Extended Data Fig. 3 | Mutational signature relative contribution comparison. a)** From top to bottom, moon plot representing the SBS, DBS and ID relative contribution differences attributed to each mutational signature. The size of circles are proportional to the relative mutation burden difference. Top stacked bars represent the relative signature exposure difference. Thicker bar edge lines represent significance (two-sided Mann-Whitney adjusted p-value < 0.05 and ≥1% difference in relative contribution). Bars are coloured according to the annotated etiology. Right bars, number of metastatic cancer types with a mutational signature significant enrichment. Only mutational signatures with known etiology or with at least one cancer type with significant enrichment are included. Diff., difference. Muts. mutations. Sig., mutational signature. Mut. mutational. Susp., suspected. Def., deficiency.

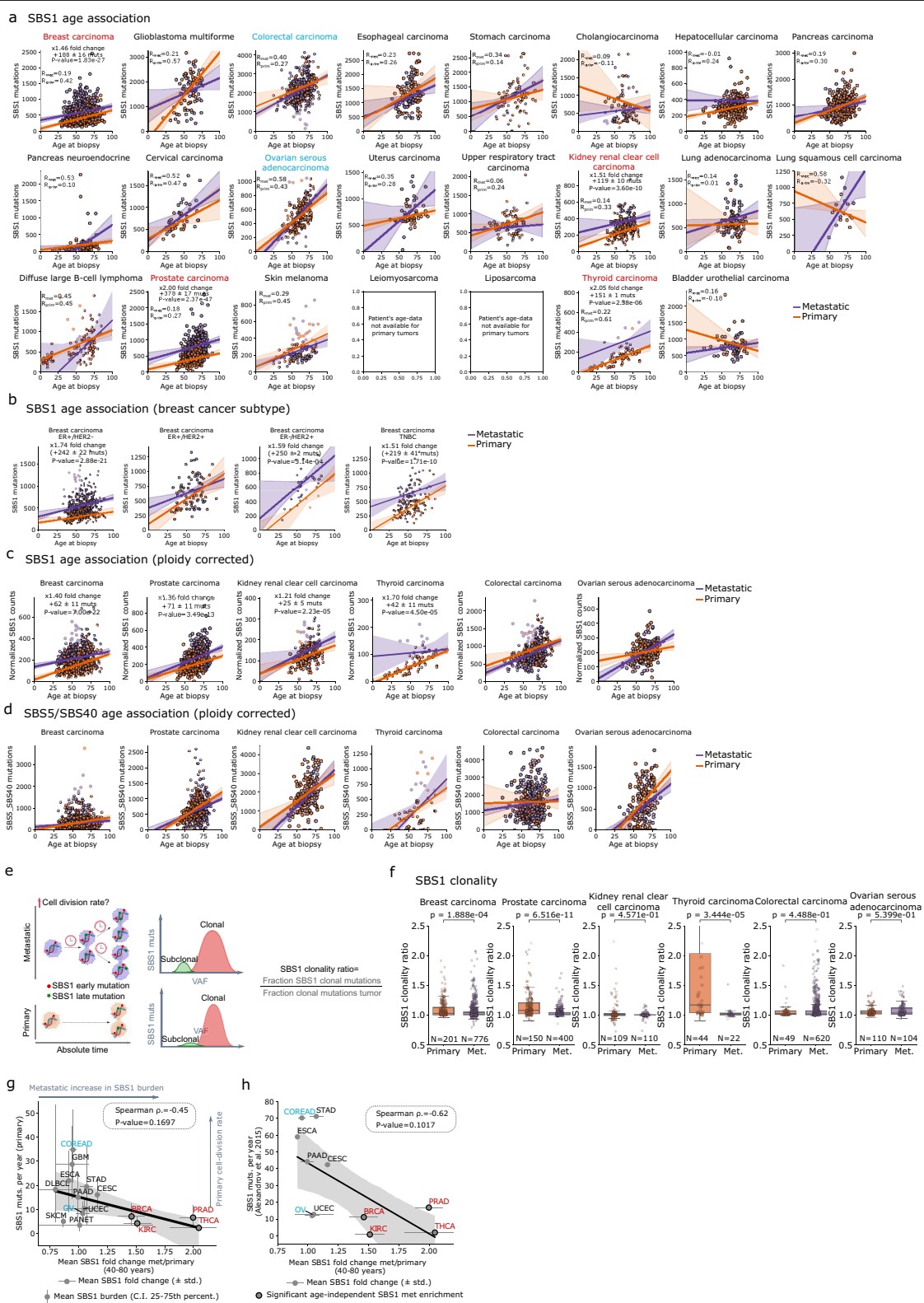

**Extended Data Fig. 4** | See next page for caption.

**Extended Data Fig. 4 | Age-corrected SBS1 mutation burden in primary and metastatic tumors. a)** Linear regression of the SBS1 mutation burden (y-axis) and patient's age at biopsy (x-axis) in primary and metastatic cancer across the 23 cancer types. The median trendline and 99% confidence intervals of the linear regression are represented as a solid line and the adjacent shaded area, respectively. The mean fold change, mean SBS1 increase per year and one sided Mann-Whitney p-value are only displayed in cancer types with an age-independent significantly different primary and metastatic distribution. Red labels, significant increase in metastatic tumors. Blue, control cancer types. $R_{met}$ and $R_{prim}$, Pearson correlation coefficient of the metastatic and primary linear regressions, respectively. **b)** Analogous representation for independent linear regressions for breast cancer subtypes. **c)** Relative to a) for ploidy corrected SBS1 in the tumor types of interest. **d)** Relative to a) for ploidy corrected SBS5/40 counts in the tumor types of interest. **e)** Depiction illustrating the potential effect of an increased cell division rate in metastatic tumors compared to primary and its expected impact on the SBS1 variant allele frequency (VAF) distribution. Partially created BioRender.com. **f)** Comparison of global SBS1 clonality ratios between primary and metastatic in breast, prostate, kidney renal clear cell, thyroid, colorectal and ovarian serous carcinomas. Boxplots are defined as in Fig. 1. P, two-sided Mann-Whitney p-value. N, number of samples. **g)** Spearman correlation analysis of the mean SBS1 year burden of primary tumors (y-axis) and the mean metastatic SBS1 fold change (x-axis) across the 15 cancer types with linear association between age and SBS1 accumulation. Vertical error bars represent the 25th and 75th percentile, respectively. Horizontal error bars represent the standard deviation of the mean fold change (metastatic divided by primary) of the SBS1 yearly mutation burden. The median trendline and 99% confidence intervals of the linear regression are represented as a solid line and the adjacent shaded area, respectively. Cancer types with a significantly different SBS1 mutation rate are marked by thicker marker borders and with red labels. Blue labels represent the control cancer types. **h)** Similar but using SBS1 year mutation rate from ref. 30. To derive vertical and horizontal error bars in panels g) and h) all tumor samples from the primary and metastatic cohorts from panel a) (see Methods for inclusion criteria) were included in the analysis. The number of included samples per cancer type and cohort are available in Supplementary Table 4. Muts, mutations.

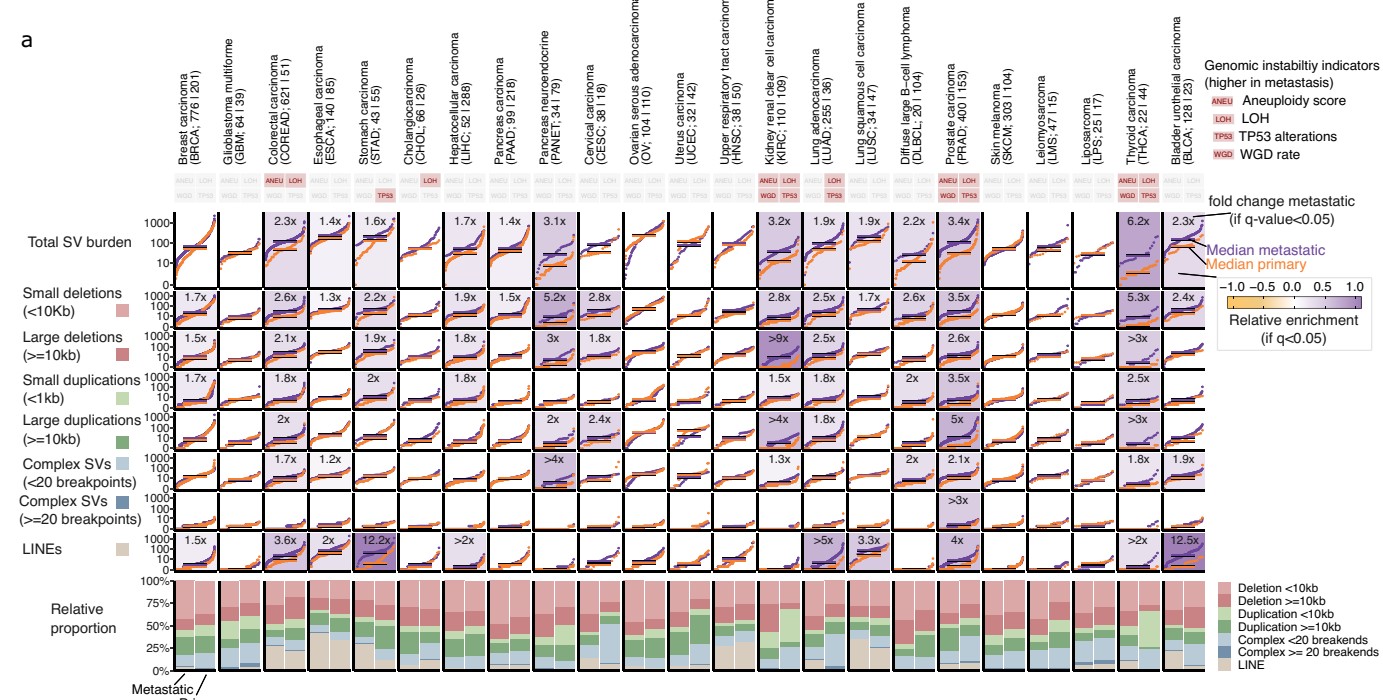

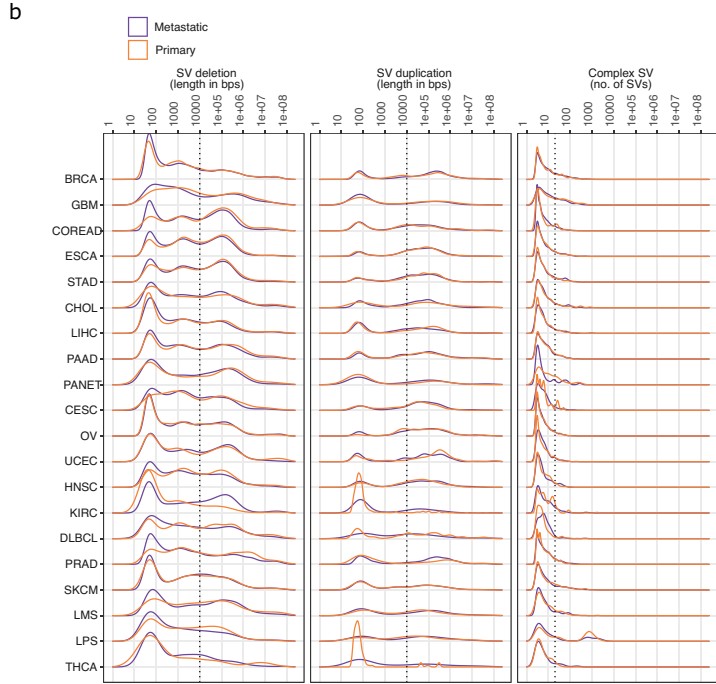

**Extended Data Fig. 5 | Structural variant burden. a)** Top rectangles represent the four genomic instability features defined in Fig. 1e. A red background represents significant enrichment in the metastatic cohort (two-sided Mann-Whitney adjusted p-value <0.01). S-plots, cumulative distribution function plot (samples ranked independently for each SV type) of tumor mutation burden for each cancer type for (from top to the bottom) the aggregated structural variant (SV) burden, small deletions (<10kb), large deletions (>=10kb), small duplications (<1kb), large duplications (>=10kb), complex events (<20 breakpoints), complex events (>=20 breakpoints) and LINEs insertions. Horizontal lines represent median values. Backgrounds are coloured according to the relative enrichment, defined as: $\log_{10}$(median SV type burden in metastatic tumors + 1) – $\log_{10}$(median SV type burden in primary tumors + 1). Fold change labels and coloured backgrounds are displayed when Mann-Whitney comparison renders a significant q-value < 0.05. Fold change labels are displayed with '>' when the SV burden for primary tumors is 0 (see Methods for more details). For each cancer type, bottom bar plots represent the relative fraction of each SV type in the metastatic (left) and primary (right) datasets. **b)** SV length frequency distribution of deletions (left panel) and duplications (middle panel). Right panel shows the frequency distribution of the number of linked breakpoints for complex SVs. Dashed vertical lines represent the chosen threshold to separate between short and large deletions, duplications and complex SVs, respectively.

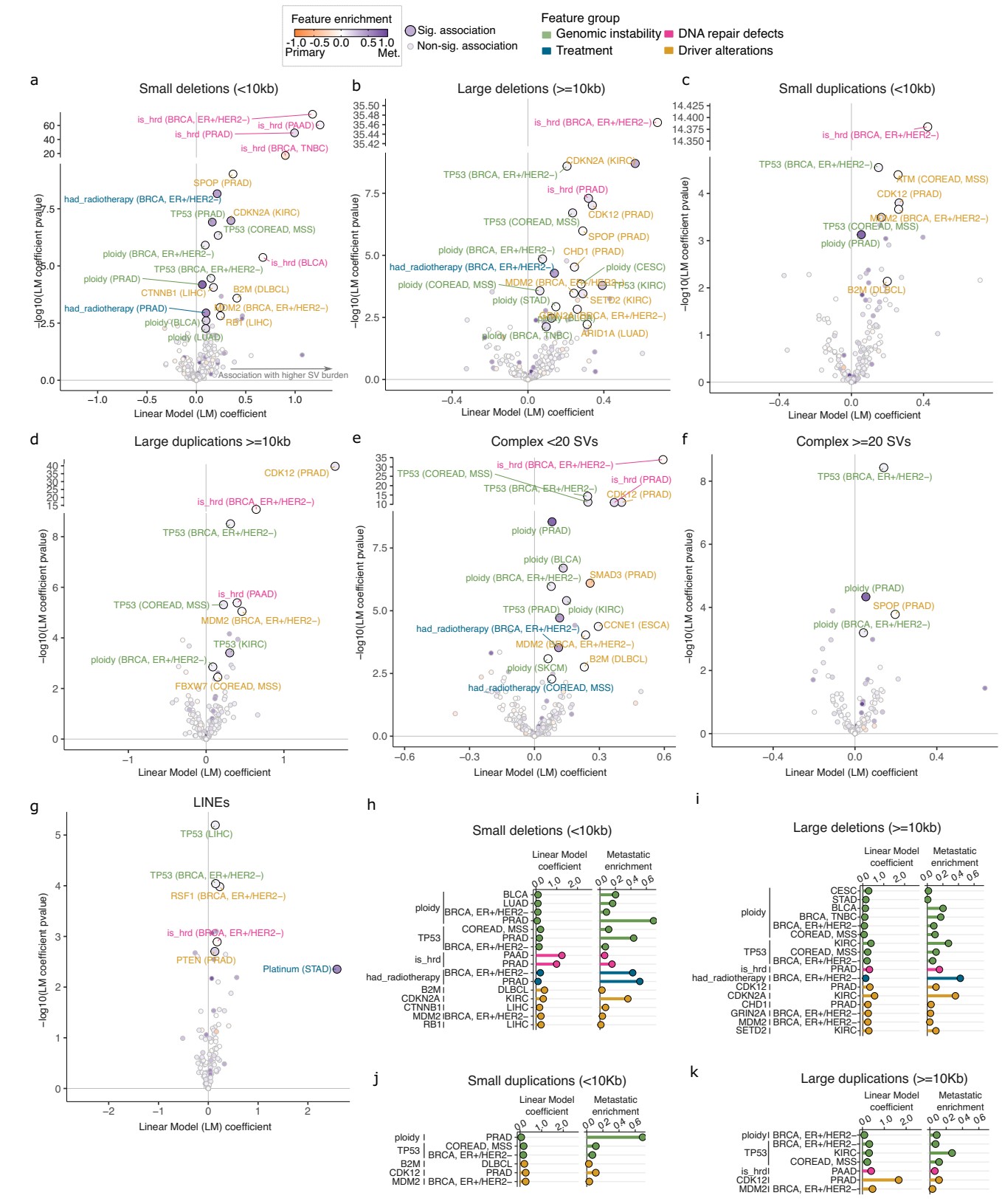

**Extended Data Fig. 6** | See next page for caption.

**Extended Data Fig. 6 | Structural variant burden associated genomic features. a)** Volcano plot representing the cancer type-specific regression coefficients (x-axis) and significance (y-axis, measured by the linear regression model coefficient p-value) of clinical and genomic features against the number of small deletions. Each dot represents one feature in one cancer type. Labels are coloured according to the feature category. Dots are coloured by the frequency enrichment in metastatic (purple) or primary (orange) patients. Analogous panels are displayed for **b)** large deletions, **c)** short duplications, **d)** large duplications, **e)** short complex SVs, **f)** large complex SVs and **g)** LINEs.

**h)** Lollipop plots representing the regression coefficients (left, relative to panel b. x-axis) and metastatic enrichment (right, relative to dots colour from panel b.) of features associated with small deletions. Only significant features (LM>0.0, LM coefficient p-value < 0.01 and with independent significance in primary or metastatic tumors) enriched in metastatic tumor patients (enrichment > 0.0) are displayed. **i), j)** and **k)** are identical but referring to large deletions, small duplications and large duplications, respectively. LM, linear model. Coef, coefficient.

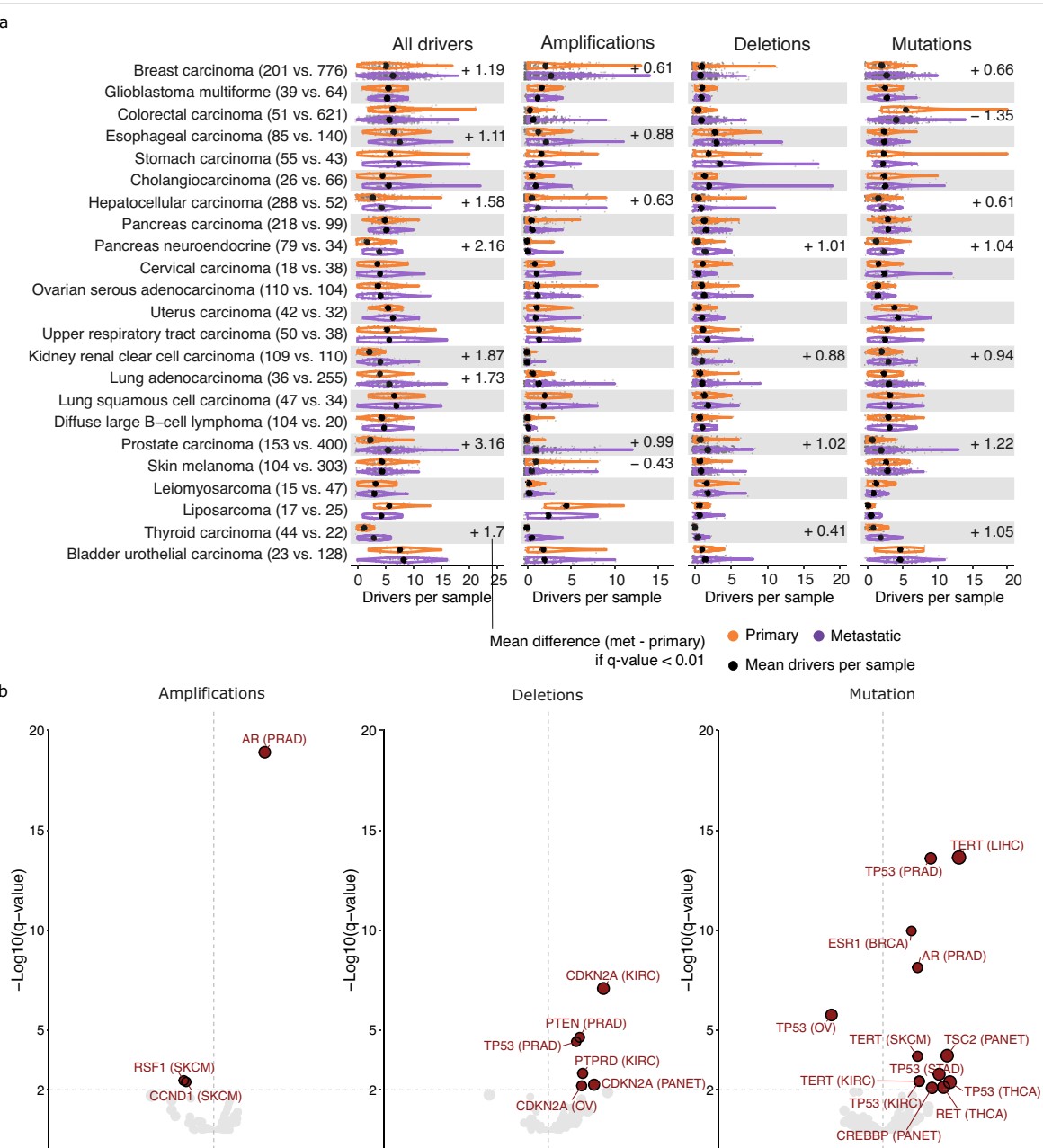

**Extended Data Fig. 7 | Driver landscape and drivers per patient. a)** Cancer type-specific distribution of number of driver alterations, amplifications, deletions and mutations per patient in primary (top) and metastatic (bottom). Black dots represent the mean values. Labels display mean differences (metastatic - primary) in cancer types with a significant difference. "*", two-sided Mann-Whitney adjusted p-value < 0.01. **b)** Volcano plots representing the cancer type-specific enrichment (x-axis) and significance (y-axis, FDR adjusted two-tailed Fisher's exact test p-value) of driver genes between primary and metastatic cohorts. From left to right, amplification drivers, biallelically deleted drivers and mutated driver genes. BRCA, Breast carcinoma. KIRC, kidney renal clear cell carcinoma. OV, Ovarian serous adenocarcinoma. PRAD, Prostate carcinoma. SKCM, Skin melanoma. THCA, Thyroid carcinoma. LIHC, Hepatocellular carcinoma. PANET, pancreatic neuroendocrine.

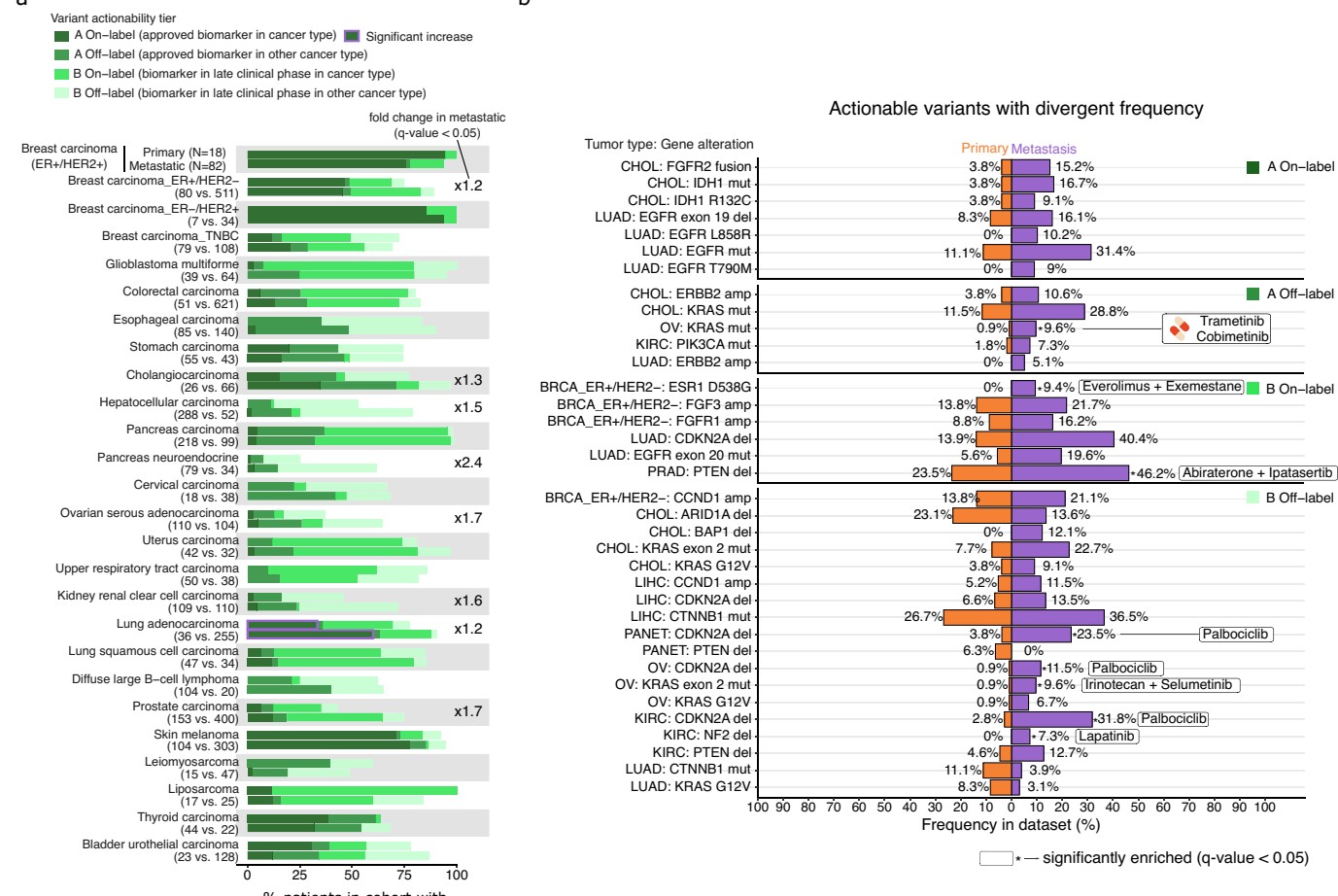

**Extended Data Fig. 8 | Therapeutic actionability of variants. a)** Cancer type-specific fraction of primary (top) and metastatic (bottom) patients with reported therapeutically actionable variants. For each patient the variant with the greatest level of evidence was considered. Bars are coloured according to the variant actionability tiers. Fold change (i.e., metastatic divided by primary fraction) labels are displayed in cancer types with a significant proportional increase (two-sided Fisher's exact test adjusted p-value < 0.05). Purple edgelines highlight significant increase in metastatic A-on label fraction

patients. **b)** Primary (left) and metastatic (right) alteration frequency of actionable variants with a high discrepancy (>5% frequency difference) from cancer types with a global significant increase of actionable variants in metastatic tumor patients from panel a). "*", a two-sided Fisher's exact test adjusted p-value < 0.05. Text boxes include the associated treatments for alterations with a significant mutation frequency increase in metastatic tumor patients.

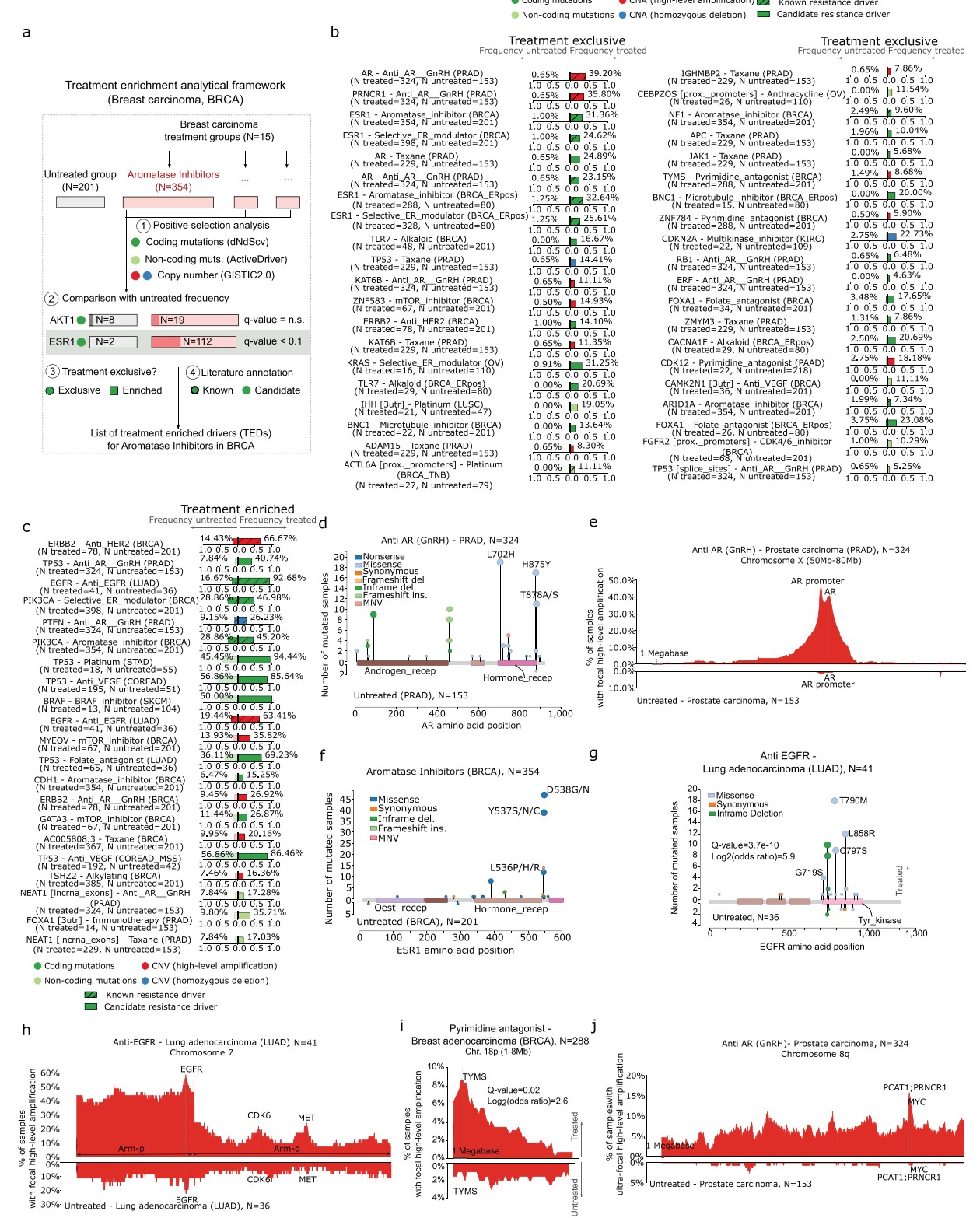

**Extended Data Fig. 9** | See next page for caption.

**Extended Data Fig. 9 | Treatment enriched drivers. a)** Visual depiction of the analytical framework to identify treatment enriched drivers (TEDs). Example, identification of TEDs in the 354 breast carcinoma patients treated with aromatase inhibitors. (1), identification of cancer driver genes from coding mutations (green), non-coding mutations (soft green), copy number amplifications (red) and deletions (blue). (2), for each driver gene, comparison of the alteration frequency in treated and untreated patients. (3) and (4), annotation of TEDs with type of enrichment and orthogonal evidence **b)** Side by side alteration frequency comparison between treated (right bar) and untreated (left bar) patients for all treatment-exclusive and **c)** treatment-enriched TEDs. **d)** Distribution of mutations along the *AR* protein sequence in prostate cancer patients treated with androgen deprivation (top) and untreated (bottom). Pfam domains are represented as rectangles. Mutations are coloured according to the consequence type. **e)** Distribution of focal copy number gains in chromosome X in prostate untreated patients (bottom) and treated with androgen deprivation (top). *AR* coding region and the promoter region are highlighted. **f)** Distribution of mutations along the *ESR1* protein sequence in breast carcinoma patients treated with aromatase inhibitors (top) and untreated (bottom). **g)** Distribution of mutations along the *EGFR* protein in lung adenocarcinoma patients treated with EGFR inhibitors (top) and untreated (bottom). Pfam domains are represented as rectangles. **h)** Distribution of focal copy number gains in chromosome 7 in lung adenocarcinoma untreated patients (bottom) and treated with anti-EGFR (top). *EGFR*, *MET* and *CDK6* genomic locations are highlighted. **i)** Distribution of focal copy number gains in chromosome chr18p:1Mb-8Mb in breast carcinoma untreated patients (bottom) and treated with pyrimidine antagonists (top). TYMS genomic location is highlighted. **j)** Similar to f) but representing ultra-focal (shorter than 3Mbs) MYC and PRNCR1 amplifications in chromosome 8q. In all copy number gain plots each bin represents 100Kbs. Mb, megabase. Kb, kilobase.

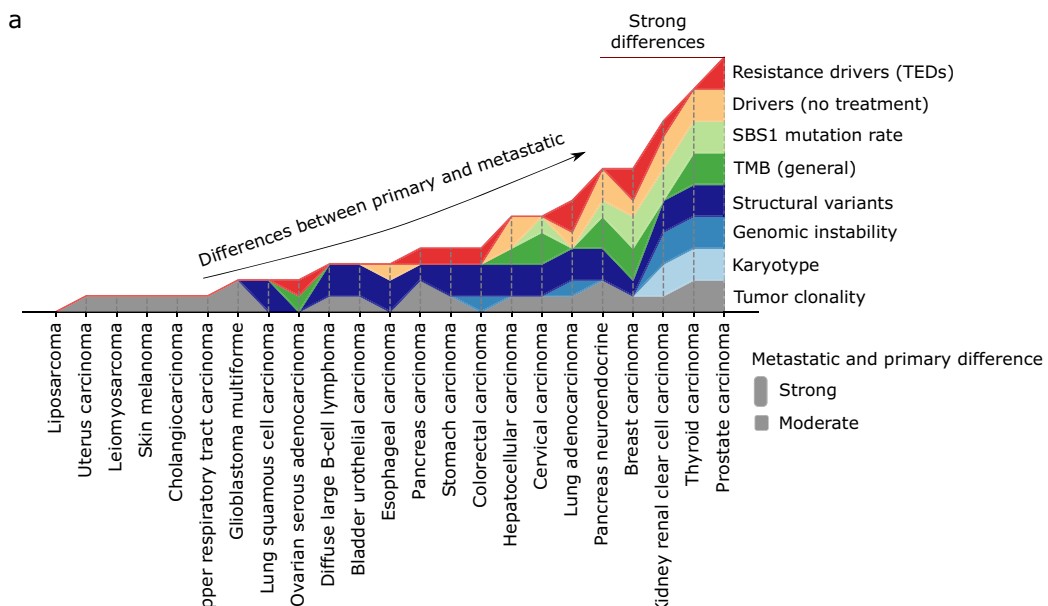

a

Strong differences

Differences between primary and metastatic

Resistance drivers (TEDs)
Drivers (no treatment)
SBS1 mutation rate
TMB (general)
Structural variants
Genomic instability
Karyotype
Tumor clonality

Liposarcoma
Uterus carcinoma
Leiomyosarcoma
Skin melanoma
Cholangiocarcinoma
Upper respiratory tract carcinoma
Glioblastoma multiforme
Lung squamous cell carcinoma
Ovarian serous adenocarcinoma
Diffuse large B-cell lymphoma
Bladder urothelial carcinoma
Esophageal carcinoma
Pancreas carcinoma
Stomach carcinoma
Colorectal carcinoma
Hepatocellular carcinoma
Cervical carcinoma
Lung adenocarcinoma
Pancreas neuroendocrine
Breast carcinoma
Kidney renal clear cell carcinoma
Thyroid carcinoma
Prostate carcinoma

Metastatic and primary difference
Strong
Moderate

**Extended Data Fig. 10 | Pan-cancer differences between primary and metastatic tumors. a)** Stacked plot representing the qualitative differences of the eight studied genomic features across the 23 cancer types included in this study. Cancer types are sorted in ascending order according to the cumulative number of diverging genomic features between primary and metastatic tumors. Each horizontal track represents a genomic feature. The presence (and height) of each feature for a specific cancer type correlates with the magnitude of the observed differences.

# Reporting Summary

## Statistics

For all statistical analyses, confirm that the following items are present in the figure legend, table legend, main text, or Methods section.

| n/a | Confirmed | |
|---|---|---|
| ☐ | ☒ | The exact sample size ($n$) for each experimental group/condition, given as a discrete number and unit of measurement |
| ☐ | ☒ | A statement on whether measurements were taken from distinct samples or whether the same sample was measured repeatedly |
| ☐ | ☒ | The statistical test(s) used AND whether they are one- or two-sided *Only common tests should be described solely by name; describe more complex techniques in the Methods section.* |
| ☒ | ☐ | A description of all covariates tested |
| ☐ | ☒ | A description of any assumptions or corrections, such as tests of normality and adjustment for multiple comparisons |
| ☐ | ☒ | A full description of the statistical parameters including central tendency (e.g. means) or other basic estimates (e.g. regression coefficient) AND variation (e.g. standard deviation) or associated estimates of uncertainty (e.g. confidence intervals) |
| ☐ | ☒ | For null hypothesis testing, the test statistic (e.g. $F$, $t$, $r$) with confidence intervals, effect sizes, degrees of freedom and $P$ value noted *Give P values as exact values whenever suitable.* |
| ☒ | ☐ | For Bayesian analysis, information on the choice of priors and Markov chain Monte Carlo settings |
| ☐ | ☒ | For hierarchical and complex designs, identification of the appropriate level for tests and full reporting of outcomes |
| ☐ | ☒ | Estimates of effect sizes (e.g. Cohen's $d$, Pearson's $r$), indicating how they were calculated |

*Our web collection on statistics for biologists contains articles on many of the points above.*

## Software and code

Policy information about availability of computer code

| | |
|---|---|
| Data collection | We have matched tumor-normal whole genome sequencing data from cancer patients from two independent cohorts: the Hartwig Medical Foundation (Hartwig) and the Pan-Cancer Analysis of Whole Genomes (PCAWG) cohort. |
| | The Hartwig cohort was provided under data transfer agreement (DR-247) by Hartwig Medical Foundation on 6 February 2020 with an update received on 4 Februari 2022, that was downloaded using via Hartwig download portal. |
| | The PCAWG-US was approved by National Institutes of Health (NIH) for the dataset General Research Use in The Cancer Genome Atlas (TCGA) on 25 February 2021 under application number 100344-3 and downloaded via dbGAP download portal. Raw sequencing access to the non-US PCAWG samples was granted via the Data Access Compliance Office (DACO) Application Number DACO-1050905 on 6 October 2017 and downloaded via https://console.cancercollaboratory.org on 4 December 2017. |
| Data analysis | The PCAWG samples were reanalyzed with the Hartwig somatic variant calling pipeline (https://github.com/hartwigmedical/pipeline5) which was hosted on the Google Cloud Platform using Platinum (https://github.com/hartwigmedical/platinum). This pipeline uses the following software packages: SamToFastq PICARD (v2.1.0). BWA (v0.7.17): read mapping GATK (v3.8.0) Haplotype Caller: calling germline variants in the reference sample SAGE (v2.2): somatic SMNVs and indels calling GRIDSS (v2.9.3): simple and complex structural variant calling AMBER (v3.3) COBALT (v1.7) |

PURPLE (v2.53): combines B-allele frequency (BAF) from AMBER (v3.3), read depth ratios from COBALT (v1.7), and structural variants from GRIDSS to estimate copy number profiles, variant allele frequency (VAF) and variant clonality. PURPLE also determines sample gender based on sex chromosome ploidy.
LINX (v1.16): interpretation of simple mutations and structural variants
mutSigExtractor (v1.23)
SigProfilerExtractor (v1.1.1)
SnpEff (v5.1)
dNdScv (v0.0.1)
ActiveDriverWGS (v1.1.2)
GISTIC2 (v2.0.23)
MutationTimeR (v0.1)

The source code to reproduce the analysis of the manuscript is available in this repository https://github.com/UMCUGenetics/primary-met-wgs-comparison: https://github.com/UMCUGenetics/PCAWG_Hartwig_comparison

For manuscripts utilizing custom algorithms or software that are central to the research but not yet described in published literature, software must be made available to editors and reviewers. We strongly encourage code deposition in a community repository (e.g. GitHub). See the Nature Portfolio guidelines for submitting code & software for further information.

# Data

Policy information about availability of data

All manuscripts must include a data availability statement. This statement should provide the following information, where applicable:
- Accession codes, unique identifiers, or web links for publicly available datasets
- A description of any restrictions on data availability
- For clinical datasets or third party data, please ensure that the statement adheres to our policy

Metastatic WGS data and metadata from the Hartwig Medical Foundation are freely available for academic use through standardized procedures. Request forms can be found at https://www.hartwigmedicalfoundation.nl/en/data/data-acces-request/

Somatic variant calls, gene driver lists, copy number profiles and other core data of the PCAWG cohort generated by the Hartwig analytical pipeline are available for download at https://dcc.icgc.org/releases/PCAWG/Hartwig. Researchers will need to apply to the ICGC data access compliance office (https://daco.icgc-argo.org) for the ICGC portion of the dataset. Similarly, users with authorized access can download the TCGA portion of the PCAWG dataset at https://icgc.bionimbus.org/files/5310a3ac-0344-458a-88ce-d55445540120. Additional information on accessing the data, including raw read files, can be found at https://docs.icgc.org/pcawg/data/.

GRCh37 reference genome used in this study: https://console.cloud.google.com/storage/browser/hmf-public/HMFtools-Resources/ref_genome/37

PCAWG clinical data: https://dcc.icgc.org/releases/PCAWG/clinical_and_histology
Consensus PCAWG mutation calls: https://dcc.icgc.org/releases/PCAWG/consensus_snv_indel
Consensus PCAWG mutation calls: https://dcc.icgc.org/releases/PCAWG/consensus_sv
Consensus PCAWG driver dataset: https://dcc.icgc.org/releases/PCAWG/driver_mutations

Independent validation datasets:
Primary and metastatic breast WGS data: DOI: 10.1016/j.ccell.2017.07.005 The link for the dataset is: http://dx.doi.org/10.17632/g7kpzkhz8c.1
Primary and metastatic kidney renal clear cell carcinoma WES data: 10.1038/ng.2891 Raw sequencing data was downloaded via: https://ega-archive.org/datasets/EGAD00001000734
Breast, Kidney, Prostate, Thyroid, Colorectal, Ovarian primary WES TCGA data:10.1038/ng.2764 downloaded via https://gdc.cancer.gov/about-data/publications/pancanatlas
Prostate metastatic WES data: 10.1073/pnas.1902651116 The link for the dataset is: https://github.com/cBioPortal/datahub/tree/master/public/prad_su2c_2019
Prostate metastatic WGS data: 10.1016/j.cell.2018.06.039 and downloaded via dbgap under phs001648.v2.p1
kidney renal clear cell carcinoma metastatic WES data: 10.1126/science.aan5951 and downloaded via https://www.cbioportal.org/

COSMIC reference signatures: https://cancer.sanger.ac.uk/signatures/
SIGNAL reference signatures: https://signal.mutationalsignatures.com/

OncoKB: https://www.oncokb.org/
CIViC: https://civicdb.org/
CGI: https://www.cancergenomeinterpreter.org/home

# Human research participants

Policy information about studies involving human research participants and Sex and Gender in Research.

| Reporting on sex and gender | Consistent gender proportions were observed across all cancer types except for thyroid adenocarcinomas, which had higher male representation in the metastatic cohort (metastatic: 72% male, 28% female; primary: 25% male, 75% female). |
|---|---|
| Population characteristics | The Hartwig cohort includes late-stage adult (>18 years old) cancer patients recruited across Dutch hospitals. Patients had frequently recived pre-biopsy treatment. The PCAWG cohort primarily include adult and early-stage cancer patients that in most cases have not recived any treatment prior to tumor biopsy. We refer to the Hartwig (doi: 10.1038/s41586-019-1689-y) |

and PCAWG (doi: 10.1038/s41586-020-1969-6.) flagship papers for further description of patient's population, recruitment and ethics oversight.

**Recruitment**

Patient recruitment was originally performed by the clinical institutions and hospitals. This study did not play any role in patient recruitment.

**Ethics oversight**

NA

Note that full information on the approval of the study protocol must also be provided in the manuscript.

# Field-specific reporting

Please select the one below that is the best fit for your research. If you are not sure, read the appropriate sections before making your selection.

☒ Life sciences ☐ Behavioural & social sciences ☐ Ecological, evolutionary & environmental sciences

For a reference copy of the document with all sections, see nature.com/documents/nr-reporting-summary-flat.pdf

# Life sciences study design

All studies must disclose on these points even when the disclosure is negative.

**Sample size**

We requested the data for all possible samples from the Hartwig and PCAWG cohorts. The Hartwig cohort included 4902 metastatic tumor samples from 4572 patients. The PCAWG cohort consisted of 2835 tumor samples from unique patients. After several filtering criteria we used 5,365 samples in the current study (1,914 from the PCAWG and 3,451 from the Hartwig cohort) to compare primary to metastatic tumors.

**Data exclusions**

A selection of samples for all analyses was made based on several criteria. To exclude duplicate samples from the same patient for the Hartwig cohort, we selected the tumor sample with the most recent biopsy date, and if this information did not exist we selected the sample with the highest tumor purity. However, some patients had biopsies from different primary tumor locations (likely independent or secondary tumors). In these cases, we kept at least one sample from each primary tumor location, and when there were multiple samples from the same primary tumor location, we applied the aforementioned biopsy date and tumor purity filtering criteria. For the PCAWG cohort, we processed one tumor sample per donor and tumor sample IDs are included in Supp. Table 1 of the manuscript. As with Hartwig QC filter criteria, samples with a tumor purity lower than 20% were removed as somatic variant calling was less reliable for these samples. PCAWG samples that were gray- or blacklisted by the PCAWG consortium were also removed (see https://dcc.icgc.org/releases/PCAWG/donors_and_biospecimens). For both cohorts, we only kept samples with >=50 SNVs/indels (likely no tumor cells present in the sample), and removed an additional set of samples for several reasons including failed variant calling, insufficient informed consent for use of the WGS data, unnatural SV landscape, and one duplicate PCAWG patient (DO217844) that was also included in the Hartwig cohort. After strict QC filtering, the PCAWG whitelisted cohort includes 2,376 samples and this dataset will be made available for the cancer research community via the PCAWG resource page. The metadata for every sample including those selected for analyses is detailed in supplementary table 1. Lastly, for this study, we only selected samples from cancer types with at least 15 samples that resulted in a final dataset consisting of 3,835 Hartwig samples and 1,916 PCAWG samples.

**Replication**

The source data and the source code used in this study are publicly available for academic purposes to ensure the reproducibility of the analysis conducted in this study

**Randomization**

Patients from both datasets (Hartwig and PCAWG) were independently recruited by clinical institutions and hospitals. Patients from the Hartwig Medical Foundation cohort represent late-stage cancer patients while PCAWG patients are primarily early-stage untreated cancer patients. This study did not play any role in patient's recruitment and randomization into experimental groups.

**Blinding**

This study did not play any role in patient's recruitment .

# Reporting for specific materials, systems and methods

We require information from authors about some types of materials, experimental systems and methods used in many studies. Here, indicate whether each material, system or method listed is relevant to your study. If you are not sure if a list item applies to your research, read the appropriate section before selecting a response.

## Materials & experimental systems

| n/a | Involved in the study |
|---|---|
| ☒ | Antibodies |
| ☒ | Eukaryotic cell lines |
| ☒ | Palaeontology and archaeology |
| ☒ | Animals and other organisms |
| ☒ | Clinical data |
| ☒ | Dual use research of concern |

## Methods

| n/a | Involved in the study |
|---|---|
| ☒ | ChIP-seq |
| ☒ | Flow cytometry |
| ☒ | MRI-based neuroimaging |

