## [Peer Review File · Nature]

Manuscript Title: Pan-cancer whole genome comparison of primary and metastatic solid tumors

Reviewer Comments & Author Rebuttals

Reviewer Reports on the Initial Version:

Referee expertise:

Referee #1: cancer genomics

Referee #2: cancer genomics and evolution

Referee #3: cancer genomics and evolution

Referees' comments:

Referee #1 (Remarks to the Author):

In this manuscript, Martinez-Jimenez and colleagues present a comparison of primary and metastatic cancer genomes. The study is a meta-analysis/re-analysis of PCAWG primary tumor and Hartwig cancer metastasis, very carefully conducted and controlled for differences between the cohorts (to the extent possible). The manuscript itself is very dense and written as a broad 'landscape' paper touching upon many aspects (clonality, genomic instability, tumor mutation and structural variant burden, driver landscapes, treatment-associated drivers, and clock-like mutational signature activities). It is a clear tour-de-force and both the combined uniformly reanalyzed data and the results are important resources for the community, which I'm sure will spark many dozens of follow-up studies. The main downside for me is that I find it very hard to distill this manuscript into a single punch-line novel result. Nevertheless, the manuscript solidifies multiple previously made claims (e.g. that metastases are more clonal than primary tumors; previously shown in Priestley et al, but now for the first time formally compared in a carefully controlled re-analysis) and contains several novel findings and analyses (of which the treatment-associated driver landscape is perhaps my favorite). Overall, the study is scientifically sound, notwithstanding some addressable weaknesses detailed below.

Major comments:

1. The increase in clock-like mutational signature SBS1 mutations in metastatic breast, prostate, clear cell kidney and thyroid cancers is interesting, and the authors' hypothesis that their originating cell lineages have low cell division rates and acquire higher turnover rates later in tumor evolution, is possible and supported by some (though circumstantial) data. However, this is definitely not the only possible explanation: (i) An independent, non-age-related mutational process could be operative in these tumors, with a similar profile to SBS1, such that what the authors read out as

SBS1 is a sum of both this novel signature and the clock-like SBS1. (ii) Breast, prostate, clear-cell kidney and thyroid cancers don't all metastasize under present treatment regimens, and (at least breast, prostate and thyroid cancers) have multiple subtypes; the metastases we see for these cancer types may be a select group of cases with higher proliferation rates. To get more insight into this, the authors should re-analyze paired primary/metastasis cases (at least for breast and prostate, these data exist in the literature) and compare their SBS1 mutation rates. This could potentially be done from existing mutation calls, as long as SBS1 mutations are reliably identified.

2. The authors identify increases in the frequencies of gain events, mostly in clear cell kidney, prostate and thyroid cancers. However, these tumor types also have higher frequencies of WGD. I would hypothesize this signal mainly originates from WGD events, rather than gains of multiple individual gains. A more carefully controlled analysis (e.g. in allele-specific copy number space, and defining gains and losses relative to 1+1 (non-WGD tumors) vs. 2+2 (WGD tumors) copy number states) could clarify this. It would also be worthwhile to perform this analysis for WGD and non-WGD tumors separately.

3. The analyses in Figures 1e, 2a, 4a and 5a contain a substantial number of statistical tests but there doesn't appear to be any correction for multiple testing. It would be worth running a correction on these tests as has been done for other analyses.

Minor comments:

1. The authors mention a few times they use paired samples for analysis, and while they do use tumor/normal pairs in both PCAWG and Hartwig, there are no primary/metastasis pairs included. While this caveat of the study is clearly mentioned in the discussion, it would be helpful to readers not familiar with the data if the authors would make this point abundantly clear in the introduction and results as well.

2. In the analysis of early clonal mutation rates of SBS1 in Fig. 3c/d, it is unclear if SNVs from all regions are included, or only gained regions. If so, some of the increase in the clonal rate ratios may be reflective of more copy number gains in metastases than primaries. It would likely be best to subset this analysis to 2+0 and 2+2 regions, for which all mutations can be classified as either clonal early or clonal late (i.e. there would be no clonal unassigned mutations).

3. In the discussion, what does the statement 'metastatic tumors share [...] fewer short mutations than primary tumors' mean? It is mentioned in an earlier section that 'TMB in metastatic tumors was only moderately increased compared with primary tumors'.

4. Minor figure comments:

a. Supplementary Figure 3a-d misses a color legend (blue=metastasis; red is primary); also typo: association -> association

b. The Y-axis labels in Supplementary Figure 4a are misaligned

Referee #2 (Remarks to the Author):

The authors compare the genomic features of unpaired metastatic and primary cancer tissue biopsies from two published retrospective cohorts (Hartwig–metastatic only; and PCAWG/TCGA–primary only). They report that five cancer types (including prostate, breast, and kidney cancer) show clear differences between localized and metastatic disease while most other cancer types (e.g. ovarian) show relatively few differences. In addition, the authors find evidence for genomic impact

of prior treatment, including apparent evolutionary bottlenecks. The authors conclude that 1) they have observed evidence of evolution between the localized and metastatic niche (in breast cancer, prostate cancer, and others), and that 2) whole-genome sequencing of cancer tissue is a valuable tool to understand tumor evolution. Unfortunately, the first conclusion is not adequately supported by the data supplied (details below), while the second conclusion has modest novelty value.

To their credit, the authors analyse an impressive number of samples – clearly requiring a significant investment in time and expertise. However, the major limitation lies in the study design for evaluating evolution – namely the lack of patient-matched primary-metastatic samples. Coupled with some cohort-level heterogeneity, these factors massively constrain the extent to which this dataset can provide new biological or clinical insight (about evolution), and consequently most of the findings presented are low resolution or largely just recapitulate known cancer-specific observations. I am also concerned about the highly comprehensive data dredging – although this type of approach is typical for many pan-cancer “landscape” studies, it makes interpretation of result significance to be very challenging. And of course, we do not see which data tests did not make the final manuscript. Even after incorporating multiple test corrections for some subanalyses, the scale of testing means that many associations run a significant risk of being spurious.

To flesh out the comment above regarding lack of patient-matched samples, the fundamental issue is that several of the highlighted cancers with genomic differences between primary and metastatic tissue (e.g. prostate, breast, kidney) can have limited metastatic potential. For example, it is well-established that most localized prostate cancers are relatively indolent and may not require radical interventions. However, surgery/radiation therapy is often performed due to inadequate risk stratification models (to accurately resolve aggressive from indolent). The same is true (albeit to varying extent) in kidney, breast cancer, thyroid, pancreatic neuroendocrine. Therefore, to study primary-metastatic evolution in the absence of patient matched primary and metastatic samples, it is necessary to enrich the primary datasets with samples from patients that subsequently (or in parallel) developed metastatic disease. In prostate, this can be only ~10% of patients that present with local cancer, in pancreatic neuroendocrine and thyroid cancer, it is even fewer. Whereas in prostate it is known that localized and metastatic cancers have distinct driver landscapes (e.g. Armenia et al Nature Genetics 2018), it is also known that primary tissue samples taken from patients with metastatic disease show fairly similar driver gene profiles (e.g. Mateo et al. JCI 2020).
Additional major comments

1. Samples are processed with different pre-analytical laboratory workups and sequencing parameters, and notably there is a large imbalance in sample availability between comparator groups (1916 primary versus 3835 metastatic samples).

2. Metastatic and primary biopsies are profiled at different sequencing depths of (109x for metastatic versus only 38x or 60x for primary). Low primary sample depth is liable to constrain detection sensitivity across the spectrum of genomic events (especially copy alterations and ploidy). The distribution of depth (38x vs. 60x for primary samples) is also somewhat cancer specific, e.g. bladder cancer is almost exclusively 38x, whereas esophageal cancer is almost exclusively 60x. The authors perform some benchmarking to rule out this large discrepancy in sequencing depth as a confounder in somatic variant detection rates, since the higher sequencing depth in metastatic

samples could create an 'illusion' of enrichment for SNVs / SVs due to improved depth-related detection sensitivity. However I'm concerned that this confounder is not possible to completely mitigate.

3. Figure 1d: the conclusions of substantial karyotype changes in specific cancer types may benefit from some clarification. The distribution of arm level gains/losses in primary vs. metastatic kidney clear cell cancer and prostate cancer in fact appears quite similar (i.e. most arms are copy-altered in the same 'direction'). In prostate cancer, 8p loss / 8q gain is an established early event and is therefore not expected to be different across primary and metastatic disease (contrary to what the $Q\text{-value} < 0.05$ may indicate). Are there other technical or analytical reasons that may partly explain some of these differences? For example, can the authors rule out systemic differences in primary vs. metastatic sample tumor purity (e.g. lower primary purity reducing CNV detection sensitivity compared to metastatic disease)?

4. It would be helpful to better understand how the variable of differential whole-genome doubling (WGD) status impacts the normalisation of CNVs relative to mean ploidy. For example, haploid LOH in a diploid tumor will have half the absolute difference (relative to mean ploidy) of the same biological event in a WGD tumor. Does comparing average absolute CN difference account for the apparent "enrichment" of gains and losses in some of the metastatic cases (which are overall slightly enriched for WGD)? Are there better ways the authors can control for variable WGD status (or indicate that this does not underlie some of these potential differences)?

5. Relatedly, the methods text for "Karyotype" determination need substantial development. It is unclear how genome-wide CN information was derived (and related metrics including arm-level CN changes and mean ploidy).

6. De novo signature analyses: The observation that DBD_denovo_clust_1 and DBS_denovo_clust_2 are associated with POLE mutations and MSI (in proposing potential etiologies) suggests that these 'de novo' signatures could be plausible matrix factorization artefacts, given that POLE mutations and MSI also causally underlie established COSMIC DBS signatures 3 (POLE) and 7/10 (mismatch repair deficiency). What are the cosine similarities of these de novo DBS signatures to their established COSMIC signature counterparts? Are their assigned per-sample signature weights correlated (i.e. de novo versus COSMIC DBS)? Do samples with high DBS_denovo_clust_2 (i.e. MMRd) weights also have high COSMIC SBS 6/15 weights? It would be helpful to produce DBS context plots for all de novo signatures generated so readers can intuitively compare to previously catalogued DBS signatures. Overall some deeper analysis will need to be performed in order to claim DBD_denovo_clust_1 and DBS_denovo_clust_2 as 'novel' signatures.

7. Figure 2: Comparing median differences in mutation/indel count (stratified by signature assignment) is confounded by broader differences in mutation rates (between cancer types and between primary vs. metastatic tumors overall). It would be easier to interpret these plots (and all Results text) if instead comparing median signature proportions. Do the proposed etiological differences between primary and metastatic disease still hold after normalising for differences in mutation counts? It is very difficult to evaluate the soundness of most signature-related claims while the results are in units of mutation count. The entire Results section "Differential SBS1 mutation

rates in primary and metastatic cancers” needs to be reanalyzed in the unit of signature proportions.

8. The authors should adjust some of their language describing treatment-associated enrichment of mutation signatures, since these are correlative associations (i.e. not causal). E.g. “... DBS2 occasionally led...” etc.

9. Results text: “In fact, the excess in DBS mutation burden observed in six cancer types (stomach, esophagus, cervix, upper respiratory tract, non-small cell lung, and urothelial cancer) could be fully attributed to platinum treatment (Fig. 2d top bars).” Can the authors clarify what they mean by “fully attributed”? Had the metastatic cohort patients included in these analyses received confirmed cytotoxic or platinum treatment prior to metastatic biopsy (and conversely, were the primary biopsies treatment-naive)?

10. Results text: “The systematic enrichment of SBS2/13 mutations in metastatic cancers suggests enhanced activity of APOBEC mutagenesis during the progression of advanced tumors” – the enrichment of APOBEC related mutational processes appears largely driven by only a few cancer types (notably breast and stomach cancer), so it feels incorrect to claim ‘systemic enrichment’ across metastatic cancers.

11. Results text: “... the SV analyses revealed a much more widespread effect, with larger increases per metastatic cancer type that affected almost every cancer type studied, indicating that metastatic tumors appear to evolve primarily by genomic changes at the structural level.” – this conclusion appears to contradict the earlier result that copy-number landscapes appear relatively unchanged between primary and metastatic disease (SVs and CNVs can often be reflective of the same underlying process). Can the authors clarify or offer some deeper analysis (perhaps of the CNV landscape beyond broad arm-level changes)?

12. I am worried that attempting to connect SBS1 mutation counts and “yearly mutation accumulation” and turnover rates is potentially tenuous. Metrics using clonality are very challenging because the primary samples are low sequencing depth (2-3 fold lower than compared to metastatic samples) and I suspect this systematically biases some of the results. Copy number analysis is also crude (e.g. insufficient controlling for WGD, impact of variable purity on calling of copy changes) further roughening all clonality measurements. The Figure title, “Cell cycle division rates in primary and metastatic tumors” is inappropriate: this has not been directly tested and is only surmised through several layers of simplifying assumptions.

13. The “long tail of candidate drivers of resistance” is not really appropriate, given lack of paired (patient-matched) samples. With the confounding dataset issues raised in my introductory comments, the dataset probably too crude to pick up meaningful associations beyond established common resistance alterations (like ESR1, PIK3CA, AR, etc.).

14. Code availability statement: can the authors please provide their full documented source code so this can be reviewed?

Minor:

- Figure 1a: The Biorender patient schematic is fairly low resolution, you may wish to adjust this.
- Supplementary Note: Similarly, many of the Figures are very low resolution and are difficult to scrutinise for details (e.g. Supp Note 1 Figure 5).
- Figure 5a: It looks like the figure did not render properly or was excessively rasterized. There is a stray vertical line abutting the leftmost column.

Referee #3 (Remarks to the Author):

Martínez-Jiménez characterize the genomic differences between early-stage untreated primary tumors and late-stage treated metastatic tumors using a harmonized pan-cancer re-analysis of 7,152 whole-genome sequenced tumors. The authors reveal notable differences between primary and metastatic tumors in this unpaired analysis.

This is an important area of research, exploring an impressive data-set. On the whole, the manuscript is clear (with a few exceptions, see below). One key limitation is the fact that the analysis is unpaired. While I appreciate this is nevertheless an impressive dataset, I would urge the authors to consider whether they can control for some of the biases introduced by virtue of the types of comparison. For instance, it would be informative to consider two types of untreated primary tumors - those that subsequently metastasize and those that never metastasize. And, for many of the analyses, there is the possibility that subtypes are heavily influencing the results if not controlled for. Likewise, all mets of a given tumor type are considered equally (for the most part) - an analysis of how sites of met influences the genomic landscape would be very informative.

There is no doubt this a valuable resource that can be leveraged for further study. However, beyond the resource aspect, there is somewhat limited truly novel findings. And, there is a lack of validation for key results.

I have the following specific comments:

- Sampling bias and sampling technique. are any tumors subject to multi-region sequencing? While the clonality appears different between primary and metastatic tumors, is there a possibility that this simply reflects size of biopsy? Or other technical factors? For instance, it could be that additional samples would reveal extensive diversity between metastatic regions from the same metastasis, and there is less intermingling of subclones? I think these caveats need to be considered, even if it is not feasible to perform additional sampling.

- lymph node may be local or not? what is the distinction here?

- "loss of heterozygosity (LOH) genome fraction in diploid tumors" - why only in diploid tumours?

- many of the primary tumors subsequently metastasize, while others do not. It would considerably strengthen the analysis if the authors could distinguish these two groups. This would be particularly relevant for much of the mutational signature analysis

- The analysis suggests 'enhanced activity of APOBEC mutagenesis during the progression of advanced tumors'. It would be interesting to explore whether this reflects met-specific mutations or the tumor as a whole. In other words, there are two possible scenarios here, APOBEC tends to be dominant after the metastatic transition, or, alternatively, APOBEC is more active in tumors that metastasize. While evaluation of met-specific mutations would require paired primary-met analysis, the authors could consider comparing mutations during different epochs of tumour development. For instance, if the authors explore early pre-WGD mutations in primary and met tumours are they more similar, compared to post-WGD mutations?

- Do the authors observe SBS1 mutation burden enrichment regardless of the site of metastasis? In general, the authors appear to lump all mets together based on the site of the primary tumour rather than the location of the metastasis. An analysis exploring how the site of metastasis influences the genomic landscape would be interesting.

-Differential SBS1 mutation rates in primary and metastatic cancers - this seems rather speculative, further orthogonal validation would be beneficial. Indeed, this whole section, while interesting, seems rather preliminary. The authors must ensure there are not additional factors influencing the results - e.g. tumor purity, sequencing coverage etc. And, again, is this an intrinsic feature of metastasis or simply aggressive tumors? Likewise, the differences observed in breast cancer could simply reflect a difference in subtype between primary and met (e.g. ER+ vs. triple negative).

-SBS and DBS signatures are compared to signatures on COSMIC and two novel DBS signatures are found - have these signatures been reported in Degasperi 2022 before?

-'Compared with TMB, the SV analyses revealed a much more widespread effect, with larger increases per metastatic cancer type that affected almost every cancer type studied, indicating that metastatic tumors appear to evolve primarily by genomic changes at the structural level.' We can't be certain that tumors evolve in this way because primary and met samples are not paired. The primary samples which subsequently went on to metastasise could have already presented more structural variants. As stated earlier, an analysis which considers primary tumors which certainly do not metastasize may help shed light on the extent to which this is likely true.

- Figure 2c-e: Where there is an enrichment of a signature in primary/metastasis it would be good to see the proportions of signatures in primary/metastasis respectively to be able to tell whether one signature is enriched in say the primary cohort which other signature becomes prevalent in the metastasis samples

- The analysis of treatment related signatures and treatment related mutations requires further work. For instance, Figure 6D compares the mutation distribution in EGFR in those tumours receiving treatment for Anti-EGFR and those untreated. Is this a meaningful comparison? Those receiving treatment for anti-EGFR will necessarily harbor activating EGFR mutations or else they would not be eligible for treatment. Thus, what is the main message here? It ends up being a somewhat circular argument. While this analysis makes sense in the context of pre-treatment and exploration of treatment resistance, I don't believe it does without further exploration. Thus, the authors should

seek to distinguish between TEDs which are treatment induces (or selected as a result of treatment) and those which are simply the targets for treatment.

-Related to the above, there is a need to ensure the analysis is not influenced by cancer subtypes. E.g. anti-EGFR therapy will be given in the context of LUAD, not LUSC. Thus simply considering NSCLC may confound the results and interpretation.

-Figure 4 - I find this a rather confusing figure (in particular b-f). The colors of the labels overlap with the colors in the enrichment scale. And, I assume the 'Association with higher SV burden arrow' is a typo? In this case further separation of tumour subtypes would also be useful.

-Figure 5. For driver comparison it would be useful to ensure the differences do not simply reflect higher number of mutations; i.e. increased likelihood of observing a driver alteration? Again, separating different subtypes would be useful here. For instance, the colorectal samples have a higher mutation burden in Primary tumors. This likely reflects an enrichment for MSI tumors which don't metastasize. This would be worth evaluating.

-Figure 7. While I think it would be useful to have a summary Figure, I don't find this figure very helpful at all. I find it hard to compare cancer types, and distinguishing strong and moderate is not easy.

- Evaluating the correlation in purity between PCAWG and Hartwig it is notable that there are a number of samples with 100% purity. Can the authors confirm that the VAF of mutations in these samples is consistent with this estimation?

-it is only mentioned in the discussion that primary and met samples are not paired, while it is also not stated otherwise before this might lead to misinterpretation

Author Rebuttals to Initial Comments:

Point-by-point response to reviewer comments

Color text code:

Response to reviewers

Changes in the main text

Referee #1 (Remarks to the Author):

In this manuscript, Martinez-Jimenez and colleagues present a comparison of primary and metastatic cancer genomes. The study is a meta-analysis/re-analysis of PCAWG primary tumor and Hartwig cancer metastasis, very carefully conducted and controlled for differences between the cohorts (to the extent possible). The manuscript itself is very dense and written as a broad 'landscape' paper touching upon many aspects (clonality, genomic instability, tumor mutation and structural variant burden, driver landscapes, treatment-associated drivers, and clock-like mutational signature activities). It is a clear tour-de-force and both the combined uniformly reanalyzed data and the results are important resources for the community, which I'm sure will spark many dozens of follow-up studies. The main downside for me is that I find it very hard to distill this manuscript into a single punch-line novel result. Nevertheless, the manuscript solidifies multiple previously made claims (e.g. that

metastases are more clonal than primary tumors; previously shown in Priestley et al, but now for the first time formally compared in a carefully controlled re-analysis) and contains several novel findings and analyses (of which the treatment-associated driver landscape is perhaps my favorite). Overall, the study is scientifically sound, notwithstanding some addressable weaknesses detailed below.

We are very glad that the reviewer appreciates the relevance of our work.

Major comments:

1. The increase in clock-like mutational signature SBS1 mutations in metastatic breast, prostate, clear cell kidney and thyroid cancers is interesting, and the authors' hypothesis that their originating

cell lineages have low cell division rates and acquire higher turnover rates later in tumor evolution, is possible and supported by some (though circumstantial) data. However, this is definitely not the only possible explanation: (i) An independent, non-age-related mutational process could be operative in these tumors, with a similar profile to SBS1, such that what the authors read out as SBS1 is a sum of both this novel signature and the clock-like SBS1.

We agree with the reviewer that it is possible that the observed increase in SBS1 may be caused by other factors, including an increase from a different mutational signature with shared mutational contexts. In fact, in the original manuscript we tried to minimize this factor by restricting the definition of SBS1 mutation burden as the sum of mutations overlapping the XCpG>T contexts and not the mutational signature exposure, because the latter may include low contribution from other channels beyond CpG that could result in “mutational signature leaking”. Moreover, we explicitly excluded mutations with TCpG>T mutational context due to the overlap with the APOBEC SBS2 mutational signature (which also showed a tendency towards enrichment in metastatic tumors).

If the SBS1 increase was still explained by a different mutational process/es, its mutational profile/s should **i)** display high resemblance to SBS1 **ii)** display activity in the four cancer aforementioned types and **iii)** be enriched in metastatic tumors on these cancer types.

We did not find any evidence supporting these three points. In fact, none of the non-SBS1 attributed mutational signatures displayed high resemblance to SBS1 as illustrated by their low cosine similarity (see table below, Supp. Table 3 and Supp. Data 1 for full information about SBS1 mutational signature pairwise similarity).

assigned_sig_name	denovo_sig_name	cos_sim.SBS1
SBS_denovo_1	Kidney.SBS_C	0.4744711
SBS5	Prostate.SBS_C	0.4255138
SBS44	Prostate.SBS_A	0.4054119
SBS44	Breast.SBS_J	0.3983211
SBS_denovo_11	Prostate.SBS_J	0.3100743
SBS5	Breast.SBS_G	0.2937173
SBS5	Thyroid.SBS_B	0.2930252
SBS_denovo_14	Thyroid.SBS_G	0.2274541
SBS26	Prostate.SBS_G	0.1626004
SBS_denovo_10	Prostate.SBS_H	0.1578151

This suggests that a novel metastatic-enriched signature is likely not the main underlying cause of the observed enrichment. However, we acknowledge this possibility in the manuscript text:

Nevertheless, we can not entirely rule out the contribution of other tissue type specific underlying factors such as higher rates of 5-methylcytosine deamination, decreased fidelity to repair these mismatches or higher contribution from other metastatic specific mutational processes with overlapping mutational contexts.

(ii) Breast, prostate clear-cell kidney and thyroid cancers don't all metastasize under present treatment regimens, and (at least breast, prostate and thyroid cancers) have multiple subtypes; the metastases we see for these cancer types may be a select group of cases with higher proliferation rates. To get more insight into this, the authors should re-analyze paired primary/metastasis cases (at least for breast and prostate, these data exist in the literature) and compare their SBS1 mutation rates. This could potentially be done from existing mutation calls, as long as SBS1 mutations are reliably identified.

As we believe this is a highly relevant point, we have now performed an extensive orthogonal validation of our observations regarding the age-independent SBS1 mutation increase in these cancer types (comprehensively described in newly created Supp. Note 3).

Specifically we have complemented our current analysis and controls by performing:

1. An independent validation of the age-independent SBS1 mutation rate increase in metastatic tumors using two datasets of **paired primary-metastatic biopsies of breast and kidney renal clear cell carcinomas**.
2. A validation of our observations using multiple orthogonal datasets of **unpaired primary and metastatic tumors for the cancer types of interest**. Specifically, we have included positive controls from breast, prostate, kidney renal clear cell and thyroid carcinomas as well as negative controls for colorectal and for ovarian serous carcinomas.
3. An analysis of **SBS1 mutation rate across breast cancer subtypes** in our primary and metastatic cohorts.
4. An analysis of **SBS1 mutation rate in prostate cancer metastatic biopsies from PCAWG** prostate cancer cohort.
5. An analysis of **SBS1 mutation rate enrichment by metastatic biopsy location** in the cancer types of interest.

Overall, the results from all aforementioned analyses strongly support our initial observation of an age-independent increase in SBS1 mutation rate in these four metastatic cancer types that is not observed in other cancer types. These results also complement the comprehensive controls previously performed in our cohort (e.g., ploidy correction, timing and clonality of SBS1 mutations, etc).

Updated text:

[...]However, four cancer types (i.e., breast, prostate, kidney renal clear cell and thyroid carcinomas) showed an age-independent and significant enrichment of SBS1 mutations at the metastatic stage (Fig. 3a, Extended Data Fig. 4a). For instance, metastatic breast cancer had a nearly uniform fold increase of 1.46 (188 ± 17 SBS1 mutations, mean \pm SD) across the ages of biopsies that was generally consistent across breast cancer subtypes (Extended Data Fig. 4b and Supp. Note 3). Importantly, this pattern was highly cancer type specific and was not observed for most cancer types, including those with similar intra-tumor heterogeneity in the primary cohort (e.g., colorectal, ovarian serous, pancreas and stomach carcinomas) (Fig. 3b, Extended Data Fig. 4a). Moreover, this pattern was not explained by differences in tumor genome ploidy (Extended Data Fig. 4c), by metastatic biopsy location (Supp. Note 3), was observed in paired primary-metastatic biopsies from breast and kidney renal clear cell carcinomas and rendered consistent patterns when relying on independent unpaired cohorts (see Supp. Note 3).

Since this phenomenon was not observed for cell-division independent clock-like mutational processes (i.e., SBS5/SBS40), we believe that it can be tightly associated with an increased cell division rate of metastatic tumors, given its known association with SBS1 mutation rates (see Alexandrov et al. 2015). Such an increase would be particularly relevant for slowly dividing primary tumors (e.g., prostate, thyroid, kidney renal clear cell and breast carcinomas) where an increased cell division rate would lead to higher pace of clock-like SBS1 mutation accumulation. Nevertheless, as mentioned by the reviewer, we can not entirely rule out that this increase is caused by other tissue specific factors such as higher rates of 5-methylcytosine deamination, decreased fidelity to repair these mismatches or higher contribution from other metastatic specific mutational processes with overlapping mutational contexts. As mentioned in the previous response, we now acknowledge alternative plausible explanations in the main text.

Updated text:

Nevertheless, we cannot entirely rule out the contribution of other tissue specific underlying factors such as higher rates of 5-methylcytosine deamination, decreased fidelity to repair these mismatches or higher contribution from other metastatic specific mutational processes with overlapping mutational contexts.

Please, see below and Supp. Note 3 for full description of the SBS1 age corrected validation results, methods and discussion of the implications.

1. Independent validation in using paired biopsies

We gathered two cohorts of primary and metastatic paired biopsies in breast (Lucy R Yates et al. 2017 Cancer Cell) and renal clear cell carcinomas (TRACERx renal, Marco Gerlinger et al. 2012 NEJM). SBS1 mutation rate was substantially higher in metastatic tumors compared to primary biopsies. Moreover, in all except for one patient we observed an increased SBS1 mutation rate in the metastatic biopsies compared to primaries (see Supp. Note 3 for further details about these analyses).

2. Independent validation in external (unpaired) primary and metastatic datasets

We also gathered multiple independent primary and metastatic WGS/WES cohorts for the six cancer types of interest (i.e., breast, prostate, kidney clear cell and thyroid carcinoma and the two controls, ovarian serous and colorectal carcinomas).

In all cases the observed SBS1-age regressions were consistent with the reported patterns in our dataset (see charts below). This is particularly relevant considering the considerable heterogeneity of the somatic mutation calling strategies used by each of the independent studies.

Breast carcinoma
(Primary=TCGA, Metastatic=Hartwig (WES))

Prostate carcinoma
David Quigley et al. Metastatic WGS dataset (corrected)

Prostate carcinoma
(Primary=TCGA, Metastatic=SU2C-PCF)

Kidney clear cell carcinoma
(Primary=TCGA, Metastatic=Diana Miao et al.)

Thyroid carcinoma
(Primary=TCGA, Metastatic=Hartwig (WES))

Colorectal carcinoma (Primary=TCGA, Metastatic=Hartwig (WES)) Ovarian serous adenocarcinoma (Primary=TCGA, Metastatic=Hartwig (WES))

3. Breast cancer subtype analysis

We annotated breast cancer subtypes (see methods of Supp. Note 2) for primary and metastatic samples and performed SBS1-age regression comparison for each breast cancer subtype.

SBS1 age association (breast cancer subtype)

Our results revealed that three breast subtypes (i.e., ER-positive/HER2-negative, ER-negative/HER2-positive and triple negative) showed an age-corrected SBS1 mutation burden increase (see above, now part of Supp. Fig 3b). The only exception was ER-positive/HER2-positive breast cancer which did not show a clear enrichment in metastatic tumors, although given the low number of samples and their narrow age-distribution we cannot formally exclude insufficient statistical power for this subtype.

4. SBS1 mutation burden in metastatic prostate samples from the PCAWG cohort.

We realized that there were a few metastatic prostate biopsies in the PCAWG cohort (see Supp. Table 1). Our data showed that the five metastatic prostate cancer biopsies included in the PCAWG cohort were perfectly aligned with the expected number of SBS1 mutations given by the metastatic regression fitted with Hartwig metastatic samples (see above and Supp. Note 3). This result suggests that the SBS1 mutation burden differences do not seem to be driven by differences in sequencing protocols and that are genuinely attributed to different mutation rates between primary and metastatic tumors.

5. SBS1 mutation rate enrichment by metastatic biopsy location

We also checked whether metastatic biopsy location may influence the increase in SBS1 mutation burden across the aforementioned cancer types of interest. Overall, the four cancer types showed a trend towards age-independent SBS1 mutation burden increase across the three types of metastatic location. Moreover, we did not observe any difference across primary-met age-corrected SBS1 burden in the control groups.

Distant met biopsy

Lymph node biopsy

Local biopsy

2. The authors identify increases in the frequencies of gain events, mostly in clear cell kidney, prostate and thyroid cancers. However, these tumor types also have higher frequencies of WGD. I would hypothesize this signal mainly originates from WGD events, rather than gains of multiple individual gains. A more carefully controlled analysis (e.g. in allele-specific copy number space, and defining gains and losses relative to 1+1 (non-WGD tumors) vs. 2+2 (WGD tumors) copy number states) could clarify this. It would also be worthwhile to perform this analysis for WGD and non-WGD tumors separately.

We agree with the reviewers (see also comment 3 and 4 from reviewer #2) that the karyotype analysis can benefit from more clarification and improved methodology. Therefore, we revised the karyotype approach with the following adjustments:

- For the karyotype analysis, we already used an approach that corrects for single (or multiple) whole genome duplication events where we scored the chromosomal gains and losses relative

to the average genome ploidy of the tumor sample. However, we found that many samples had an average genome ploidy ~ 3 , likely because WGD samples typically undergo more chromosomal losses than gains (see Taylor et al 2018). Consequently, a 4n ploidy chromosomal arm in a WGD sample with an average genome ploidy level of 3 is scored as a gain, whereas a 3n ploidy arm would have scored as a copy number neutral event, which is mechanistically not correct. Moreover, by relying on the absolute copy number values, chromosomal losses in WGD samples would have higher overall weight than in diploid samples (as pointed out by the reviewer). To mitigate this, we now compare each arm against the expected 2n ploidy stage (expected copy number in diploid=2, after whole genome doubling=4, after two whole genome doubling=8) and not against the average genome ploidy. The WGD status was taken from PURPLE and WGD samples with an average genome ploidy of 6 or more were considered as cancers with 2 whole genome doublings (see methods).

- We now perform a normalized scoring of arm gains and losses (i.e., -1: chr. arm loss; 0 neutral and 1 chr. arm gain). This is different than in the first version of the manuscript where we rely on absolute ploidy levels (e.g., arms could have ploidy levels of 20 or higher). We believe this adjustment simplifies the interpretation of the karyotype analysis and this adjustment also mitigates the statistical bias introduced by chromosome arms with high ploidy levels. We defined a chromosomal arm gain and loss for those cases where the chromosomal arm ploidy level is, at least, one ploidy higher or lower than the expected 2n ploidy stage, respectively.
- Finally, we now include a more strict criteria for calling karyotypic changes between primary and metastatic tumors by including a minimum effect size of 0.25 combined with the significance given by the corrected p-value (i.e., minimum average copy number change between primary and metastatic tumors of 0.25 and q-value < 0.05). After applying this criteria, we found that 43 of the 47 (91%) significant hits are observed in kidney renal clear cell, prostate and thyroid cancers indicating that the metastatic tumors of these three cancer types undergo substantial karyotypic changes.
- Of note, we have also revised the LOH score, now extended also to non-diploid tumor samples (see comment by reviewer #3) where we assess the minor allele copy number that is calculated based on the observed BAF and depth ratio within PURPLE. Thus, genomic regions with minor allele copy number of ~ 0 and major copy number > 0 are defined as LOH (exact definition minor allele copy number of < 0.25 and major allele copy number of ≥ 0.8). The LOH score of a given sample represents the sum of all LOH regions divided by GRCh37 total genome length.
- We also assessed major and minor allele segments to generate allele specific karyotype profiles to control for WGD-driven arm alterations.

After applying these adjustments our results revealed the following karyotype changes:

- **Better representation of copy number neutral status** in primary prostate and thyroid carcinoma samples of which most samples are fully diploid.
- **Improved representation of high fraction of single arm losses in pancreas neuroendocrine (PANET).** This cancer type includes patients bearing recurrent patterns of whole chromosomal loss (RPCL) across several chromosomes (see <https://doi.org/10.1038/nature21063>).

- The more strict statistical criteria **reduced the statistical arm alteration hits** between primary and metastatic from 85 to 47 (see below).
- The majority (91%, 43 of 47) of **karyotypic changes now involve three cancer types**: renal clear cell carcinoma, prostate carcinoma and thyroid carcinomas. Importantly, these changes were not exclusively associated with higher WGD metastatic rates (see explanation below).
- Finally, all but one (the well-known bad prognosis factor of 8q gain in prostate carcinoma) significant karyotypic changes are associated **with increased frequency of copy number losses in the metastatic setting**. This pattern is maintained when subsampling for non-WGD tumors (see below).

Updated Fig.1d and Fig.1e with the normalized estimations:

Concerning the impact of different WGD rates on the comparison, the reviewer is indeed correct that cancers with whole genome duplication events are associated with a higher incidence of arm-level alterations. This is in line with our findings where we overall observe higher aneuploidy scores for WGD samples (mean aneuploidy scores higher than 20) than for non-WGD samples (mean aneuploidy scores lower than 20) (Extended Data. Fig 1c-d, see below). As previously shown in <https://doi.org/10.1016/j.ccell.2018.03.007>, this indicates that WGD event(s) accelerates aneuploidy, and are thus an important marker for a higher incidence of chromosomal gains and losses.

However, it is still important to mention that diploid (non-WGD) prostate, renal clear cell and thyroid carcinoma tumors still showed important gains and losses in the metastatic cohort, although the statistical power is lower likely because of lower sample size (see figure below now included in Extended Data Fig. 1c-d). Furthermore, these tumor types consistently showed a significant aneuploidy score increase when subsetting for non-WGD tumors. Finally, we noticed that non-WGD metastatic thyroid carcinoma showed a higher number of arm losses, which may be closely related to

the increased LOH rates. Altogether, our data suggests that WGD is not the only marker for karyotypic changes in metastatic tumors and that other genomic instability indicators, such as TP53 mutation status or RPCL, may also play an important role.

Karyotype comparison in tumor samples that did not undergo whole genome duplication (Extended Data Fig. 1c):

Tumor karyotype in non-WGD tumors

Karyotype comparison in tumor samples that underwent whole genome duplication (Extended Data Fig. 1d):

d Tumor karyotype in WGD tumors

Finally, to fully mitigate the effect of WGD-driven copy number alterations, we conducted an allele-specific karyotype analysis in which a gain/loss event is only scored when there is an imbalance in major and minor allele copy number ploidy. This WGD controlled karyotype analysis resulted in 48 significant arm alterations which is nearly identical to the 47 significant arm alterations when not correcting for allele specific changes (see figure below). Since we aim to showcase the global karyotypic changes per cancer type independent of underlying cause (WGD-driven, RPCL-driven, chromosome segregation errors, DNR repair, among others) or evolutionary process we kept our analysis based on normalized ploidy estimates in the manuscript.

We have updated the Fig. 1d and 1e with the aforementioned changes, we also now include the non-WGD and WGD-only in comparison in Extended Data Fig. 1c and 1d. Finally, we have updated the manuscript results, methods and discussion accordingly.

Karyotype:

Comparison of chromosome arm aneuploidy profiles revealed a generally conserved portrait, which was strongly shaped by the tissue of origin (Fig. 1d, Supp. Table 2), supporting the notion that tumor karyotype is generally defined at early stages of tumor evolution²⁹. Only metastatic kidney renal clear cell, prostate and thyroid carcinomas showed substantial changes compared to primary, encompassing 91% (43 of 47) of all significant discrepancies. Besides the poor prognostic marker 8q gain in metastatic prostate carcinoma³⁰, all discrepancies were associated with an increased prevalence of chromosomal arm losses at the metastatic setting. Remarkably, 30% (14 of 46) of the

metastatic-enriched chromosome arm losses were retained when comparing non-WGD tumors (Extended Data Fig. 1c-d), indicating that other factors, aside whole genome doubling^{31,32}, play an important role in the accumulation of arm-sized chromosomal losses.

Genomic instability indicators:

The same three cancer types also showed persistent increases in four genomic instability indicators (*i.e.*, chromosomal aneuploidy score³¹, loss of heterozygosity (LOH) genome fraction, WGD³³, and *TP53* alterations^{33,34}) in the metastatic cohort (Fig. 1e, Supp. Table 2). While these indicators showed overall higher scores in WGD tumors (Extended Data Fig. 1c-d), non-WGD metastatic tumors of the three aforementioned cancer types also harbored significantly greater aneuploidy and LOH scores. Furthermore, metastatic patients from other cancer types, including lung and colorectal adenocarcinomas, also displayed a moderate increase in aneuploidy and LOH scores, although they seemed to be primarily associated with higher metastatic WGD rates (Extended Data Fig. 1c). Our results thus revealed that the majority of cancer types have already acquired variable degrees of this genomic feature early in tumor evolution. However, in certain cancers such as kidney renal clear cell, prostate and thyroid carcinomas, significantly increased levels of genomic instability were induced in later evolutionary stages, which were, in turn, associated with substantial karyotypic changes.

3. The analyses in Figures 1e, 2a, 4a and 5a contain a substantial number of statistical tests but there doesn't appear to be any correction for multiple testing. It would be worth running a correction on these tests as has been done for other analyses.

We have now performed multiple testing corrections in the analysis displayed in Fig. 1b, Fig. 1e, Fig. 4a and Fig. 5a. Thanks.

Minor comments:

1. The authors mention a few times they use paired samples for analysis, and while they do use tumor/normal pairs in both PCAWG and Hartwig, there are no primary/metastasis pairs included. While this caveat of the study is clearly mentioned in the discussion, it would be helpful to readers not familiar with the data if the authors would make this point abundantly clear in the introduction and results as well.

We now make this point explicitly clear in the manuscript introduction. Thanks.

Introduction:

Importantly, this dataset encompasses unpaired primary and metastatic tumor samples from 71 cancer types, including 23 cancer types with large representation from both clinical stages.

2. In the analysis of early clonal mutation rates of SBS1 in Fig. 3c/d, it is unclear if SNVs from all regions are included, or only gained regions. If so, some of the increase in the clonal rate ratios may be reflective of more copy number gains in metastases than primaries. It would likely be best to subset this analysis to 2+0 and 2+2 regions, for which all mutations can be classified as either clonal early or clonal late (i.e. there would be no clonal unassigned mutations).

Originally, all SNVs were included in the timing analyses involving both SBS1 and SBS5/SBS40 mutations. This analysis suggested that certain metastatic tumors (breast, prostate, kidney renal clear cell carcinomas) showed an increased SBS1 clonal late ratio compared to their primary counterparts that was not observed in the control cancer types (see below).

Nevertheless, this enrichment was not observed for clonal late SBS5/SBS40 mutations.

Our data thus suggests that differences in the number of amplified regions (where mutationTimerR can estimate the relative timing of both clonal-early and clonal-late SBS1 mutations, copy number ≥ 2 with minor allele ploidy $\neq 1$) is not the only factor contributing to the reported differences.

In any case, following the reviewer's suggestion, we performed the analysis subsetting for mutations overlapping with amplified regions with minor allele copy number !=1. After this correction, breast carcinoma still showed a strong significant enrichment in clonal late SBS1 mutations in metastasis compared to primaries (see below). However, the significance is no longer present for prostate and kidney renal clear cell carcinomas (although sample size is very limited and a modest median increase was still observed in the prostate metastatic patients, see below). Thyroid carcinomas could not be analyzed owing to the limited number of primary samples with sufficient mutations in amplified regions. Finally, colorectal and ovarian serous carcinomas did not show an relative increase in SBS1 clonal late ratio.

Given the complexity of this analysis for a general reader and the challenges associated with the limited number of SBS1 mutations overlapping with these amplified regions in certain primary tumors, we have decided to move the clonal late ratio comparison to the Supp. Note 3.

We thank the reviewer for bringing up this point.

3. In the discussion, what does the statement 'metastatic tumors share [...] fewer short mutations than primary tumors' mean? It is mentioned in an earlier section that 'TMB in metastatic tumors was only moderately increased compared with primary tumors'.

This was a typo. In the revised discussion we have corrected this. Thanks.

4. Minor figure comments:

a. Supplementary Figure 3a-d misses a color legend (blue=metastasis; red is primary); also typo: assotiation -> association

This has now been corrected. Thanks.

b. The Y-axis labels in Supplementary Figure 4a are misaligned

Corrected. Thanks.

Referee #2 (Remarks to the Author):

The authors compare the genomic features of unpaired metastatic and primary cancer tissue biopsies from two published retrospective cohorts (Hartwig–metastatic only; and PCAWG/TCGA–primary only). They report that five cancer types (including prostate, breast, and kidney cancer) show clear differences between localized and metastatic disease while most other cancer types (e.g. ovarian) show relatively few differences. In addition, the authors find evidence for genomic impact of prior treatment, including apparent evolutionary bottlenecks. The authors conclude that 1) they have observed evidence of evolution between the localized and metastatic niche (in breast cancer, prostate cancer, and others), and that 2) whole-genome sequencing of cancer tissue is a valuable tool to understand tumor evolution. Unfortunately, the first conclusion is not adequately supported by the data supplied (details below), while the second conclusion has modest novelty value.

To their credit, the authors analyse an impressive number of samples – clearly requiring a significant investment in time and expertise. However, the major limitation lies in the study design for evaluating evolution – namely the lack of patient-matched primary-metastatic samples. Coupled with some cohort-level heterogeneity, this factors massively constrain the extent to which this dataset can provide new biological or clinical insight (about evolution), and consequently most of the findings presented are low resolution or largely just recapitulate known cancer-specific observations.

First, we would like to thank the reviewer for critical reading of the manuscript and for raising those points that were not sufficiently clear in the previous version of the manuscript.

The primary objective of our study was to compare primary and metastatic cancers, to detect and quantify differences and do this using a systematic (and harmonized) pan-cancer approach. We believe that the outcome of this analysis can provide highly relevant insights about the underlying factors governing metastatic tumor development and resistance to cancer therapies. Moreover, as mentioned by reviewer #1, we believe that the released harmonized cohort can eventually trigger many follow-up studies by other researchers.

However, we agree with the reviewer that our study is not suited for delineating the evolutionary processes and the precise timing of such events because that would require longitudinal paired samples from the same patient. Other studies, such as TRACERx and GLASS, among others; are better suited for this purpose. Nevertheless, we believe that performing a pan-cancer study of this magnitude (>7,000 patients from more than 20 cancer types) relying on longitudinal samples would currently be extremely difficult due to a broad range of logistical and timeline challenges. Lastly, to the best of our knowledge, most of currently publicly available prim-met longitudinal studies relied on exome only - or more targeted- sequencing approaches, which confines the analytical domain in comparison with

WGS approaches (e.g., this is highly relevant for structural variants or mutational signature analyses such as presented in our work).

Taken together, we believe that our work has a strong value for better understanding advanced tumor development and agree with the reviewer that we should be careful using the term 'evolution' for that. We have critically reviewed the main text to avoid suggestions that we intended to study the evolutionary processes in individual tumors or patients.

The cohort-level heterogeneity comment is a very valid point, but also an inevitable challenge as some tumor (sub) types do not evolve into metastatic phase, which by itself is an interesting question and requires datasets and analysis as presented here for better understanding. Nevertheless, we have now performed an extensive harmonization of the patients included in each of the 23 primary and metastatic cancer types. To do so, we have relied as much as possible on the clinical cancer type definition previously used in the TCGA and PCAWG pan-cancer studies (refs DOI: doi.org/10.1016/j.cell.2018.03.022, DOI: 10.1038/s41586-020-1969-6) and further excluded cancer subtype samples from the Hartwig dataset that could not be attributed to one of these 23 cancer type categories. This harmonization has led into the following changes:

- **Lung adenocarcinoma (LUAD) and Lung squamous cell carcinoma (LUSC)** are now two independent cancer types. Therefore, NSCLC as an aggregated cancer type has been removed.
- To prevent subtype intrinsic differences across renal carcinomas we now focus our comparison to the most prevalent subtype, **renal clear cell carcinoma**. Metastatic renal cancer patients from other renal cell carcinomas subtypes (e.g., papillary) are not included in the comparison both in the primary and metastatic cohorts.
- Similarly, we now narrow the definition of ovarian cancer to **ovarian serous carcinomas**. Consequently ovarian cancer metastatic patients lacking this cancer type annotation were not included.
- **Pancreatic neuroendocrine** patients are restrained to well-differentiated pancreatic neuroendocrine tumors.
- For consistency with the PCAWG cohort, Uterus carcinoma is limited to **Uterus endometrial carcinomas**.
- For breast carcinoma we have now annotated the **breast carcinoma subtypes** for primary and metastatic tumors (i.e., ER-/HER2-, ER-/HER2+, ER+/HER2+ and triple negative breast cancer). The comparison by breast cancer subtype is now part of the Supplementary Note 2.
- For **colorectal and uterus carcinomas** we have now annotated the **MSI/POLE** status. The comparison splitting by MSS and MSI/POLE groups is now part of the Supplementary Note 2.
- For the rest of cancer types we have discarded metastatic tumor samples with neuroendocrine or sarcomatoid subtypes. This included samples with any recorded pathological annotations of “neuroendocrine” (except for pancreas neuroendocrine and gastrointestinal neuroendocrine) and “sarcoma” (except for liposarcoma, leiomyosarcoma, and osteosarcoma).

Hartwig cancer type	Cohort Size Preprint	Samples excluded	Cohort Size rebuttal
Bladder urothelial carcinoma	175	47	128
Breast carcinoma	787	11	776
Cervical carcinoma	40	2	38
Colorectal carcinoma	628	7	621
Hepatocellular carcinoma	53	1	52
Kidney renal clear cell carcinoma	129	19	110
Leiomyosarcoma	48	1	47
Non Small lung cancer	511	222	
Lung adenocarcinoma			255
Lung squamous cell carcinoma			34
Ovarian serous adenocarcinoma	168	64	104
Pancreas carcinoma	103	4	99
Pancreas neuroendocrine	40	6	34
Upper respiratory tract carcinoma	39	1	38
Uterus carcinoma	33	1	32
Grand Total	2754	386	2368

On top of these cancer subtype exclusions, we had to exclude 33 additional samples from the Hartwig cohort due to insufficient informed consent that was communicated during the revision process.

After applying these sample exclusion criteria, our dataset includes 23 cancer types (i.e., 22 previously included but splitting NSCLC into LUAD and LUSC) and 8 cancer subtypes (i.e., four breast cancer subtypes and MSS/MSI colorectal and uterus endometrial carcinomas). The total number of patients considered in the updated analysis is 5,365, with 1,914 primary and 3,451 metastatic tumors. Moreover, it is important to remark that additional 1,743 WGS tumor samples (i.e., 7,108 in total) from other cancer types were uniformly processed and also part of the resource herein described.

Finally, concerning the cancer-type specific observations, we agree that some of our results have been (independently) reported in cancer type specific studies (e.g., an important fraction of TEDs have independent evidence in literature), which in our opinion, also highlights the validity of our data as a resource for further studies. Nevertheless, we also believe that putting such previously reported findings in context across multiple cancer types using an harmonized pipeline has never been done before and does, despite its lack of fundamental biology novelty, provide valuable insights.

We now stress the scope and limitations of our approach in the manuscript's introduction and discussion.

Introduction:

Importantly, this dataset encompasses unpaired primary and metastatic tumor samples from 71 cancer types, including 23 cancer types with large representation from both clinical stages.

Discussion:

However, to fully address these questions, matched biopsies from the same patient, as already implemented in cancer type specific studies^{18,65,66}, would be needed. Moreover, this would enable an accurate estimation of the precise timing of genomic events governing tumor evolution, which is very challenging in the current setting.

I am also concerned about the highly comprehensive data dredging – although this type of approach is typical for many pan-cancer “landscape” studies, it makes interpretation of result significance to be very challenging. And of course, we do not see which data tests did not make the final manuscript. Even after incorporating multiple test corrections for some subanalyses, the scale of testing means that many associations run a significant risk of being spurious.

We agree with the reviewer about the extensiveness of data and analyses herein presented may be overwhelming. However, all analyses (and consequently, their relevance) are inspired by previous pan-cancer landscape studies that are currently a reference for the research community (e.g., flagship paper of PCAWG). We believe the cancer genomics community at this stage is likely better served by

a single paper with a comprehensive pan-cancer prim-met comparison than with a large series of individual papers.

In this manuscript, we covered the main genomic topics that can be studied with WGS, but also we believe that by releasing the harmonized dataset we are enabling more dedicated studies. An example that showcases that topics we didn't cover can be eventually conducted in a more focused manner is our recent work (DOI:10.1101/2022.02.23.481444v3) analyzing the prevalence and impact of immune escape alterations in primary and metastatic tumors.

To flesh out the comment above regarding lack of patient-matched samples, the fundamental issue is that several of the highlighted cancers with genomic differences between primary and metastatic tissue (e.g. prostate, breast, kidney) can have limited metastatic potential. For example, it is well-established that most localized prostate cancers are relatively indolent and may not require radical interventions. However, surgery/radiation therapy is often performed due to inadequate risk stratification models (to accurately resolve aggressive from indolent). The same is true (albeit to varying extent) in kidney, breast cancer, thyroid, pancreatic neuroendocrine. Therefore, to study primary-metastatic evolution in the absence of patient matched primary and metastatic samples, it is necessary to enrich the primary datasets with samples from patients that subsequently (or in parallel) developed metastatic disease. In prostate, this can be only ~10% of patients that present with local cancer, in pancreatic neuroendocrine and thyroid cancer, it is even fewer. Whereas in prostate it is known that localized and metastatic cancers have distinct driver landscapes (e.g. Armenia et al Nature Genetics 2018), it is also known that primary tissue samples taken from patients with metastatic disease show fairly similar driver gene profiles (e.g. Mateo et al. JCI 2020).

We do agree with this limitation as indicated, discussed and addressed above. However, one of the most relevant outcomes of the current study is that it revealed that the differences between primary and metastatic tumors are highly cancer type specific, which does not contrast with the indicated previous studies. It should be noted that this was only possible due to having an harmonized pan-cancer analysis of this magnitude, which encompasses clinically representative cohorts of primary and metastatic patients. For instance our data revealed that while ovarian, pancreas carcinoma and glioblastoma metastatic tumors did not show strong changes in their genomic landscape, others, such as prostate, thyroid or pancreatic neuroendocrine underwent profound genomic change at metastatic setting.

As we mentioned in a previous response, a different research question is which are the specific mechanisms (and their relative timing) governing metastatic evolution in individual patients. To address that question, It is necessary to have primary and longitudinal samples of patients that eventually present with metastatic disease.

However, we agree it is still relevant to study whether the intense genomic changes that certain cancer types show at the metastatic setting are exclusively associated with differences respective to primary relatively indolent tumors or alternatively, such genomic discrepancies would also hold when comparing to primary tumors that would eventually progress. This is of course mostly relevant for cancer types showing a strong remodeling of their genomic make up (i.e., breast, prostate, thyroid, kidney clear cell and pancreatic neuroendocrine carcinomas).

Annotation of clinical progression of primary tumors was only available for a subset of patients from the cancer types of interest. In practical terms, only prostate and pancreatic neuroendocrine included sufficient numbers of patients with annotation of clinical progression of the primary tumors (see Supp. Table 1). Hence, for these two cancer types, we performed the genomic analyses conducted in this study but splitting primary tumors according to their clinically annotated progression status (now part of Supp. Note 2).

Main findings and conclusions (from Supp. Note 2):

In general, our analysis indicated that metastatic prostate and pancreatic neuroendocrine tumors are genomically different from their primary counterparts, independently of the clinical progression of the primary patients. However, the magnitude of the differences was higher when comparing to primary patients with a relatively good prognosis than when we compared to primary tumors with worse prognosis. Moreover, our results also highlighted some potential differences associated with worse prognosis and metastatic transformation (e.g., APOBEC activity, SV burden, Recurrent Chromosome Pattern Losses in pancreatic neuroendocrine, among other things, see Supp. Note 2).

Finally, concerning the prostate cancer driver landscape (explicitly mentioned by the reviewer), we noticed certain differences between the two primary prostate carcinoma groups (i.e., primary progression/relapse [PROG/RL] compared to primary patients stable or with remission [ST/RM]), which were very much in line with previous observations and that, for most of the cancer driver genes, did not explain the full extent of differences with metastatic tumors (see table below for a summary of the top prostate cancer driver genes).

Driver	Freq. J Mateo et al. 2020	Freq. Armenia et al 2018	This dataset (PCAWG vs Hartwig)
---------------	----------------------------------	---------------------------------	--

Cohort group	Prim. PROG/RL	Met	Primary	Met	Prim. ST/RM	Prim. PROG/RL	Met.
TP53	25%	>>	~18%	~35-40%	3%	17%	55%
PTEN	11%	>	3%	6%	12%	42%	47%
RB1	3%	>>	0.40%	4%	3%	0%	13%
BRCA2	8%	~	1.80%	6%	3%	0%	8%
AR	0-1%	>>>	~0%	~48%	0%	0%	43%
SPOP	3%	~	10%	6%	6%	17%	6%
MYC	6%	>>	~4%	10%	0%	0%	5%

All details of this analysis are now part of Supp. Note 2. Moreover we have updated certain parts of the manuscript to mention the most important implications of this analysis:

Mentions in the main text body.

SV Burden:

Moreover, we observed an increased SV burden in prostate and pancreatic neuroendocrine primary tumors that eventually progressed compared to those with relatively better prognosis, which in both cases were in turn lower than the median values in metastatic tumors (see Supp. Note 2). Further studies are required to elucidate whether such patterns are the footprints of underlying evolution via structural variation or reflect a preference towards selection of primary subclones with higher SV burden

Discussion:

It is noteworthy that the cancer types with strongest genomic differences typically have a very good prognosis in the primary setting. But then, are the metastatic tumors representing a unique set of primary patients that eventually progressed (i.e. primaries from the metastatic cohort were “born to be bad”) or are there stochastic triggers of metastatic disease in relatively indolent primaries?

Interestingly, a small-scale comparison in prostate and pancreatic neuroendocrine carcinomas revealed that substantial genomic differences between primary and metastatic tumors were still observed after subsampling for primary patients that eventually progressed (Supp. Note 2). However, to fully address these questions, larger sets of matched biopsies from the same patient, as already implemented in various cancer type specific studies^{18,65,66}, would be needed. Moreover, this would enable an accurate estimation of the precise timing of genomic events governing tumor evolution, which is very challenging in the current setting.

Additional major comments

1. Samples are processed with different pre-analytical laboratory workups and sequencing parameters, and notably there is a large imbalance in sample availability between comparator groups (1916 primary versus 3835 metastatic samples).

As we mentioned in the manuscript the fact that both cohorts rely on different laboratory workups and sequencing parameters remains one of the limitations of the study. However, we also believe that the biases introduced by such differences should be minimal as otherwise we would have observed systematic differences across all cancer types included, which is clearly not the case. Moreover, as mentioned by the reviewer before, our manuscript identified some previously characterized differences and similarities in cancer type specific studies, illustrating the validity of our analysis. In any case we acknowledge this limitation in the manuscript discussion:

This study also faced other limitations. First, primary and metastatic tumor samples were processed using different laboratory workups and sequencing parameters. Moreover, sequencing depth of the primary tumor cohort was lower and more variable than that of the metastatic tumor cohort, although we demonstrated that this does not severely impact on the overall detectability of clonal somatic variants (Supp. Note 1).

Concerning the dataset imbalance, all cancer types included in the analysis had a minimum of 15 patients in both primary and metastatic cohorts, after applying a strict quality control filtering and a clinical-level harmonization (see Supp. Table 1 and Supp. Note 1). Furthermore, all cancer-type specific analyses were performed either using the relative frequency normalized by the total number of patients included in that cohort or using the median/mean number of events per tumor sample. Finally, the fact that the cancer types with the highest genomic discrepancies (i.e., prostate, kidney renal clear cell, pancreatic neuroendocrine and thyroid) do not have a significant imbalance suggests that this is unlikely to be the underlying reason for the reported differences.

2. Metastatic and primary biopsies are profiled at different sequencing depths of (109x for metastatic versus only 38x or 60x for primary). Low primary sample depth is liable to constrain

detection sensitivity across the spectrum of genomic events (especially copy alterations and ploidy). The distribution of depth (38x vs. 60x for primary samples) is also somewhat cancer specific, e.g. bladder cancer is almost exclusively 38x, whereas esophageal cancer is almost exclusively 60x. The authors perform some benchmarking to rule out this large discrepancy in sequencing depth as a confounder in somatic variant detection rates, since the higher sequencing depth in metastatic samples could create an 'illusion' of enrichment for SNVs / SVs due to improved depth-related detection sensitivity. However I'm concerned that this confounder is not possible to completely mitigate.

We agree with the reviewer that it is impossible to fully mitigate all confounding factors that may influence that mutation calling sensitivity. This remains one of the drawbacks of large pan-cancer data resources that rely on international collaboration efforts for sample inclusion such as PCAWG and TCGA. Nevertheless, PCAWG dataset has been extensively used to study multiple facets of tumorigenesis with important cancer-type specific findings that are unlikely to be correlated to different sequencing depth contributions (see PCAWG flagship paper).

Since the Hartwig cohort overall has a higher sequencing depth than the PCAWG cohort, we performed an extensive sensitivity analysis. As detailed in Supp Note 1, we show that the estimated percentage of sensitivity loss is reasonable for all types of variants to perform quantitative and qualitative comparisons.

See results and Supp. Note 1:

"[...] decreased sensitivity at 60x and 38x coverage, typical for PCAWG samples, compared with 109x coverage, typical for Hartwig samples between 0% and 5% for simple mutations and between 8% and 14% for SVs respectively."

Moreover, if sequencing depth would impact the mutation and SV calling, we would observe a systematic increase in TMB and SV load across multiple metastatic cancer types which is not the case. For instance, primary glioblastoma or bladder tumors were mainly sequenced using 38x coverage, but none of them displayed significant differences in SNVs burden. Moreover, as we show in Supp. Note 1, these cancer-type specific differences in SBS/indel and SV burden remain significant when correcting for sequence depth sensitivity loss. Thus, although the reviewer is of course right to raise concerns regarding quantitative comparisons on cohorts with varying sequencing depths, we show that this effect is minimal and we believe that similar global conclusions can be drawn if all samples would have been processed with equal sequencing depth. As mentioned above, we acknowledge these limitations in the manuscript discussion.

3. Figure 1d: the conclusions of substantial karyotype changes in specific cancer types may benefit from some clarification. The distribution of arm level gains/losses in primary vs. metastatic kidney clear cell cancer and prostate cancer in fact appears quite similar (i.e. most arms are copy-altered in the same 'direction'). In prostate cancer, 8p loss / 8q gain is an established early event and is therefore not expected to be different across primary and metastatic disease (contrary to what the Q -value <0.05 may indicate). Are there other technical or analytical reasons that may partly explain some of these differences?

We agree with the reviewer that this section can benefit from better clarification and interpretation of the findings. As discussed in comment 2 from reviewer #1, we completely revised the karyotype methodology.

Briefly, we now normalize the copy number levels relative to the expected genome ploidy (i.e., $2n$ for tumors that do not undergo WGD, $4n$ for WGD tumors, $8n$ for tumors with two WGD, etc). Moreover, we perform a normalized scoring of arm gains and losses (i.e., -1: chr. arm loss; 0 neutral and 1 chr. arm gain) to render all chromosomal gains/losses comparable independently of the global genome ploidy. Finally, to address the effect size issue mentioned by the reviewer, we now require a minimum of 0.25 normalized copy number change in order to call a chromosome arm change as significant (on top of the adjusted p -value < 0.05).

After applying these changes, our analysis revealed that the karyotype portrait is strongly shaped by the cancer type and it is highly conserved between primary and metastatic stages in nearly all cancer types. However, three cancer types, namely prostate, thyroid and renal clear cell carcinoma are exceptions to this rule and collectively represent 91% (43 of 47) of all significant hits. Of note, 30% of these karyotype changes were retained when subsampling for non-WGD tumors, suggesting that there may be other factors that explain these differences in kidney, prostate and thyroid tumors beyond WGD (see response 2 to reviewer #1 for full description of new results).

It is indeed true that chromosome 8p loss and 8q gain are already present in an important fraction of primary prostate carcinomas. However, we still observed a substantial difference in the frequency of these events in metastatic tumors compared to primaries (average normalized ploidy 8p: -0.42 in PCAWG and -0.76 in Hartwig and average normalized ploidy 8q: 0.12 in PCAWG and 0.57 in Hartwig). In fact, the average ploidy values for primary tumors estimated by our pipeline is very much in line with the values reported by TCGA (8p -0.41 and 8q average ploidy of 0.13) from A.M. Taylor et al. study (10.1016/j.ccell.2018.03.007, see table below). Furthermore, the combination of 8q loss and 8p gain has been reported as a marker of poor prognosis and tumor aggressiveness (see 10.1158/1078-0432.CCR-09-1423). Finally, 8p gain have also been associated with resistance to AR deprivation therapies via MYC (10.1038/s41467-022-30257), PRNCR1 (10.1038/nature1245) and NCOA2 (DOI:10.1172/JCI76412) overexpression/amplification.

Average aneuploidy scores of the two arms from chromosome 8 of primary prostate cancer from our study and the work from Taylor et al, compared to metastatic:

Chrom Arm	Primary PCAWG (our study)	Primary TCGA (Taylor et al)	Metastatic (this study)
8p	-0.42	-0.41	-0.76
8q	0.12	0.13	0.57

For example, can the authors rule out systemic differences in primary vs. metastatic sample tumor purity (e.g. lower primary purity reducing CNV detection sensitivity compared to metastatic disease)?

It is true that low tumor purity can severely impact the sensitivity to detect both small and large variants. That is the reason we excluded all samples with tumor purity lower than 20% from our analyses (see methods and Supp. Note 1). Moreover, all analyses performed are corrected by tumor purity, which in the case of PCAWG samples, we showed to be in agreement with the tumor purity values originally reported by PCAWG (see Supp. Note 1). In any case, to rule out that the observed differences in SV burden and CNA were linked to a systematic bias in tumor purity we performed a cancer type specific comparison (see below).

Our comparison indicated that PCAWG samples tended to have overall higher tumor purity compared to Hartwig samples, particularly in some cancer types such as lung adenocarcinoma, skin melanoma, colorectal carcinoma and bladder urothelial cancer, among others. This is not surprising considering the different sample collection approaches followed by both cohorts. While most of PCAWG tumors were sequencing following surgical resection, Hartwig samples are typically taken from metastases and obtained by needle biopsies. Moreover, metastases are typically much smaller than the primary tumor masses and, in certain cancer types, they are also often more heterogeneous including more stromal cells in the lesions. Consequently, lower tumor purity alone is not a confounding factor for the identification of somatic variants and copy number estimation in the PCAWG cohort.

4. It would be helpful to better understand how the variable of differential whole-genome doubling (WGD) status impacts the normalization of CNVs relative to mean ploidy. For example, haploid LOH in a diploid tumor will have half the absolute difference (relative to mean ploidy) of the same

biological event in a WGD tumor. Does comparing average absolute CN difference account for the apparent “enrichment” of gains and losses in some of the metastatic cases (which are overall slightly enriched for WGD)? Are there better ways the authors can control for variable WGD status (or indicate that this does not underlie some of these potential differences)?

The reviewer is right (this point was also mentioned by reviewer #1) that in the previous karyotype analysis, the WGD status influenced the weight of each tumor sample in the global average, and subsequently, the results of the comparison (given the differences in WGD rates in certain cancer types). To address this issue we have performed a series of adjustments in our methodology (see response to reviewer #1 and response to this reviewer above), resulting in a normalized scoring of chromosome arms gains and losses (i.e., -1: chr. arm loss; 0 neutral and 1 chr. arm gain).

These changes resulted in much better representation of the consequent losses of chromosomal arms in WGD samples and much less found in non-WGD samples. By contrast, the patterns of amplified arms are highly similar between WGD and non-WGD samples (see Extended Data Fig. 1c-d). This global pattern of arm losses in WGD samples is well described by other studies that showed that WGD cancers can tolerate more easily the loss of a chromosome arms (reviewed in <https://doi.org/10.1038/s41568-018-0084-6>) and also shown in the Taylor et al study) because of higher buffer capacity.

Please consider reading the full response to reviewer #1 for further details about the changes introduced in this section and their implications.

5. Relatedly, the methods text for “Karyotype” determination need substantial development. It is unclear how genome-wide CN information was derived (and related metrics including arm-level CN changes and mean ploidy).

We have now re-written the Karyotype methods section to improve readability.

6. De novo signature analyses: The observation that DBD_denovo_clust_1 and DBS_denovo_clust_2 are associated with POLE mutations and MSI (in proposing potential etiologies) suggests that these ‘de novo’ signatures could be plausible matrix factorization artefacts, given that POLE mutations and MSI also causally underlie established COSMIC DBS signatures 3 (POLE) and 7/10 (mismatch repair deficiency). What are the cosine similarities of these de novo DBS signatures to their established COSMIC signature counterparts?

It would be helpful to produce DBS context plots for all de novo signatures generated so readers can intuitively compare to previously catalogued DBS signatures. Overall some deeper analysis will need to be performed in order to claim DBD_denovo_clust_1 and DBS_denovo_clust_2 as 'novel' signatures.

The reviewer is right that the claiming of a novel mutational signature/s requires further evidence. In fact, we believe that dedicated studies would be needed to robustly annotate the etiology of mutational signatures. Therefore we decided to downplay this finding in the main manuscript leaving it as suspected etiology.

Moreover, as proposed by the reviewer, we are now sharing the mutational profile and the pairwise similarity of all extracted mutational signatures (Supp. Table 3, Supp. Data 1 and <https://zenodo.org/record/7396538>), including the mutational profiles of DBS_nenovo_1 and DBS_denovo_2, so the reader could use it to further investigate potential etiologies in dedicated studies.

Appendix:

Here the profile of DBS_denovo_1/DBS_denovo_2 and DBS3/DBS7/DBS10 alongside their cosine similarities.

No de novo signatures matched to DBS3/7/10. However, below are the cosine similarities of DBS_denovo_1 and DBS_denovo_2 to DBS3/7/10 as well as the profiles of these 3 COSMIC signatures.

assigned_sig_name	denovo_sig_name	DBS3	DBS7	DBS10
DBS_denovo_1	Biliary.DBS_B	0.23	0.626	0.163
DBS_denovo_1	Liver.DBS_E	0.047	0.69	0.063
DBS_denovo_2	Biliary.DBS_C	0.202	0.49	0.142
DBS_denovo_2	Breast.DBS_D	0.241	0.784	0.166
DBS_denovo_2	Colorectum.DBS_C	0.206	0.675	0.136
DBS_denovo_2	Gastric.DBS_B	0.28	0.797	0.183
DBS_denovo_2	Uterus.DBS_A	0.189	0.676	0.137

As illustrated by the mutational profile and their pairwise similarities, DBS_denovo_1 and DBS_denovo_2 have modest similarity to known MMRd/POLE DBS mutational signatures. While the TT>NN channels are partially shared, these known signatures would not explain the high contribution from other channels in DBS_denovo_1 (e.g., TC>AA) and DBS_denovo_2 (e.g., AC>CA).

7. Figure 2: Comparing median differences in mutation/indel count (stratified by signature assignment) is confounded by broader differences in mutation rates (between cancer types and between primary vs. metastatic tumors overall). It would be easier to interpret these plots (and all Results text) if instead comparing median signature proportions. Do the proposed etiological differences between primary and metastatic disease still hold after normalising for differences in

mutation counts? It is very difficult to evaluate the soundness of most signature-related claims while the results are in units of mutation count. The entire Results section “Differential SBS1 mutation rates in primary and metastatic cancers” needs to be reanalyzed in the unit of signature proportions.

We understand the point raised by the reviewer. However, we would like to emphasize that the primary goal of the mutational signature analysis is to identify -and to quantify- the contribution of the mutational processes accounting for the increased tumor mutation burden in metastatic tumors. To do so, it is necessary to perform an analysis based on absolute median contributions across primary and metastatic tumors.

A slightly different question (although still interesting), is the relative weight of different mutation processes within primary and metastatic tumors and how they compare to each other. Nevertheless, this approach is highly sensitive to changes in the full spectrum of operative mutational processes in a particular tumor. For instance, it would be possible that mutational processes exclusively active to the metastatic tumors (e.g., treatment associated mutations) would mask the contribution of mutation signatures that may have been present from early tumorigenesis, even if this is still partially responsible for increased mutation accumulation in late-stages. Moreover, by definition, a substantial significant relative increase in one metastatic process (e.g., treatment mutations or APOBEC) would lead into a relative decrease in other mutational processes which makes the final interpretation quite challenging.

All things considered, we believe that for the SBS1 mutation rate section, as we want to explicitly focus on SBS1 mutation burden independently of other mutational process; it is better to rely on the absolute contribution.

However, we also believe these two approaches are complementary to understand primary and metastatic global TMB differences (see Fig. 2c-e and new Extended Data Fig. 3). We thus complemented our analysis by performing a mutational signature analysis using relative contributions:

Supplementary Figure 3

a

As expected, given the frequent, and essentially exclusive, exposure of metastatic patients to mutagenic cancer therapies, mutational processes associated with radio/chemotherapies were strongly enriched in metastatic tumors across multiple cancer types. For example, platinum treatment mutations showed a ~10% relative increase in metastatic ovarian serous, uterus and head and neck carcinomas (i.e., SBS1/SBS35) and a 5-30% relative increase in DBSs (DBS5) in multiple metastatic cancer types. Moreover, indels associated with defective DSB repaired by NHEJ (represented by ID8), which are correlated with radiotherapy exposure, were also relatively enriched in four metastatic cancer types.

APOBEC relative exposure (SBS2/SBS13) was significantly enriched in breast cancer tumors and it also displayed a non-significant enrichment across multiple metastatic cancer types (see dark-blue top bars of the first panel).

Mutations attributed to SBS5/SBS40 clock-like mutational processes were comparatively depleted in at least 3 metastatic cancer types (breast, kidney renal clear cell and prostate). However, as mentioned above, it is challenging to know whether this relative depletion is caused by lower mutation rate or by an increased exposure to other mutational processes in late-stage tumors. Of note, mutations attributed to SBS1, the other clock-like mutational process, did not show such relative depletion in metastatic tumors, even after considering that these metastatic tumors showed a strong contribution from metastatic exclusive mutational processes (i.e. treatment).

This is therefore compatible with our observations of an age-corrected SBS1 mutation burden increase in certain metastatic tumors.

This analysis is mentioned in the results section and part of the new Extended Data Fig. 3.

Results:

Expectedly, analysis based on relative mutational signature contribution revealed the importance of treatment associated mutations in advanced stages of tumor evolution, where platinum and radiotherapy associated mutations and indels showed a significant relative enrichment across several metastatic cancer types (Extended Data Fig. 3).

8. The authors should adjust some of their language describing treatment-associated enrichment of mutation signatures, since these are correlative associations (i.e. not causal). E.g. "... DBS2 occasionally led..." etc.

We have revised the language describing treatment-associated mutations through the manuscript.
Thanks

9. Results text: "In fact, the excess in DBS mutation burden observed in six cancer types (stomach, esophagus, cervix, upper respiratory tract, non-small cell lung, and urothelial cancer) could be fully attributed to platinum treatment (Fig. 2d top bars)." Can the authors clarify what they mean by "fully attributed"?

It means that the increase in median DBS mutations per sample observed in Hartwig compared to primary tumors in these cancer types is primarily explained by an absolute increase in DBS suspected to be associated with platinum therapy (i.e., CC>AA and CC>AG mutations, see DBS5 mutational profile <https://cancer.sanger.ac.uk/signatures/dbs/dbs5/>).

Had the metastatic cohort patients included in these analyses received confirmed cytotoxic or platinum treatment prior to metastatic biopsy (and conversely, were the primary biopsies treatment-naive)?

Indeed, metastatic patients had generally received previous treatment, which included, among other therapies; platinum based chemotherapies. Conversely, primary tumors from the PCAWG cohort were, in the majority of cases, treatment-naive (see Supp. Table 1 for patient specific information).

10. Results text: “The systematic enrichment of SBS2/13 mutations in metastatic cancers suggests enhanced activity of APOBEC mutagenesis during the progression of advanced tumors” – the enrichment of APOBEC related mutational processes appears largely driven by only a few cancer types (notably breast and stomach cancer), so it feels incorrect to claim ‘systemic enrichment’ across metastatic cancers.

In fact, there are six cancer types with a significant enrichment of median APOBEC mutation contribution in metastatic patients compared to primaries (including breast and stomach cancer). Moreover, in two additional cancer types, namely cervical cancer and bladder urothelial carcinomas, we also observed a substantial increase (increase in ~2,000 SNVs associated with APOBEC mutational signatures), but they did not reach significance owing to high intrinsic APOBEC activity in the primary tumors. Hence, 8 cancer types out of the 23 (~35%) included in our study have a strong enrichment in median exposure of APOBEC mutation load. We believe that this number is sufficiently remarkable to be explicitly mentioned in the main manuscript. However, we have replaced the term “systematic” for “**broad**”, which better recapitulates our findings.

11. Results text: “... the SV analyses revealed a much more widespread effect, with larger increases per metastatic cancer type that affected almost every cancer type studied, indicating that metastatic tumors appear to evolve primarily by genomic changes at the structural level.” – this conclusion appears to contradict the earlier result that copy-number landscapes appear relatively unchanged between primary and metastatic disease (SVs and CNVs can often be reflective of the same underlying process). Can the authors clarify or offer some deeper analysis (perhaps of the CNV landscape beyond broad arm-level changes)?

We agree with the reviewer that terminology of both somatic events may trigger confusion because both events include gains and losses of DNA sequences. However, arm-level alterations and SVs are conceptually different somatic entities with different origins and underlying mechanistic forces.

Aneuploidy refers to the presence of an aberrant number of chromosomes or chromosome arms, generated by a higher than normal rate of missegregation of chromosomes or chromosome arms during mitosis (i.e., chromosomal instability (CIN)). Although CIN can continuously generate new aneuploidies (10.1038/s41588-021-00891-2), the mean aneuploid karyotype is very stable within a cancer and is typically generated early in cancer development (see Figure 1d, but see also refs 10.1073/pnas.93.14.708 and 10.1038/s41586-019-1907-7). Moreover, the highly characteristic recurrent profiles shared between tumors within a cancer type indicates that each cancer type has its

own optimal karyotype fitness, although the underlying selective processes driving these characteristic profiles are not well understood.

For the SVs, we rely on the same terminology as previously used in the PCAWG SV paper (DOI:10.1038/s41586-019-1913-9): A ‘structural variant’ manifests as a ‘junction’ between two ‘breakpoints’ in the genome. This is different from chromosome arm-level alterations which are characterized by a single breakpoint at the centromere, although these cannot properly be localized with short read data because of highly repetitive regions at the centromeres. Thus, the SV sections integrates deletions, duplications, inversions and insertions (e.g. LINE retrotransposon insertions) and are characterized by much shorter DNA sequences than chromosome arm-based level alterations (mean size of 0.9Mbp and 0.7Mbp for duplication and deletion, equal to 1/50 of the shortest chromosome 22, see Extended Data Fig. 5a). The SVs we use in our work are called by LINX, which predicts all the SV events into local derivative chromosome structure, and are thus normalized against broad-arm level alterations. Besides of the few SVs that embeds cancer driver genes, most of the SVs are passenger mutations and can thus be leveraged with pattern recognition tools to dissect underlying mutation mechanisms (which cannot be done with arm-level based alterations due to high cancer type related selection processes). For instance, we have previously used 1-10Kbp duplications to detect BRCA1 deficient tumors (DOI: 10.1038/s41467-020-19406-4). Moreover, we have recently shown that radiation therapy induces SV deletions up to 10Kbp in healthy tissue (DOI: 10.1038/s41467-022-33663-5). Reassuringly, we do see these correlations in our linear regression model approach where we aim to link the increased SVs burden in metastatic tumors with the underlying mechanisms (see Fig. 4 panels b-f).

In order to mitigate this confusion, we now specifically mention “chromosome arm aneuploidy profiles” in the aneuploidy section and included the PCAWG SV definition in the SV section of the main text: *The total number of SVs per tumor, collectively all chromosomal junctions with at least two breakpoints, ...*

12. I am worried that attempting to connect SBS1 mutation counts and “yearly mutation accumulation” and turnover rates is potentially tenuous. Metrics using clonality are very challenging because the primary samples are low sequencing depth (2-3 fold lower than compared to metastatic samples) and I suspect this systematically biases some of the results. Copy number analysis is also crude (e.g. insufficient controlling for WGD, impact of variable purity on calling of copy changes) further roughening all clonality measurements.

The Figure title, “Cell cycle division rates in primary and metastatic tumors” is inappropriate: this has not been directly tested and is only surmised through several layers of simplifying assumptions.

First, as mentioned before, we have evaluated the potential impact of different sequencing coverages for mutation calling sensitivity, resulting in an estimated decreased sensitivity ranging from ~1% to

5%, which is several orders of magnitude lower than the observed difference in SBS1 mutation burden. Second, if there were a systematic bias, we would have observed such a bias across all cancer types (or at least a widespread effect, which is clearly not the case). Third, the analyses based on mutation clonality (i.e., Fig. 3c-d, Extended Data Fig. 4e-f and Supp. Note 3) are normalized by the total number of clonal/subclonal mutations detected in each tumor sample, rendering them comparable across independent patients. Fourth, and most importantly, we now performed an extensive validation of the age-corrected increase in SBS1 mutation burden relying on multiple independent datasets, including primary-metastatic paired biopsies with comparable sequencing depths (see response to reviewer#1 and Supp. Note 3).

Nevertheless, we agree with the reviewer that the title of the figure does not exactly recapitulate our observations, so we have now adjusted the Figure 3 and Extended Data Fig. 4 titles to more accurately describe our findings:

“Age-corrected SBS1 mutation rates in primary and metastatic tumors”.

13. The “long tail of candidate drivers of resistance” is not really appropriate, given lack of paired (patient-matched) samples. With the confounding dataset issues raised in my introductory comments, the dataset probably too crude to pick up meaningful associations beyond established common resistance alterations (like ESR1, PIK3CA, AR, etc.).

First we would like to highlight that the term used in the manuscript is Treatment Enriched Drivers (TEDs), which in our opinion accurately describes the essence of these alterations. In the majority of cases, as mentioned by the reviewer, the underlying reason for the observed enrichment is the emergence as a resistance variant to the specific cancer therapy. However, as we acknowledge in the manuscript, we can not rule that some of the identified TEDs are associated with global tumor aggressiveness and/or with overall worse prognosis, particularly for non-exclusive TEDs (e.g., TP53, see Extended Data Fig. 7b).

However, the fact that the most significant hits are well-known cancer therapy resistance drivers (AR, EGFR secondary mutations and amplifications, ESR1, PIK3CA, among others), inform about the sensitivity and specificity of our approach to detect driver resistance alterations. In fact, 23 out 61 (37%) TEDs have independent supporting evidence from literature. The updated analysis, which also includes breast cancer subtypes treatment groups, identified ACTL6A proximal promoter mutations in triple negative breast cancer as significantly enriched in platinum treated patients compared to untreated patients (see Supp. Table 8). Interestingly, a recent study proposed ACTL6A overexpression as a new mechanism of cisplatin resistance across several cancer types

(DOI:10.1073/pnas.2015808118). All things considered, we believe that among newly identified TEDs (i.e., the long tail) there could be an important representation of new candidate resistance drivers that could serve as starting point and/or orthogonal support for more dedicated studies.

14. Code availability statement: can the authors please provide their full documented source code so this can be reviewed?

Yes, we are now providing the source code to reproduce the analysis included in the manuscript and supplementary notes (<https://github.com/UMCUGenetics/primary-met-wgs-comparison>). Moreover, the source code of the Hartwig medical foundation analytical pipeline is open source and freely accessible to the community (<https://github.com/hartwigmedical/pipeline5>).

Minor:

- Figure 1a: The Biorender patient schematic is fairly low resolution, you may wish to adjust this.

Solved, thanks.

- Supplementary Note: Similarly, many of the Figures are very low resolution and are difficult to scrutinise for details (e.g. Supp Note 1 Figure 5).

We have now improved the quality of the figures in Supp. Note 1. Thanks.

- Figure 5a: It looks like the figure did not render properly or was excessively rasterized. There is a stray vertical line abutting the leftmost column.

Solved, thanks.

Referee #3 (Remarks to the Author):

Martínez-Jiménez characterize the genomic differences between early-stage untreated primary tumors and late-stage treated metastatic tumors using a harmonized pan-cancer re-analysis of 7,152 whole-genome sequenced tumors. The authors reveal notable differences between primary and metastatic tumors in this unpaired analysis.

This is an important area of research, exploring an impressive data-set. On the whole, the manuscript is clear (with a few exceptions, see below). One key limitation is the fact that the analysis is unpaired. While I appreciate this is nevertheless an impressive dataset, I would urge the authors to consider whether they can control for some of the biases introduced by virtue of the types of comparison. For instance, it would be informative to consider two types of untreated primary tumors - those that subsequently metastasize and those that never metastasize.

We thank the reviewer for the appreciative comments of our work and for critical reading of the manuscript.

We indeed agree that comparison by clinical progression of the primary tumors could be informative (also suggested by reviewer #2), particularly in cancer types displaying substantial genomic differences between primary and metastatic tumors (i.e., prostate, kidney renal clear cell, breast, pancreatic neuroendocrine and thyroid carcinomas). Unfortunately clinical follow-up data was only available for a subset of prostate and pancreatic neuroendocrine cancer patients. Therefore, we now performed the suggested analysis in these two cancer types.

In general our analysis indicated that metastatic prostate and pancreatic tumors are genomically different from their primary counterparts, independently of the clinical progression status of the primary patients. However, the magnitude of the differences was higher when comparing to primary patients with a relatively good prognosis than when we compared to primary tumors with worse prognosis. Please check the detailed response to reviewer #2 now part of the Supp. Note 2 for all the particularities of this analysis as well as the discussed implications.

And, for many of the analyses, there is the possibility that subtypes are heavily influencing the results if not controlled for.

We agree with the reviewer that with the current setting it is impossible to fully mitigate the potential differences introduced by clinical-level heterogeneity. In fact, the only way to fully address this issue

is in analysis of paired samples from the same patient, which we acknowledge is of course the best setup, but impossible in the current situation (also because it can be anywhere between 1 and 20 years that a primary tumor metastasizes, see response to reviewer #2). Moreover, differences in subtype frequencies can also be considered of interest for the reader, rather than something that should be controlled for.

Nevertheless, given the relevance of the question (also raised by reviewer #2 and partially by reviewer #1), we have performed an extensive harmonization of the clinical definitions used in the primary and metastatic cohorts (see response to reviewer #2 for all details). Additionally, we have extended the analysis to clinically relevant cancer subtypes with sufficient representation in both datasets (i.e., breast, colorectal and uterus carcinomas, see Supp. Note 2 for a full description of the results).

The main conclusions after these changes are:

- **Less karyotypic differences.** Tumor karyotypes showed cancer type specificity and higher resemblance between primary and metastatic tumors. This is consistent with an early tumor karyotype fixation early in tumor evolution.
- **Driver landscape is even more consistent.** Some previously reported driver mutation frequency discrepancies are no longer present (e.g., PIK3CA/MAP3K13 enrichment in primary NSCLC). This particularly concerns drivers from the old NSCLC cancer type, which is now split into LUAD and LUSC cancer types.
- **The majority of previously reported TEDs are still present** after cohort harmonization and are consistently found when relying on cancer subtypes. Moreover, the few that are no longer significant (e.g., MET and CDK6 amplification in LUAD patients treated with Anti-EGFR) are probably caused by the lower sample size of the updated cohorts (see Supp. Table 8).
- Breast, colon and uterus cancer type level genomic differences between primary and metastatic tumors are either **attributed to one or to multiple cancer subtypes**, rather than to an heterogeneous cancer subtype cohort composition (see Supp. Note 2 and Supp. Note 3).
- **The ER+/HER2- breast cancer subtype is the most genomically divergent** between primary and metastatic tumors, followed by triple-negative. HER2+ is the most genomically similar (see Supp. Note 2). The differences reported in the main cancer type are mostly attributed to the most populated breast cancer subtype: ER+/HER2-.
- The **observed discrepancies** between primary and metastatic tumors in **colorectal and uterus carcinomas are attributed to the MSS subtypes**, which are the most prevalent in our primary and metastatic cohorts.

Finally, as we now mention in the discussion, we think that new dedicated cancer type specific analyses will be enabled by releasing the datasets and the accompanying clinical data.

Likewise, all mets of a given tumor type are considered equally (for the most part) - an analysis of how sites of met influences the genomic landscape would be very informative.

We agree with the reviewer that this is also a relevant point. Therefore, in Supp. Note 2, we have now reproduced all analyses of the manuscript grouping metastatic tumors according to the three types of metastatic locations defined in our metastatic cohort (i.e., local, lymph nodes and distant metastasis).

Main findings and conclusions (from Supp. Note 2):

In general terms, our data suggested that, except for the tumor clonality, the genomic landscape of metastatic tumors is conserved across different metastatic biopsy locations and that the reported differences with primary tumors were metastatic location-independent.

The most remarkable exception to this trend was lung adenocarcinoma (LUAD) lymph node metastasis, which showed an increase in genomic instability, TMB, SV burden and certain differences in their driver landscape compared to both primary tumors and to other metastatic locations (e.g., increase in TP53 mutation frequency, TP53 lymph node metastasis (28/33, 83%), local mets (37/62, 59%), for distant mets (38/60 60%) and primary LUAD tumors (13/36, 36%). Importantly, in spite of the extensive changes in their genomic landscape, lymph LUAD metastasis had the LUAD characteristic tumor karyotype, supporting the notion that the tumor karyotype is fixed early in tumor evolution. Nonetheless, independent studies are required to rule out that this observation was exclusively linked to a singular clinical (e.g., treatment regime) and/or life-style (e.g., smoking status) characteristic of LUAD metastatic patients with lymph node biopsies.

As mentioned in the Supp. Note 2, a major limitation of this analysis is the high level of metastatic-location aggregation used. A more refined metastatic location annotation could reveal certain specificities that are otherwise missed when grouping multiple heterogeneous mets into the same group (e.g., different target tissues grouped into the distant groups, grouped local and distant lymph nodes, etc.). This is particularly relevant for the driver landscape analysis.

All these results and the discussion are now part of the Supp. Note 2.

We also mention the most important implications in the manuscript.

SBS1 mutation rate results:

Moreover, this pattern was not explained by differences in tumor genome ploidy (Extended Data Fig.4c), by metastatic biopsy location (Supp. Note 3), was observed in paired primary-metastatic biopsies from breast and kidney renal clear cell carcinomas and rendered consistent patterns when relying on independent unpaired cohorts (see Supp. Note 3).

Discussion:

To conclude, our dataset constitutes a valuable resource that can be leveraged to further study other aspects of tumor evolution, such as genomic differences across metastatic biopsy locations (Supp. Note 2), dedicated analysis for cancer subtypes (Supp. Note 2), genetic immune escape alterations in primary and metastatic tumors⁶⁹ as well as for the development of machine learning tools to foster cancer diagnostics⁷⁰.

There is no doubt this a valuable resource that can be leveraged for further study. However, beyond the resource aspect, there is somewhat limited truly novel findings. And, there is a lack of validation for key results.

I have the following specific comments:

- Sampling bias and sampling technique. are any tumors subject to multi-region sequencing?

While the clonality appears different between primary and metastatic tumors, is there a possibility that this simply reflects size of biopsy? Or other technical factors? For instance, it could be that additional samples would reveal extensive diversity between metastatic regions from the same metastasis, and there is less intermingling of subclones? I think these caveats need to be considered, even if it is not feasible to perform additional sampling.

While we cannot formally exclude an effect of biopsy size in the observed differences in tumor clonality, there are, at least, two critical factors supporting the validity of our observations:

i) There are cancer types where we did not observe such differences in tumor clonality (e.g., ovarian, liposarcoma, colorectal cancer and hepatocellular carcinoma), which would not be the case if the sampling technique would be the main underlying factor.

ii) Our data is very much in agreement with the results of multiple independent studies (see for instance Nguyen B, et al. Cell. 2022 and Hu, Z. et. al Nature Genetics 2020).

Concerning differences across metastatic biopsies. Our dataset includes certain metastatic patients who have multiple tumor biopsies longitudinally collected over the treatment course. In the study published by Joris Van de Haar and colleagues (Nature Medicine 2021), they performed an exhaustive analysis to identify the genomic differences across paired metastatic biopsies in this dataset. Their analysis revealed a strong concordance between the first and the second metastatic biopsies as well as a limited genomic variation between different metastatic locations. They then concluded that a single WGS analysis of a metastatic biopsy is generally sufficient to identify investigational treatment opportunities given the strong genomic similarity across metastatic biopsies in our dataset.

In any case we acknowledge this as part of the study limitations.

This study also faced other limitations. First, primary and metastatic tumor samples were processed using different laboratory workups and sequencing parameters.

- lymph node may be local or not? what is the distinction here?

Lymph nodes metastatic biopsies include any lymph node biopsy. Unfortunately, we could not make the distinction between local lymph nodes and distant lymph nodes because of lack of sufficient clinical annotations for most of the metastatic patients. This is now clarified in Supp. Note 2 and in methods.

- "loss of heterozygosity (LOH) genome fraction in diploid tumors" - why only in diploid tumours?

We originally intended to limit the analysis to tumors that did not undergo WGD because those tumors are, in principle, more prone to stochastically accumulate genomic deletions, which may hinder the comparison of LOH rates. However, following the suggestion by the reviewers we now include all tumor samples in this comparison and subsequently performed a separated comparison for diploid and non-diploid (i.e., WGD) tumors (see response to reviewers #1 and #2 above, and Extended Data Fig. 1c-d).

- many of the primary tumors subsequently metastasize, while others do not. It would considerably strengthen the analysis if the authors could distinguish these two groups. This would be particularly relevant for much of the mutational signature analysis

We agree that this is an interesting point. Therefore, we have conducted an analysis, including mutational signature comparison, for the two cancer types (i.e., prostate and pancreas neuroendocrine) with substantial genomic landscape differences that also included available clinical follow-up of the primary tumors. Please see response to reviewer #2 and Supp. Note 2 for all the details.

- The analysis suggests 'enhanced activity of APOBEC mutagenesis during the progression of advanced tumors'. It would be interesting to explore whether this reflects met-specific mutations or the tumor as a whole. In other words, there are two possible scenarios here, APOBEC tends to be dominant after the metastatic transition, or, alternatively, APOBEC is more active in tumors that metastasize. While evaluation of met-specific mutations would require paired primary-met analysis, the authors could consider comparing mutations during different epochs of tumour development. For instance, if the authors explore early pre-WGD mutations in primary and met tumours are they more similar, compared to post-WGD mutations?

In general terms we agree with reviewers #1 and #2 that this dataset is not optimal to accurately delineate the relative timing of genomic events in tumor evolution. To effectively address these evolutionary questions, paired longitudinal biopsies are better suited.

For this specific question, an analysis of the timing of APOBEC mutations relative to WGD would identify (dis)similarities about their mutation rate/dynamics relative to this genomic event, but not necessarily about their precise timing relative to the metastatic transformation.

In any case, we find compelling that in our analysis splitting primary prostate (PRAD) and pancreas neuroendocrine (PANET) patients by clinical prognosis, APOBEC mutation load was generally higher in tumors with bad prognosis (progression/relapse, PROG/RL) compared to good prognosis tumors (stable/remission, ST/RM), which in both cases were in turn significantly lower than in metastatic tumors. This observation suggests that APOBEC activity could be enhanced over the course of tumorigenesis.

Finally, we believe that dedicated studies can leverage the wealth of this dataset to address cancer genomic specific questions such as the mutational dynamics of APOBEC mutagenesis.

Now part of Supp. Note 2 (raw data in Supp. Table 3):

“The main exception was a moderate enrichment in APOBEC mutational load in primary tumors that progressed (PROG/RL) compared to primary tumors with good prognosis (ST/RM) in both PRAD and PANET (PANET APOBEC median sample exposure PROG/RL=139, ST/RM=40, MET=215; PRAD APOBEC median exposure PROG/RL=111, ST/RM=51, MET=395 in PRAD). Further analysis (ideally relying on paired longitudinal biopsies) are required to unravel whether enhanced APOBEC activity is per-se a driver of late-stage evolution and/or is the natural consequence of richer soil to perform its mutagenic activity (e.g., higher availability of single stranded DNA). The mutational exposure for the rest of operative mutational processes was on the same order of magnitude across both primary groups (see also Supp. Table 3).”

- Do the authors observe SBS1 mutation burden enrichment regardless of the site of metastasis? In general, the authors appear to lump all mets together based on the site of the primary tumour rather than the location of the metastasis. An analysis exploring how the site of metastasis influences the genomic landscape would be interesting.

This is indeed an interesting point. As part of the extensive SBS1-age corrected enrichment validation described in Supp. Note 3 we also considered an analysis splitting metastatic tumors by metastatic biopsy location. Our analysis suggests that the age-corrected SBS1 burden increase was generally observed across multiple biopsy sites (see response to reviewer #1 and Supp. Note 3 for further details).

Distant met biopsy

Lymph node biopsy

Moreover, as mentioned above, we have now performed an exhaustive re-analysis of the genomic landscape of metastatic tumors compared to primaries splitting metastatic tumors by biopsy location (see Supp. Note 2).

-Differential SBS1 mutation rates in primary and metastatic cancers - this seems rather speculative, further orthogonal validation would be beneficial. Indeed, this whole section, while interesting, seems rather preliminary. The authors must ensure there are not additional factors influencing the results - e.g. tumor purity, sequencing coverage etc. And, again, is this an intrinsic feature of metastasis or simply aggressive tumors? Likewise, the differences observed in breast cancer could simply reflect a difference in subtype between primary and met (e.g. ER+ vs. triple negative).

Thanks for the comment. We agree with the reviewers (this point was also raised by reviewer #1 and #2) that this section would benefit from performing extra controls. In the newly created Supp. Note 3, we have performed a thorough evaluation of the SBS1 age-corrected burden enrichment in metastatic tumors compared to primaries, which also includes validation with paired prim-met datasets and independent cohorts.

Concerning the technical aspects mentioned by the reviewer, we have also shown that tumor purity is generally higher in the PCAWG cohort and that the impact of sequencing coverage is substantially lower than the magnitude of the observed differences across stages (see Supp. Note 1). We also believe that if any of these aspects would introduce a systemic bias, we would observe such enrichment across the majority of cancer types, which is not the case. Moreover, our observations are robustly supported by independent cohorts.

Finally, in Extended Data Fig. 4b and the Supp. Note 3 we have also compared the SBS1 mutation burden differences across the main breast cancer subtypes. Our results revealed that three subtypes (i.e., ER+/HER2-, ER-/HER2+ and triple negative breast cancer (TNBC) showed an age-corrected SBS1 mutation burden increase (see response to reviewer #1 and Supp. Note 3). The only exception was ER+/HER2+ breast cancer which did not show a clear enrichment in metastatic samples. However, given the low number of samples for this subtype and their narrow age-distribution we can not formally exclude insufficient statistical power for this subtype (see below).

SBS1 age association (breast cancer subtype)

Consequently, the reported differences were not driven by a unique breast cancer subtype and were consistent across all screened subtypes except for the ER+/HER2+.

-SBS and DBS signatures are compared to signatures on COSMIC and two novel DBS signatures are found - have these signatures been reported in Degasperi 2022 before?

Given concerns raised by reviewer #2 (and we agree) about the lack of solid evidence for the claim of a novel mutational signature/s we decided to downplay this finding in the revised version of the manuscript.

Moreover, we are now sharing the mutational profile and the pairwise similarity of all extracted mutational signatures (<https://zenodo.org/record/7396538> and Supp. Data 1, respectively), including the mutational profiles of DBS_nenovo_1 and DBS_denovo_2, so the reader could use it to further investigate potential etiologies in dedicated studies.

-‘Compared with TMB, the SV analyses revealed a much more widespread effect, with larger increases per metastatic cancer type that affected almost every cancer type studied, indicating that metastatic tumors appear to evolve primarily by genomic changes at the structural level.’ We can’t be certain that tumors evolve in this way because primary and met samples are not paired. The primary samples which subsequently went on to metastasise could have already presented more structural variants. As stated earlier, an analysis which considers primary tumors which certainly do not metastasize may help shed light on the extent to which this is likely true.

The reviewer is right that an alternative explanation could be that primary subclones with higher SV burden/aneuploidy rates being selected for metastatic colonization and/or treatment resistance. In fact, our analysis splitting primary prostate and pancreatic neuroendocrine tumors according to their

prognosis revealed that primary tumors with worse prognosis (i.e., PROG/RL group, higher chances to metastasize and resist to treatment) have higher SV burden compared to more indolent primary tumors (ST/RM group). However, metastatic tumors still showed an increased SV burden compared to both groups (see Supp. Note 2 and below).

We believe that both scenarios are not necessarily incompatible, as primary tumors/subclones with higher aneuploidy rates and SV burden may have higher rates of progression but still primarily evolving at the structural level compared to the point mutation level.

We now rephrase this claim to acknowledge both explanations:

“Compared with TMB, the SV analyses revealed a much more widespread effect, with larger increases per metastatic cancer type that affected almost every cancer type studied. Moreover, we observed an increased SV burden in prostate and pancreatic neuroendocrine primary tumors that eventually progressed compared to those with relatively better prognosis, which in both cases were in turn lower than the median values in metastatic tumors (see Supp. Note 2). Further studies are required to elucidate whether such patterns are the footprints of underlying evolution via structural variation or reflect a preference towards selection of primary subclones with higher SV burden.”

- Figure 2c-e: Where there is an enrichment of a signature in primary/metastasis it would be good to see the proportions of signatures in primary/metastasis respectively to be able to tell whether one signature is enriched in say the primary cohort which other signature becomes prevalent in the metastasis samples

This is a point also raised by reviewer #2 (see full response above).

Briefly, in our opinion, an analysis based on relative mutational signature exposure is complementary to the absolute exposure comparison. However, this analysis also entails important caveats introduced by the high relative dominance of metastatic-exclusive mutational processes (e.g., treatment associated mutations), which could mask the contribution of other mutational processes already present in primary tumors.

The results and the discussion of this analysis can be found the response to reviewer #2 and also included in the new Extended Data Fig. 3, in the Supp. Table 3 and mentioned in the main text:

Expectedly, analysis based on relative mutational signature contribution revealed the importance of treatment associated mutations in advanced stages of tumor evolution, where platinum and radiotherapy associated mutations and indels showed a significant relative enrichment across several metastatic cancer types (Extended Data Fig. 3).

- The analysis of treatment related signatures and treatment related mutations requires further work. For instance, Figure 6D compares the mutation distribution in EGFR in those tumours receiving treatment for Anti-EGFR and those untreated. Is this a meaningful comparison? Those receiving treatment for anti-EGFR will necessarily harbor activating EGFR mutations or else they would not be eligible for treatment. Thus, what is the main message here? It ends up being a somewhat circular argument. While this analysis makes sense in the context of pre-treatment and exploration of treatment resistance, I don't believe it does without further exploration. Thus, the authors should

seek to distinguish between TEDs which are treatment induces (or selected as a result of treatment) and those which are simply the targets for treatment.

We thank the reviewer for raising this point. It is true that certain targeted treatments, such as Anti-EGFRs or Anti-BRAF, are only given in the presence of specific alterations (i.e., , EGFR-L858R and BRAF-V600E mutations). But it is widely known that secondary drug target mutations (or other somatic variants such as copy number gains) play a critical role in resistance to targeted therapies. For instance, EGFR-T790M and EGFR-C797S mutations are the main mechanisms of resistance to EGFR TKIs (DOI:10.1073/pnas.0709662105, DOI:10.1158/1078-0432.CCR-15-0560). In fact, in our LUAD metastatic cohort there is a broad representation of EGFR resistance variants (T790M and G719S, among others see Fig. 6d). Therefore, even in those cases, identification of higher mutation prevalence may indicate an enrichment in secondary resistance mutations

In order to share this information we now include a column with the treatment-target information for all identified TEDs in the Supp. Table 8.

-Related to the above, there is a need to ensure the analysis is not influenced by cancer subtypes. E.g. anti-EGFR therapy will be given in the context of LUAD, not LUSC. Thus simply considering NSCLC may confound the results and interpretation.

We now considered LUAD and LUSC as two separate cancer types, therefore all analyses are now performed independently.

-Figure 4 - I find this a rather confusing figure (in particular b-f). The colors of the labels overlap with the colors in the enrichment scale. And, I assume the 'Association with higher SV burden arrow' is a typo? In this case further separation of tumour subtypes would also be useful.

We have now changed the color of the feature group labels to avoid the overlap with the colors from the primary-met enrichment.

We have also corrected the typo.

Concerning the subtypes, as we have explained before we have made a substantial effort to harmonize as much as possible the composition of primary and metastatic cohorts used in the main manuscript.

Moreover, given the importance of genomic features in this analysis, we now rely on cancer subtype annotations for breast, colon and uterus carcinomas (see updated Fig. 4b-f and Extended Data Fig. 5).

-Figure 5. For driver comparison it would be useful to ensure the differences do not simply reflect higher number of mutations; i.e. increased likelihood of observing a driver alteration? Again, separating different subtypes would be useful here. For instance, the colorectal samples have a higher mutation burden in Primary tumors. This likely reflects an enrichment for MSI tumors which don't metastasize. This would be worth evaluating.

An increase tumor mutation burden does not necessarily translate into higher number of driver alterations when properly controlling for the background mutation rate (see DOI:10.1038/s41568-020-0290-x, DOI: 10.1038/s41586-021-03771-1, DOI: 10.1101/190330). Specifically, our methodology controls for the TMB to assign the likelihood of a mutation to be driver (see <https://github.com/hartwigmedical/hmftools/blob/master/purple/DriverCatalog.md>). Moreover, it is important to remark that the total number of driver alterations per sample also include CNV drivers and fusion derived driver alterations.

Whether an increased TMB/SV burden may be associated with higher driver burden in certain cancer types (e.g., prostate carcinoma) is a kind of “chicken and egg” thing. It is possible that the acquisition of certain somatic alterations (e.g., TP53/CDK12/BRCA2) may lead to higher mutation and SV burden, but it could also be that somatic driver mutations in these genes are caused by higher background mutation rates. We believe that with the current setting it is impossible to delineate the relative timing of such events in tumor evolution.

Following the suggestion by the reviewers, in the revised version of the manuscript we have made an effort to harmonize the characteristics of the patients included in the primary and metastatic cohorts (see above). Moreover, we now perform a driver-per-sample comparison also splitting by MSI/MSS subtypes in colorectal and uterus carcinoma.

In MSS colorectal cancers the reported number of drivers-per-sample are highly concordant between primary and metastatic tumors, which was in turn highly concordant with the aggregated cancer type comparison (see below). Conversely, MSI/POLE colorectal cancer primary patients displayed a higher average number of drivers per sample compared to metastatic MSI/POLE metastatic patients. Such increase may be reflective of higher intra-tumor heterogeneity in the primary tumors (see Supp. Note 2), although results should be taken with caution due to the low sample size and the lack of statistical significance. Uterus carcinoma displayed a non-significant and comparable distribution of number of driver events per sample in both the MSI/POLE and the MSS groups (see below and Supp. Note 2 tumor clonality comparison and driver landscape comparison sections for all the details).

-Figure 7. While I think it would be useful to have a summary Figure, I don't find this figure very helpful at all. I find it hard to compare cancer types, and distinguishing strong and moderate is not easy.

We agree that it is difficult to draw a threshold between strong and moderate (similar for the "minimal" category). Similarly, it is difficult to compare the weight or relevance of different genomic features in the global comparison (e.g., are the TMB and TEDs equally relevant?). Therefore we have simplified our summary section by only mentioning the five cancer types with the most remarkable genomic differences (prostate, thyroid, breast, pancreatic neuroendocrine and kidney renal clear cell carcinomas). Consequently, we have moved this panel into the Extended Data Figure 9.

Updated text:

However, the magnitude of genomic differences between primary and metastatic tumors is highly cancer type specific and is influenced by the exposure to cancer treatments. Overall, five cancer types (prostate, thyroid, kidney renal clear cell, breast and pancreas neuroendocrine carcinomas) showed an intense transformation of the genomic landscape in advanced tumorigenic stages (Extended Data Fig.9, labeled as Strong). The rest of cancer types displayed variable genomic differences, although the global genomic portrait tended to be more conserved.

- Evaluating the correlation in purity between PCAWG and Hartwig it is notable that there are a number of samples with 100% purity. Can the authors confirm that the VAF of mutations in these samples is consistent with this estimation?

There are 29 whitelisted tumor samples with ~100% estimated tumor purity in our cohort, The aggregated VAF distribution of likely-clonal somatic variants displayed a bimodal distribution peaking around 0.5 VAF (i.e., heterozygous variants) and around 1.0 VAF (i.e., homozygous variants). This is very much in agreement with the expected distribution for clonal somatic variants in a tumor sample with ~100% tumor content.

-it is only mentioned in the discussion that primary and met samples are not paired, while it is also not stated otherwise before this might lead to misinterpretation

Thanks for the comment. We now explicitly mention the nature of our dataset in the manuscript introduction.

Importantly, this dataset encompasses unpaired primary and metastatic tumor samples from 71 cancer types, including 23 cancer types with large representation from both clinical stages.

Reviewer Reports on the First Revision:

Referees' comments:

Referee #1 (Remarks to the Author):

The authors have substantially revised their manuscript, and have fully addressed all my comments, as well as the comments of the other reviewers. The revised study is a clear tour-de-force and will be a very important resource for the community. As such, I am happy to recommend publication.

Minor issues I could find:

1. Figures in supplementary notes 2 and 3 would benefit from figure legends. In addition, the 3 figures on p. 27-28 of supplementary note 2 are too small to read
2. Typo in Extended Data Figure 7a: hepatocellular carcinoma: "+1.5" -> "x1.5"
3. Extended Data Figure 8a/b lacks a color legend

Referee #2 (Remarks to the Author):

I appreciate the authors attempt to leverage public data to confirm many of their findings – especially given the paucity of paired primary/metastatic samples in the literature. The rebuttal letter is detailed and clearly a large body of work in its own right. Overall most of the broad conclusions in the manuscript seem generally supported, including by prior data as the authors point out in response to some reviewer comments. I also thank the authors for moving away from strong conclusions with regards to evolution between the primary and metastatic niche: needed given the absence of patient-matched samples in their dataset.

My original personal stance (in my initial review) on the level of conceptual advancement offered by this large reanalysis, as well as its broader potential utility as a public resource remains largely the same. I feel that the cohort is perhaps not appropriately designed to reveal nuanced insight into treatment resistance: this can probably only come from analysis of paired samples in clinically-standardised contexts. To some extent I think this stance is partly validated in the fact that most of the key results have been reported in prior pan-cancer studies (and therefore may reflect a ceiling for discovery with this type of dataset) – including prior work by the same authors, Priestly et al, Nature 2019 PMID: 31645765. This is particularly true for the broad descriptions of genomic features (TMB, LST, LOH, WGD rates, arm-level karyotype, enrichment of SVs in metastatic samples, etc.), identification of known treatment-resistance alterations (e.g. ESR1 mutation in endocrine-resistant breast cancer; AR amplifications and mutations in prostate cancer progressing on AR-targeted therapies, etc), as well as the link between specific mutation signatures and platinum or radiotherapy exposure (e.g. Alexandrov et al, Nature 2020, PMID: 32025018; Pleasance et al, Nat Cancer 2020, PMID: 35121966). However I totally recognize the general points made by the other reviewers and the authors about the value of this manuscript, and of course it is absolutely of key importance to see prior hypotheses validated in such a large / extensive dataset.

I have some remaining technical concerns, as well as some minor phrasing adjustments, all of which

should be addressed if possible.

1. I thank the authors for recognizing the limitation of the lack of patient-matched samples in the introduction and discussion. However, given the importance of this detail for reader interpretation, this keyword (i.e. unmatched / unpaired, or similar) should also be mentioned in the title and/or abstract.

2. The authors should carefully remove all causal language where it is not appropriate – since most analyses are strictly correlative – as well as be more precise about measured versus inferred quantities. Some examples:

○ Line 172: “... we found that mutations *caused by* cytotoxic treatments were significantly enriched in 10 cancer types”

○ Line 179-179: “Likewise, median mutation contribution from the radiotherapy ID

○ signature (ID8) was systematically enriched in multiple cancer types *as a response* to widespread exposure to radiation-based treatment.”

○ Line 213-214: “... *rates of SBS1 mutation acquisition* were highly tissue specific” – the authors are not directly measuring mutation acquisition rates.

3. Some sections of the Supplementary Material are still somewhat unclear/ambiguous. I found it difficult to follow precisely how analyses had been performed and what quantities were being compared (e.g. page 13 of Supplementary Note 1, discussion of clonality and variant calling; page 8 of Supplementary Note 1, discussion of SAGE mutation calling and parameter adjustments). Please consider editing for clarity/readability.

4. Line 283-284: “We also found potential novel associations, such as large deletions linked to the chromatin regulators SETD2 and CHD1.” Are the authors referring to an association between deletions to SETD2/CHD1 and an enrichment of SVs (i.e. language is unclear)? If so, this has been described previously (Baca et al, Cell 2013, PMID: 23622249) in the context of CHD1 and chromoplexy in prostate cancer. Similar observations have also been previously observed with SETD2 in CLL (Parker et al, Leukemia 2016, PMID: 27282254).

5. There is some published evidence suggesting that the evolutionary bottleneck for lymph node metastasis (which is often an early/initial dissemination event in progression to metastatic disease) is more permissive compared to the colonisation of distant organs (e.g. Reiter et al Nat Genetics 2020, PMID: 32451459). This difference in evolutionary selective pressures has been linked to greater LN genomic heterogeneity (versus distant organ sites). In contrast the authors’ data (e.g. Figure 1C) suggest that metastatic LN are fairly similar to distant metastatic sites (in terms of overall clonality). It would be interesting for the authors to put their results in context of some of these prior findings. Can this be explained by location of biopsied LN (i.e. locoregional vs. distant)? Tumor-specific features of metastatic dissemination? etc.

6. Minor: there is a stray dot (datapoint?) at the bottom of Figure 6a.

7. I agree that the apparent increase in metastatic SBS1 mutation rate is likely a collider variable causally linked to both higher cell proliferation rates (compared to primary disease) and/or newly

acquired deficiencies in DNA repair. However I am still concerned that the 2-3x difference in sequencing depth between the Hartwig (metastatic) and PCAWG (primary) samples may be contributing to some of this difference in Figure 3 (in addition to other primary-metastatic comparisons). The dramatically lower sequencing depth of primary samples will result in lower sensitivity for subclonal mutations, and the Supplementary analyses showing comparable bulk/mean effect sizes for Hartwig samples with downsampled coverage does not fully clarify this concern. At minimum I would like to see these Figures replotted using SBS1 mutation rate (i.e. SBS1 mutation count divided by total sample mutation burden) rather than absolute count, to accommodate potential differences in global mutation rate between primary and metastatic samples. At the very least this will simplify readers' experience (in addressing an obvious question/concern upfront), rather than requiring readers to cross-reference with the lengthy supplementary material.

8. In reading the Supplementary material, it was not clear to me whether the relationship between SNV detection sensitivity, SNV clonality, and sequencing depth was fully elucidated. For example, in downsampling the 109x Hartwig samples to 38x (the lower of the modal depths for primary PCAWG samples), the authors noted a 26% drop in SNV sensitivity, but appeared to mitigate this loss by lowering the minimum QC – i.e. dropping specificity to compensate (although the exact approach here was not clearly described). After this minimum QC adjustment the authors see similar mean SNV, DBS etc burden between the downsampled and original 109x samples. However the experiment I would have expected to see is the proportion of mutations originally called in the 109x samples that are recapitulated in the downsampled samples, stratified by their original CCF (in the 109x samples). The authors touch on this concept by showing their adjusted mutation calling thresholds for low-coverage samples recapitulate ~95% of n=172 driver mutations in the original 109x samples (Supplementary Note 1 page 9), but of course driver mutations are highly likely to be clonal/initiating events, and therefore this analysis doesn't inform on subclonal mutation sensitivity (with clonality being a major focus of Figure 3). There is a line in Supplementary Note 1 "Given the consistent results in passenger and driver mutation detection over the tested sequencing depth modes..." but I do not see the former actually tested.

9. Relatedly, this difference in metastatic versus primary sequencing depth is liable to underlie some of the other results. For example, the observation that there is an average increase of ~1 driver mutation in metastatic compared to primary cancers (i.e. Line 300: an increase of 4.5 to 5.3 in mean driver mutation count) seems broadly consistent with the observation that only ~94-95% of 109x Hartwig driver mutations were redetected in their 38x downsampled counterparts (Supplementary Note 1 page 9) (i.e. 4.5/5.3 is 85%). And of course these benchmarking experiments were only conducted in 25 of 4,784 (i.e. 0.5%) samples, so the generalizability of many of these benchmarking experiments to the whole cohort is not demonstrated. I am not asking for any specific changes here, but I wonder if the authors feel that this point is valid and worth addressing / mentioning.

10. I would like to hear the authors' thoughts on their observation of increased SBS1 mutation rate in later tumorigenic stages when one may expect it to be supplanted by other mutational processes later in disease? Relatedly, an interesting comparator for this finding would be to look at the serial metastatic-only biopsies also included in the Hartwig dataset (i.e. van de Haar et al, 2021 Nat Medicine, PMID: 34373653). Are there opportunities for deeper analysis here?

11. The authors should check whether there is a difference in SBS1 mutation count (rate) between good prognosis and poor prognosis primary tumors. Are the latter more similar to the age-adjusted SBS1 mutation rate for metastatic tumors? I would expect there to be such a difference for primary cancers that are frequently cured by effective localised treatment (i.e. never become metastatic or were never on a metastatic trajectory).

12. Please annotate the Pearson R value for metastatic and primary categories for all plots in Extended Data Figure 4.

13. It looks like there was a code error in generating panels for leiomyosarcoma and liposarcoma in Extended Data Figure 4a. Similar errors for thyroid and ovarian cancer in Supp Note 3 page 4; thyroid Supp Note 3 page 8. If no data is available for a given cancer type please annotate that directly on the Figure.

14. Minor semantic issue in manuscript line 215: "... and SBS1 mutation burden increased linearly with age in the majority of cancer types in both primary and metastatic cohorts" – the claim of linearity seems to be an assumption rather than a result (i.e. by choosing to quantify via linear regression). Did the authors actually test other models (e.g. exponential) to prove that a linear model most appropriately explains their data? Perhaps the authors can remove the word 'linear' when describing these results (and/or consider visualising these correlations with a nonparametric approach, e.g. KNN). Relatedly, Pearson correlation (to measure the degree of linear correlation between two variables) can also be applied to nonlinear data (i.e. does not 'prove' linearity).

Referee #3 (Remarks to the Author):

Overall, the authors have done a good job of addressing my comments. While I still think this is an interesting analysis - I would tend to agree with Reviewer 2 that there are no striking novel findings, and the caveats given the lack of paired samples are considerable. Indeed, the new analysis in response to the reviewers comments highlights that if appropriate controls are not used conclusions can be accidentally overstated.

I have a few either additional comments or points where I feel still more analysis could be done.

- APOBEC timing. I still think it would be worth exploring APOBEC pre and post WGD.

- I still think it would be worth exploring whether the signatures are consistent with those reported in Degasperi 2022. While I appreciate that readers could explore the data independently, it would be useful to include that information here.

- I still think the comparison for the treatment induced changes is misleading. Ideally the authors should consider the same population with and without treatment; e.g. a population of patients with EGFR mutant tumours, and then explore the impact of treatment. I still think this section is confusing

otherwise.

- I appreciate the authors further exploration of SBS1 in metastatic samples. Could the authors also further evaluate the relationship when restricting to clonal/subclonal pre and post WGD mutations?

- I would avoid the term 'metastatic patients' - it is the cancer cells that are subject to metastasis not the patient.

Author Rebuttals to First Revision:

Responses to referees

Changes in main text and supplementary notes

Referees' comments:

Referee #1 (Remarks to the Author):

The authors have substantially revised their manuscript, and have fully addressed all my comments, as well as the comments of the other reviewers. The revised study is a clear tour-de-force and will be a very important resource for the community. As such, I am happy to recommend publication.

Thanks. We appreciate the comments and suggestions raised by the reviewer as they have contributed to substantially improve the manuscript's quality.

Minor issues I could find:

1. Figures in supplementary notes 2 and 3 would benefit from figure legends. In addition, the 3 figures on p. 27-28 of supplementary note 2 are too small to read

Figure legends have been added to the Supp. Notes 2 and 3. Also we have increased the font size and quality of figures in these pages.

2. Typo in Extended Data Figure 7a: hepatocellular carcinoma: "+1.5" -> "x1.5"

Corrected. Thanks

3. Extended Data Figure 8a/b lacks a color legend

Corrected. Thanks

Referee #2 (Remarks to the Author):

I appreciate the authors attempt to leverage public data to confirm many of their findings – especially given the paucity of paired primary/metastatic samples in the literature. The rebuttal letter is detailed and clearly a large body of work in its own right. Overall most of the broad conclusions in the manuscript seem generally supported, including by prior data as the authors point out in response to some reviewer comments. I also thank the authors for moving away from strong conclusions with regards to evolution between the primary and metastatic niche: needed given the absence of patient-matched samples in their dataset.

My original personal stance (in my initial review) on the level of conceptual advancement offered by this large reanalysis, as well as its broader potential utility as a public resource remains largely the same. I feel that the cohort is perhaps not appropriately designed to reveal nuanced insight into treatment resistance: this can probably only come from analysis of paired samples in clinically-standardised contexts. To some extent I think this stance is partly validated in the fact that most of the key results have been reported in prior pan-cancer studies (and therefore may reflect a ceiling for discovery with this type of dataset) – including prior work by the same authors, Priestly et al, Nature 2019 PMID: 31645765. This is particularly true for the broad descriptions of genomic features (TMB, LST, LOH, WGD rates, arm-level karyotype, enrichment of SVs in metastatic samples, etc.), identification of known treatment-resistance alterations (e.g. ESR1 mutation in endocrine-resistant breast cancer; AR amplifications and mutations in prostate cancer progressing on AR-targeted therapies, etc), as well as the link between specific mutation signatures and platinum or radiotherapy exposure (e.g. Alexandrov et al, Nature 2020, PMID: 32025018; Pleasance et al, Nat Cancer 2020, PMID: 35121966). However I totally recognize the general points made by the other reviewers and the authors about the value of this manuscript, and of course it is absolutely of key importance to see prior hypotheses validated in such a large / extensive dataset.

We are glad that the reviewer/s values our efforts to address their previous concerns. We also appreciate the thoroughness and technical soundness of their comments as they have contributed to improve the manuscript quality.

I have some remaining technical concerns, as well as some minor phrasing adjustments, all of which should be addressed if possible.

1. I thank the authors for recognizing the limitation of the lack of patient-matched samples in the introduction and discussion. However, given the importance of this detail for reader interpretation, this keyword (i.e. unmatched / unpaired, or similar) should also be mentioned in the title and/or abstract.

We agree that this is an essential aspect for the reader's interpretation. Therefore we have included a clarifying sentence at the beginning of the manuscript abstract:

Here, we characterize the genomic differences between early-stage untreated primary tumors and late-stage treated metastatic tumors using a harmonized pan-cancer (re-)analysis of two unpaired primary and metastatic cohorts of 7,108 whole-genome-sequenced (WGS) tumors.

2. The authors should carefully remove all causal language where it is not appropriate – since most analyses are strictly correlative – as well as be more precise about measured versus inferred quantities. Some examples:

o Line 172: "... we found that mutations *caused by* cytotoxic treatments were significantly enriched in 10 cancer types"

o Line 179-179: "Likewise, median mutation contribution from the radiotherapy ID

o signature (ID8) was systematically enriched in multiple cancer types *as a response* to widespread exposure to radiation-based treatment."

o Line 213-214: "... *rates of SBS1 mutation acquisition* were highly tissue specific" – the authors are not directly measuring mutation acquisition rates.

Although multiple studies (including some from our lab, see for example DOI: 10.1038/s41467-019-12594-8) have clarified the causal link between some of these mutational signatures and their underlying triggers both *in-vitro* and *in-vivo* it is true that this study only leverages correlations/associations. Therefore we have adjusted the text to better describe the nature of these associations. Thanks.

3. Some sections of the Supplementary Material are still somewhat unclear/ambiguous. I found it difficult to follow precisely how analyses had been performed and what quantities were being compared (e.g. page 13 of Supplementary Note 1, discussion of clonality and variant calling; page 8 of Supplementary Note 1, discussion of SAGE mutation calling and parameter adjustments). Please consider editing for clarity/readability.

We have now rewritten the introduction of section "Impact of sequencing depth coverage and analysis pipeline on somatic variant calling sensitivity" of Supp. Note 1 to better explain the rationale of the downsample analysis on 25 Hartwig samples. We also revised the text related to clonality and variant

calling and improved the readability on parameter adjustments. The sample quantities on the analysis we performed on page 13 are now depicted at the top of Figure 5 in Supp Note 1.

4. Line 283-284: “We also found potential novel associations, such as large deletions linked to the chromatin regulators SETD2 and CHD1.” Are the authors referring to an association between deletions to SETD2/CHD1 and an enrichment of SVs (i.e. language is unclear)? If so, this has been described previously (Baca et al, Cell 2013, PMID: 23622249) in the context of CHD1 and chromoplexy in prostate cancer. Similar observations have also been previously observed with SETD2 in CLL (Parker et al, Leukemia 2016, PMID: 27282254).

We have clarified the language and cited the references in the main text. Thanks.

5. There is some published evidence suggesting that the evolutionary bottleneck for lymph node metastasis (which is often an early/initial dissemination event in progression to metastatic disease) is more permissive compared to the colonisation of distant organs (e.g. Reiter et al Nat Genetics 2020, PMID: 32451459). This difference in evolutionary selective pressures has been linked to greater LN genomic heterogeneity (versus distant organ sites). In contrast the authors’ data (e.g. Figure 1C) suggest that metastatic LN are fairly similar to distant metastatic sites (in terms of overall clonality). It would be interesting for the authors to put their results in context of some of these prior findings. Can this be explained by location of biopsied LN (i.e. locoregional vs. distant)? Tumor-specific features of metastatic dissemination? etc.

Interesting point. First, it is important to mention that, unfortunately, in the Hartwig dataset we do not have harmonized annotation of the type biopsied LN (locoregional or distant), which may undergo different colonization patterns. Second, while LN biopsied tended to show lower intra-tumor heterogeneity than primary tumors and that local metastatic biopsies, for several of cancer types LN metastatic biopsies also displayed slightly lower median clonality than distant metastatic biopsies (e.g., breast and esophageal carcinomas Fig. 1b, kidney renal clear cell, lung adeno, prostate carcinoma Extended Data Fig. 1b), although it is true that this is not observed for other cancer types such as colorectal carcinoma and skin melanoma. Considering the prevailing knowledge on LN metastatic dissemination, a plausible explanation is that the Hartwig cohort is enriched in distant LN biopsies, but this is currently impossible to be formally assessed in this dataset. We have now included a discussion of our findings in the context of current knowledge for LN metastatic dissemination (see Supp. Note 2).

“It is also important to mention that in the Hartwig dataset, LN biopsies, which have been reported to undergo wider evolutionary bottleneck than distant metastases (Reiter et al. 2020) (i.e., higher intra-tumor heterogeneity), only displayed a modest and non-homogeneous decrease in tumor clonality

compared to distant metastasis (see breast and esophageal carcinomas Fig. 1b and kidney renal clear cell, lung adeno, prostate carcinoma in Extended Data Fig. 1b). The lack of significant differences between LN and distant biopsies could be explained by an enrichment of distant LN biopsies (i.e., non locoregional), but this is currently impossible to be formally assessed given the lack of harmonized clinical annotation for this information in the Hartwig dataset.”

As a side note, we observed that lung adenocarcinoma (and kidney renal clear cell carcinoma to a lower extent) LN biopsies displayed a distinctive genomic profile, compared to local and to distant metastatic biopsies of these cancer types, respectively (see Supp. Note 2 for further information).

6. Minor: there is a stray dot (datapoint?) at the bottom of Figure 6a.

Corrected, thanks.

7. I agree that the apparent increase in metastatic SBS1 mutation rate is likely a collider variable causally linked to both higher cell proliferation rates (compared to primary disease) and/or newly acquired deficiencies in DNA repair. However I am still concerned that the 2-3x difference in sequencing depth between the Hartwig (metastatic) and PCAWG (primary) samples may be contributing to some of this difference in Figure 3 (in addition to other primary-metastatic comparisons). The dramatically lower sequencing depth of primary samples will result in lower sensitivity for subclonal mutations, and the Supplementary analyses showing comparable bulk/mean effect sizes for Hartwig samples with downsampled coverage does not fully clarify this concern.

Of course we agree that it is impossible to fully mitigate the effect of different sequencing coverages in somatic detection sensitivity, particularly for subclonal variants (see response 8. below). However, we have shown that:

i) This observation is independently reproduced when relying on independent datasets, for which there is not such detection sensitivity difference (Supp. Note 3)

ii) The five metastatic prostate cancer samples processed from the PCAWG cohort did show an age-adjusted increase in SBS1 burden that perfectly matched with the expected given by the linear regression of the metastatic tumors from Hartwig (Supp Note 3, Fig. 6), these samples have similar sequencing coverage and sequencing parameters than the primary PCAWG samples.

iii) This observation is highly cancer type specific. Moreover, the cancer types with increased SBS1 mutation burden are not those with the overall lowest sequencing coverage in the PCAWG dataset.

iv) When narrowing down the analysis to SBS1-clonal mutation burden (which are in principle less affected by differences in sequencing coverage, see response 8. below) we still observe a very consistent pattern (see image below, now part of Supp. Note 3).

SBS1 clonal only mutations

In any case, we now stress the limitations associated with differences in sequencing coverages of both cohorts in the manuscript discussion.

Moreover, sequencing depth of the primary tumor cohort was lower and more variable than that of the metastatic tumor cohort, although we demonstrated that this does not severely impact on the overall detectability of clonal somatic variants (Supp. Note 1). However, we cannot formally exclude the possibility of missing subclonal (driver) mutations for samples that were sequenced with lower depth.

At minimum I would like to see these Figures replotted using SBS1 mutation rate (i.e. SBS1 mutation count divided by total sample mutation burden) rather than absolute count, to accommodate potential differences in global mutation rate between primary and metastatic samples. At the very least this will simplify readers' experience (in addressing an obvious question/concern upfront), rather than requiring readers to cross-reference with the lengthy supplementary material.

While we perfectly understand the point of the reviewer we previously discussed in the point by point responses to reviewer #3 and to this reviewer the caveats of relying on relative mutational exposure (i.e., number of SBS1 mutations divided by the TMB) to make a comparison between primary and metastatic SBS1 mutation rate. To recapitulate, to render relative SBS1 exposures comparable, one would need to assume that the activity of other mutational processes are steady between the two groups, which is clearly not the case. Specifically, an increase in SBS1 mutation burden may be masked by the accumulation of metastatic-exclusive (e.g., mutations associated with cytotoxic therapies) and metastatic-enriched (e.g., APOBEC) mutational processes (which in some cancer types have an average contribution of hundreds/thousands of mutations per sample).

Alternatively, instead of using the TMB as baseline, one could rely on the SBS5/SBS40 mutation burden as a normalization factor, since it is widely present across cancer types and it generally displays similar mutation dynamics in both stages. The figure below shows the cancer type specific SBS1 mutation rate (normalized by SBS5/40 mutation burden) comparison between primary and metastatic tumors. Except for thyroid carcinoma, the cancer types displaying an age-corrected increased metastatic SBS1 mutation burden show an increased SBS1 mutation rate, whereas the two controls do not show any significant difference. Overall, this is in line with an increased SBS1 mutation burden that is not observed for cell-division independent and age-correlated mutational processes (i.e., SBS5/40).

8. In reading the Supplementary material, it was not clear to me whether the relationship between SNV detection sensitivity, SNV clonality, and sequencing depth was fully elucidated. For example, in downsampling the 109x Hartwig samples to 38x (the lower of the modal depths for primary PCAWG samples), the authors noted a 26% drop in SNV sensitivity, but appeared to mitigate this loss by lowering the minimum QC – i.e. dropping specificity to compensate (although the exact approach here was not clearly described). After this minimum QC adjustment the authors see similar mean SNV, DBS etc burden between the downsampled and original 109x samples. However the

experiment I would have expected to see is the proportion of mutations originally called in the 109x samples that are recapitulated in the downsampled samples, stratified by their original CCF (in the 109x samples). The authors touch on this concept by showing their adjusted mutation calling thresholds for low-coverage samples recapitulate ~95% of n=172 driver mutations in the original 109x samples (Supplementary Note 1 page 9), but of course driver mutations are highly likely to be clonal/initiating events, and therefore this analysis doesn't inform on subclonal mutation sensitivity (with clonality being a major focus of Figure 3). There is a line in Supplementary Note 1 "Given the consistent results in passenger and driver mutation detection over the tested sequencing depth modes..." but I do not see the former actually tested.

This is a good suggestion from the reviewer. We have added this analysis in Supp. Note 1 as follows:

To assess the mutation recall rate across the three tested sequence depths, we calculated the fraction of somatic mutation overlap between 38X & 109X, and 60X & 109X in the 25 downsampled Hartwig samples. Subsequently, we binned the mutation based on its purity adjusted VAF score of the original 109X sample.

Overall, we found consistent results. Mutations with VAF scores higher than 0.3 (4th bin) displayed a minimal recall rate of 96% for 38X sequencing depth and 98% for the 60X sequence mode. This percentage dropped marginally for the DBS (recall rate of at least 96% and 96% at 38X and 60X respectively), and indel (recall rate of at least 89% and 94% at 38X and 60X respectively) mutation types, however these mutation types were not included in any of the the clonality ratio measurements. Mutations with a VAF lower than 0.3 (1st to 3rd bins) showed SNV recall rates of 93% ($0.2 < \text{VAF} < 0.3$) and 84% ($0.1 < \text{VAF} < 0.2$) at 38X sequence depth, and 97% ($0.2 < \text{VAF} < 0.3$) and 91%

($0.1 < \text{VAF} < 0.20$) at 60X sequence depth. However, low VAF mutations marginally contribute to the TMB for SNV, DBS and INDELS as depicted by the barplots on top.

In summary, our results indicate that the SNVs recall rates are higher than 95% for the vast majority of mutations and that only subclonal mutations with really low VAF display an noticeable sensitivity loss in tumor samples with lower sequencing coverage (i.e., ~10% sensitivity loss for 60X and ~15% for 38X).

9. Relatedly, this difference in metastatic versus primary sequencing depth is liable to underlie some of the other results. For example, the observation that there is an average increase of ~1 driver mutation in metastatic compared to primary cancers (i.e. Line 300: an increase of 4.5 to 5.3 in mean driver mutation count) seems broadly consistent with the observation that only ~94-95% of 109x Hartwig driver mutations were redetected in their 38x downsampled counterparts (Supplementary Note 1 page 9) (i.e. 4.5/5.3 is 85%). And of course these benchmarking experiments were only conducted in 25 of 4,784 (i.e. 0.5%) samples, so the generalizability of many of these benchmarking experiments to the whole cohort is not demonstrated. I am not asking for any specific changes here, but I wonder if the authors feel that this point is valid and worth addressing / mentioning.

As previously discussed, we agree with the reviewer that lower sequencing depths undoubtedly hinders mutation detection sensitivity for subclonal variants. This is also the main reason why we extensively validated the impact lower sequencing depths on variant calling as delineated in Suppl Note 1.

However, we believe the impact on driver identification is minimal for the following reasons:

1. The PCAWG consortium has studied many facets of tumorigenesis, including the driver panorama. These studies have shown that most cancer type specific driver genes were already reported in previous studies (such as high read depth WES from TCGA), showing that driver mutation detection from WGS samples at lower read depth can be executed. This stems from the rationale that driver mutations are typically highly clonal and emerge at early stages of tumorigenesis.
2. While it is true the PCAWG study include tumor samples sequenced at 38X, a large fraction of tumor samples were sequenced at higher sequenced coverages (60X and 90X), for which the sensitivity to detect clonal (and subclonal) variants is substantially higher (see previous point).
3. A substantial fraction of the driver genes that were found to be enriched in the Hartwig cohort, could be linked to treatment resistance pathways (Main Fig 5b). Moreover, our drug-specific analysis shows that a large fraction of TEDs are treatment-exclusive indicating that

mutations in the TED gene were not found in any sample from the respective cancer type of the PCAWG cohort. This suggests that driver detection is not seriously biased towards samples sequenced at higher sequencing depths.

We do of course agree that cohorts sequenced with equal read depth distributions would yield more fair information in comparison studies, especially on subclonal driver variants. The limitations of the study design such as varying sequencing depths were already discussed at the end of the paper, but now we have adapted the discussion part to reflect the limitations of sequencing depth regarding driver claims. We have now added the following sentence:

However, we cannot formally exclude the possibility of missing subclonal (driver) mutations for samples that were sequenced with lower depth.

10. I would like to hear the authors' thoughts on their observation of increased SBS1 mutation rate in later tumorigenic stages when one may expect it to be supplanted by other mutational processes later in disease? Relatedly, an interesting comparator for this finding would be to look at the serial metastatic-only biopsies also included in the Hartwig dataset (i.e. van de Haar et al, 2021 Nat Medicine, PMID: 34373653). Are there opportunities for deeper analysis here?

SBS1 mutations are attributed to unrepaired C>T mismatches caused by deamination of 5-methylcytosine to thymine in CpG islands. Recent literature indicates a strong correlation between the rate of cell division and the SBS1 mutation rate (L.B. Alexandrov et al. Nat. Genetics 2015, DOI:10.1038/ng.3441). SBS1 mutations accumulate both in healthy tissues and in malignant lesions. Rates of SBS1 acquisition in healthy and primary tumors are highly tissue/tumor specific and seem to be primarily dictated by cell of origin (e.g., doi: 10.1038/ng.3441). Recent studies as well as our data indicates that SBS1 mutations keep accumulating along the evolution of the tumor, although its mutation rate might be subjected to changes in later tumorigenic stages, which in turn may be caused by differences in cell division rates as well as changes in repair fidelity, different methylation dynamics, etc. Therefore, while we think that SBS1 mutations keep accumulating (even at increased rates in certain cancer types) in tumor evolution, it is also true that other endogenous and exogenous mutational processes become (more) active in later evolutionary stages (e.g., treatment induced mutations, APOBEC mutagenesis, etc). These late-stage enriched mutational processes that can be highly variable in nature and burden may mask the contribution of SBS1 mutations that are therefore relatively less important compared to earlier stages of tumor evolution (this is the reason why using a relative SBS1 mutation rate comparison is not entirely fair and in our view will be more confusing than using absolute numbers). An interesting follow-up question is whether any of these late-stage mutational processes may also directly interfere with the SBS1 mutation rate (e.g., do mismatch repair deficiency lead to a higher SBS1 mutation burden? Can increased rates of damaged DNA caused by alkylating agents interfere with the fidelity of C>T mismatch repair?). This is something that dedicated

studies can explore and we expect that by releasing this dataset we are providing the means to address this type of questions.

The paired-metastatic tumor biopsies from the van de Haar et al, et al. study were originally analyzed to address the question of whether a single metastatic biopsy was sufficient to detect the burden of actionable alterations for metastatic cancer patients. As a result, they found limited tumor evolution across metastatic biopsies, meaning that a single WGS analysis of a metastatic biopsy is generally sufficient to identify treatment opportunities. Given the scope of the study, these metastatic paired biopsies lack a primary biopsy and were generally collected in a limited amount of time (mean of 6 months between met. biopsies, but some biopsies were simultaneously collected) relatively to the full evolution of the somatic tissue. Considering the aforementioned constraints and the limited number of available met-pairs across cancer types, we think this dataset is not optimal to study SBS1 mutation rate, as this generally requires a longer time window between biopsies. Moreover, such analysis would be difficult to frame within the scope of this study, which focuses on identifying genomic differences between primary and metastatic tumors.

11. The authors should check whether there is a difference in SBS1 mutation count (rate) between good prognosis and poor prognosis primary tumors. Are the latter more similar to the age-adjusted SBS1 mutation rate for metastatic tumors? I would expect there to be such a difference for primary cancers that are frequently cured by effective localised treatment (i.e. never become metastatic or were never on a metastatic trajectory).

Interesting point. As hypothesized by the reviewer, prostate carcinoma and pancreatic neuroendocrine primary tumors with worse prognosis (progression/relapse) have higher age-adjusted SBS1 mutation burden compared to better prognostic tumors (stable/remission group). However, in both cases the yearly SBS1 mutation burden was substantially lower than in metastatic tumors (see figure below).

Our results thus suggest that the yearly SBS1 mutation rate increases as PRAD/PANET tumors progress and that primary tumors with relatively worse prognosis have already acquired higher SBS1 mutation rates than better prognosis ones. This analysis has been integrated into the Supp. Note 2.

12. Please annotate the Pearson R value for metastatic and primary categories for all plots in Extended Data Figure 4.

Pearson correlation coefficient values have been added to Extended Data Figure 4a. This information alongside other statistical measurements of the linear regressions have also been included in Supp. Table 4. Thanks.

13. It looks like there was a code error in generating panels for leiomyosarcoma and liposarcoma in Extended Data Figure 4a. Similar errors for thyroid and ovarian cancer in Supp Note 3 page 4; thyroid

Supp Note 3 page 8. If no data is available for a given cancer type please annotate that directly on the Figure.

Patient's age at biopsy was not available for the sarcoma cohorts of the PCAWG dataset (see also Fig. 1a). This is now explicitly annotated in the figure panel. Thanks

14. Minor semantic issue in manuscript line 215: "... and SBS1 mutation burden increased linearly with age in the majority of cancer types in both primary and metastatic cohorts" – the claim of linearity seems to be an assumption rather than a result (i.e. by choosing to quantify via linear regression). Did the authors actually test other models (e.g. exponential) to prove that a linear model most appropriately explains their data? Perhaps the authors can remove the word 'linear' when describing these results (and/or consider visualising these correlations with a nonparametric approach, e.g. KNN). Relatedly, Pearson correlation (to measure the degree of linear correlation between two variables) can also be applied to nonlinear data (i.e. does not 'prove' linearity).

The reviewer is right that for this analysis linearity is an assumption based on previous studies and on the current knowledge of SBS1 mutation rate dynamics. We have corrected the text following the reviewer's suggestion.

Referee #3 (Remarks to the Author):

Overall, the authors have done a good job of addressing my comments. While I still think this is an interesting analysis - I would tend to agree with Reviewer 2 that there are no striking novel findings, and the caveats given the lack of paired samples are considerable. Indeed, the new analysis in response to the reviewers comments highlights that if appropriate controls are not used conclusions can be accidentally overstated.

We are glad that the reviewer finds the responses to his/her previous concerns satisfying.

I have a few either additional comments or points where I feel still more analysis could be done.

- APOBEC timing. I still think it would be worth exploring APOBEC pre and post WGD.

We used the MutationTimeR(Gerstung et al. 2020) tool to estimate the relative timing of APOBEC mutations across primary and metastatic tumors. From MutationTimeR, we selected all APOBEC mutations in copy-number amplified regions that has been annotated as clonal-early or clonal-late, because only these annotations reflect the relative timing before (clonal-early) or after (clonal-late) the amplification event (see also section “Clonality and timing of SBS1 mutations” in Suppl Note 3). More specifically, we selected mutations, with nucleotide contexts including T[C>T]A, T[C>T]T, T[C>G]A, T[C>G]T, in amplified regions with a minor allele copy number !=1 and with the major allele copy number is >=2. Subsequently, we filtered for samples with PURPLE WGD status == TRUE and with an estimated genome ploidy of 4n to select samples with only 1 round of WGD. Lastly, we excluded genome ploidy regions higher than 5 to mitigate the effect of duplication events unrelated to WGD (e.g. in double minutes, break fusion bridges, etc.).

Then, for each sample, the APOBEC clonal late ratio was defined as the ratio between the proportion of clonal late APOBEC mutations (i.e., likely generated after WGD) divided by the total proportion of APOBEC alterations in the sample (see below), where the total of clonal APOBEC mutations was computed as the sum of clonal late and clonal early mutations from MutationTimeR. The clonal late ratio thus represents the fraction of APOBEC mutations that happened after WGD. Only samples with at least five informative clonal early/late annotated APOBEC mutations in amplified regions were considered. Moreover, cancer types where any of the cohorts (primary or metastatic) included less than five samples were not analyzed.

1. We observed that in certain cancer types, namely breast carcinoma (BRCA), ovarian carcinoma (OV) and colorectal carcinoma (COREAD); the median fraction of clonal late mutations is certainly higher than 50%, indicating most APOBEC attributed mutation load occurred after WGD. Remarkably, the fact that only certain types displayed such a pattern may be related to the relatively early timing of WGD in these cancer types (see Moritz Gerstung et al. 2020, DOI: [10.1038/s41586-019-1907-7](https://doi.org/10.1038/s41586-019-1907-7)), which enabled the accumulation of higher APOBEC mutation burden. However, we think that dedicated studies are needed to better delineate the interplay between WGD and APOBEC mutagenesis in the context of cancer.
2. We observed that most cancer types showed no significant difference in relative clonal late APOBEC mutation proportion between primary and metastatic cancer cohorts. Only metastatic breast carcinomas, and non-small cell lung cancer to a lower extent, showed a significant relative enrichment in clonal late APOBEC mutations compared to primaries. The significant effect in skin melanoma must be interpreted with caution given the possible overlap in mutation contexts from UV exposure.

3. The enrichment in APOBEC mutations in certain metastatic tumors might be associated with the higher WGD rates in those cancer types with clonal late ratio $\geq 50\%$ (see point 1). However, this is unlikely to explain the full extent of the APOBEC mutation load observed across multiple cancer types. We think that other factors, such as the availability of ssDNA and the number of DNA double strand breaks (which potentially result in complex rearrangements that have previously been linked to higher incidence of APOBEC mutations (Bergstrom, E.N. et al 2022, DOI: 10.1038/s41586-022-04398-6) may be an important contributing factor.

- I still think it would be worth exploring whether the signatures are consistent with those reported in Degasperi 2022. While I appreciate that readers could explore the data independently, it would be useful to include that information here.

We have now included a comparison matrix of the *de novo* extracted signatures from this study with the cancer mutational signatures from Signal (<https://signal.mutationalsignatures.com/explore/study/6?mutationType=1>). This table contains the cosine similarity scores between all signatures and is now added to Suppl. Table 3. In general, we observed a very good agreement (i.e., inferred etiology) between the mutational signatures used in this study and those extracted from Signal. Thanks.

- I still think the comparison for the treatment induced changes is misleading. Ideally the authors should consider the same population with and without treatment; e.g. a population of patients with EGFR mutant tumours, and then explore the impact of treatment. I still think this section is confusing otherwise.

As previously mentioned, this a fair point for these cases where the treatment enriched drivers (TEDs) are also the primary target of the drug (e.g., as mentioned by the reviewer EGFR inhibitors are generally given to NSCLC patients with EGFR-mutant tumors). In this dataset, we identified three of such cases: Anti-EGFR in NSCLC EGFR-mutant tumors, Anti-BRAF in skin melanoma BRAF-mutant tumors and Anti-HER2 in breast carcinoma ERBB2 mutant tumors. For these cases, we further analyzed the presence of secondary target resistance mutations that were exclusively found in the treatment group. For instance, we found that EGFR^{T790M} were exclusively found in the metastatic Anti-EGFR treated cohort (18 EGFR^{T790M} samples mutated out of 78 patients treated with Anti-EGFR, none of them in the primary cohort bearing EGFR mutations, see table below). Moreover, we also found that there were ERBB2 alterations exclusively found in the treated cohort that were not present in the primary untreated dataset bearing ERBB2 alterations. Finally, we did not find treatment-exclusive secondary mutations in the Anti-BRAF cohort of patients in skin melanoma.

Overall, our results indicate that the global mutation frequency of *EGFR* and *ERBB2* in the treated cohorts can not solely be explained by the design of the analysis, and thus include secondary mutations associated with treatment resistance.

This analysis and further discussion of the limitations of the TEDs analysis are now integrated into the new Supp. Note 4 (partially pasted below):

“While the majority of TEDs are exclusively mutated in the treated cohort group, certain TEDs are *de-facto* original targets of the anti-cancer treatment. For instance, EGFR inhibitors are generally given to non-small cell lung cancer (NSCLC) patients bearing EGFR alterations. In our dataset, we detected three TEDs that are also the original targets of the cancer treatment: EGFR mutations in the Anti-EGFR therapy group in NSCLC, BRAF mutations in the Anti-BRAF therapy group in skin melanoma, and ERBB2 mutations in the Anti-ERBB2 therapy group of breast carcinoma patients. These cases are marked in Supp. Table 8 (*is_drug_target* column). Naturally, the identification of these TEDs is expected given that the presence of the mutated target is a prerequisite for the treatment initiation. However, it is important to consider that secondary target mutations are one of the most common mechanisms of resistance to targeted therapies. For instance, EGFR^{T790M} and EGFR^{C797S} mutations are the main mechanisms of resistance to *EGFR* TKIs^{1,2}, whereas secondary *BRAF*^{L514V} mutations may confer resistance to *BRAF* inhibitors³. Thus, to further provide insights into these three cases we have performed an additional analyses at the variant level, aiming at identifying alterations that were exclusively found in the treated group (i.e., the primary untreated group eligible for treatment did not harbor any of these alterations).

This analysis revealed that *EGFR* and *ERBB2* harbored secondary mutations that were exclusively found in the treated group (see table below, now part of the Supp. Table 8) (e.g., EGFR^{T790M}, EGFR^{C797S}, ERBB2^{V777L}, etc.) whereas we did not observe *BRAF* secondary mutations in our cohort. Of note, none of the secondary mutations were significantly enriched in the metastatic cohort, likely owing to the low sample size.

gene	mutation	drug	cancer_type	ed_found	total_s	total_s	odds_ratio	G-test	is_exclusive
					sample	sample			
					s_treat	s_untreat			
					cancer_ed_mut	ated_gene			
					type_co	ation_f			
					_mutat	utation			
					_found	ne_mut			
					ated	ted_ge			

ERBB2	L755S	Anti_HE R2	BRCA	2	12	0	3	5.0	0.78030 412050 1725	True
ERBB2	V777L	Anti_HE R2	BRCA	2	12	0	3	5.0	0.78030 412050 1725	True
EGFR	.	Anti_EG FR	LUAD	20	78	3	6	0.51282 051282 05128	0.64123 650528 49492	False
EGFR	A750P	Anti_EG FR	LUAD	2	78	0	6	1.53846 153846 15383	0.41850 480158 97672	True
EGFR	C797S	Anti_EG FR	LUAD	8	78	0	6	6.15384 615384 6153	0.97385 703825 57281	True
EGFR	G719A	Anti_EG FR	LUAD	2	78	0	6	1.53846 153846 15383	0.41850 480158 97672	True
EGFR	G719S	Anti_EG FR	LUAD	2	78	0	6	1.53846 153846 15383	0.41850 480158 97672	True
EGFR	L858R	Anti_EG FR	LUAD	12	78	0	6	9.23076 923076 9232	0.73724 326839 94703	True
EGFR	L861Q	Anti_EG FR	LUAD	2	78	1	6	0.15384 615384 615385	0.61924 653056 131	False
EGFR	T790M	Anti_EG FR	LUAD	18	78	0	6	13.8461 538461 53847	0.50210 767425 91722	True
BRAF	.	BRAF_i nhibitor	SKCM	2	13	15	56	0.57435 897435 89744	0.73573 832382 87619	False
BRAF	V600E	BRAF_i nhibitor	SKCM	11	13	31	56	1.52853 598014 88834	0.50262 229621 96717	False

Overall, our analysis suggests that the global mutation frequency of *EGFR* and *ERBB2* in the treated cohorts can not solely be explained by the design of the analysis, and thus include mutations associated with treatment resistance.”

- I appreciate the authors further exploration of SBS1 in metastatic samples. Could the authors also further evaluate the relationship when restricting to clonal/subclonal pre and post WGD mutations?

Regarding the impact of clonality on SBS1 mutations and its potential influence in our results (this is a point also raised by reviewer #2, see response to point 7.), we have now performed a primary vs. metastatic comparison of the age-adjusted SBS1 mutation burden restricted to clonal mutations, which according to our technical controls (Supp. Note 1) are marginally affected by differences in sequencing coverage (see also response to reviewer #2).

SBS1 clonal only mutations

As shown by the Figure above (now Fig. 2 in Supp. Note 3), restricting to clonal-only mutations rendered highly consistent patterns, suggesting that the observed age-adjusted SBS1 burden increase is not primarily explained by differences in somatic mutation detection sensitivity for subclonal mutations across the two cohorts.

Concerning the pre-post WGD SBS1 mutations, this analysis was partly covered in Supp. Note 3 clonality and timing section (Figures 8 and 9) where we compared the clonal late ratio of SBS1 mutations (i.e., ratio of SBS1 mutations acquired post-WGD and/or copy number gains compared to those acquired pre-WGD and/or copy number gains) between primary and metastatic tumors (see also response to reviewer #1 from previous rounds of reviews). Briefly, our analysis revealed that there was a significant enrichment of clonal late mutations (e.g., after copy number gains and/or WGD) in breast and kidney renal clear cell carcinomas. The results for prostate and thyroid were not clear, likely owing to the limited number of informative SBS1 mutations and primary tumor patients bearing WGD/copy number mutations in these cancer types. Thanks.

- I would avoid the term 'metastatic patients' - it is the cancer cells that are subject to metastasis not the patient.

The reviewer is absolutely right. We have corrected this in the manuscript and supplementary text.
Thanks

Reviewer Reports on the Second Revision:

Referees' comments:

Referee #2 (Remarks to the Author):

Thank you to the authors for replying to the specific technical comments in my 2nd review. I am satisfied with the further changes to the manuscript and the rebuttals. I have no additional comments or requests.

Referee #3 (Remarks to the Author):

The authors have done a good job of addressing all my concerns.